# ZDHHC20-mediated S-palmitoylation of YTHDF3 stabilizes MYC mRNA to promote pancreatic cancer progression

Huan Zhang [1], Yan Sun[2], Zhaokai Wang[1], Xiaoju Huang[3], Lu Tang [4] ✉, Ke Jiang [1] ✉ & Xin Jin [5,6] ✉

Post-translational modifications of proteins in malignant transformation and tumor maintenance of pancreatic ductal adenocarcinoma (PDAC) in the context of *KRAS* signaling remain poorly understood. Here, we use the KPC mouse model to examine the effect of palmitoylation on pancreatic cancer progression. ZDHHC20, upregulated by *KRAS*, is abnormally overexpressed and associated with poor prognosis in patients with pancreatic cancer. Dysregulation of ZDHHC20 promotes pancreatic cancer progression in a palmitoylation-dependent manner. ZDHHC20 inhibits the chaperone-mediated autophagic degradation of YTHDF3 through S-palmitoylation of Cys474, which can result in abnormal accumulation of the oncogenic product MYC and thereby promote the malignant phenotypes of cancer cells. Further, we design a biologically active YTHDF3-derived peptide to competitively inhibit YTHDF3 palmitoylation mediated by ZDHHC20, which in turn downregulates MYC expression and inhibits the progression of *KRAS* mutant pancreatic cancer. Thus, these findings highlight the therapeutic potential of targeting the ZDHHC20−YTHDF3−MYC signaling axis in pancreatic cancer.

Pancreatic ductal adenocarcinoma (PDAC) is a lethal solid tumor and causes considerable morbidity worldwide[1]. The incidence of PDAC has been increasing in recent decades[2]. Moreover, due to the lack of early disease diagnosis and the tendency of PDAC to metastasize and drug resistance, the mortality of PDAC is almost equal to its morbidity[3]. Oncogenic *KRAS* mutations are initiating events in pancreatic tumorigenesis, and these mutations, most commonly *KRAS G12D*, occur in more than 90% of pancreatic intraepithelial neoplasias (PanINs)[4–7]. Genetic control of malignant transformation and tumor maintenance in PDAC in the context of *KRAS* signaling remains largely unexplored[8]. Further elucidation of PDAC pathogenesis is essential for early diagnosis and identification of promising therapeutic targets for this cancer.

Protein activity relies on several modifications, including ubiquitination, phosphorylation, glycosylation and lipid modifications, after translation in the cytoplasm[9]. Palmitoylation, one kind of lipid modification, constitutes the attachment of the saturated 16-carbon fatty acid palmitate to specific cysteine residues[10]. The thioester linkages of S-palmitoylation are more common in vivo than N-palmitoylation. More importantly, palmitoylation is dynamically reversible, and protein activity can be regulated through palmitoylation and depalmitoylation, thereby regulating intracellular transport, targeted fusion and downstream signal transduction[11–13].

[1]Department of Thoracic Surgery, Union Hospital, Tongji Medical College, Huazhong University of Science and Technology, Wuhan 430022, China. [2]Department of Pancreatic Surgery, Union Hospital, Tongji Medical College, Huazhong University of Science and Technology, Wuhan 430022, China. [3]Cancer center, Union Hospital, Tongji Medical College, Huazhong University of Science and Technology, Wuhan 430022, China. [4]Institute of Hematology, Union Hospital, Tongji Medical College, Huazhong University of Science and Technology, Wuhan 430022, China. [5]Department of Urology, the Second Xiangya Hospital, Central South University, Changsha, Hunan 410011, China. [6]Uro-Oncology Institute of Central South University, Changsha, Hunan 410011, China. ✉e-mail: lu_tang@hust.edu.cn; kkkj_77@aliyun.com; jinxinxy2@csu.edu.cn

Palmitoylation is catalyzed mainly by palmitoyl acyltransferases (PATs), which belong to the aspartate–histidine–histidine–cysteine (DHHC) protein family and have a conserved DHHC catalytic domain[14]. ZDHHC20 is one of the DHHC protein family members that regulate protein-cysteine S-palmitoyltransferase activity and palmitoyltransferase activity. It has been reported that palmitoylation regulates the trafficking and functions of multiple cancer-related proteins[15–18]. However, it remains unclear whether palmitoylation may play a substantial regulatory role in the occurrence and development of pancreatic cancer.

As one of the main m6A readers, YTH domain-containing family protein 3 (YTHDF3) recognizes m6A-modified RNAs and then regulates their stability and translation[19,20]. To date, YTHDF3 has been reported to play an important role in the progression of a variety of tumors[21,22]. However, little is known about the regulatory mechanisms of YTHDF3 itself, especially its posttranslational modifications.

In the present study, we discover that ZDHHC20, a protein upregulated by *KRAS*, is abnormally overexpressed and predict an unfavorable prognosis in pancreatic cancer. Importantly, ZDHHC20 suppresses the lysosomal localization and degradation of YTHDF3 via palmitoylation of Cys474, which further regulates the mRNA stability of MYC through m6A modification. Thus, these findings identify the mechanism by which the ZDHHC20–YTHDF3–MYC axis promotes pancreatic cancer progression in a palmitoylation-dependent manner.

## Results

### Aberrant ZDHHC20 upregulation predicts unfavorable prognosis in pancreatic cancer

To examine whether palmitoylation affects the development of pancreatic cancer, we used the KPC (*LSL- KRAS G12D/+; LSL-TrpS3R172H/+; Pdx-1-Cre*) mouse model[23] to examine the effect of 2-bromopalmitate (2-BP) on pancreatic cancer progression (Fig. 1A). Notably, treatment with 2-BP, a general palmitoylation inhibitor, resulted in lower tumor burdens and longer survival times in KPC mice (Fig. 1B, D). To further evaluate the expression profile of palmitoyl acyltransferases in pancreatic cancer, we analyzed the expression of all known palmitoyl acyltransferases in pancreatic cancer based on the Gene Expression Profiling Interactive Analysis (GEPIA) web server[24]. Interestingly, multiple palmitoyl acyltransferases, including ZDHHC3, ZDHHC4, ZDHHC5, ZDHHC6, ZDHHC7, ZDHHC9, ZDHHC13, ZDHHC14, ZDHHC16, ZDHHC18 and ZDHHC20, were significantly upregulated in pancreatic cancer tissues compared to adjacent nontumor tissues (NATs) (Figs. 1E, S1A and S2), which may suggest that palmitoyl acyltransferases play an integral role in pancreatic cancer progression. We next explored the relationship between the expression of palmitoyl acyltransferase family members and patient survival in a pancreatic cancer dataset from The Cancer Genome Atlas (TCGA) (Fig. S3). High expression levels of ZDHHC20 were obviously correlated with unfavorable prognosis in patients with pancreatic cancer (Fig. 1F, G). Survival analysis based on data from the Tumor Immune Estimation Resource (TIMER) database further indicated that ZDHHC20 is a poor prognostic factor in pancreatic cancer (Fig. S1B)[25]. Further analysis of the GSE16515 dataset showed consistently increased mRNA levels of ZDHHC20 in pancreatic cancer specimens (Fig. 1H). The protein level of ZDHHC20 was additionally evaluated by immunohistochemical staining in a tissue microarray containing tumor tissue and NAT specimens from 29 pancreatic cancer patients. ZDHHC20 was abnormally upregulated in tumor tissues (Figs. 1I and S1C). We next examined the mRNA and protein expression levels of ZDHHC20 in pancreatic cancer tissues and NATs collected from our hospital. Our data showed higher protein and mRNA levels of ZDHHC20 in pancreatic cancer tissues than in NATs. These findings prompted us to investigate the regulatory role of ZDHHC20 in pancreatic cancer (Fig. 1J, K).

### *KRAS* mutations induce the accumulation of ZDHHC20 in pancreatic cancer via STAT3

The expression of ZDHHC20 in pancreatic cancer cell lines was significantly higher than that in normal pancreatic epithelial tissues (Fig. 2A). Interestingly, the expression level of ZDHHC20 was relatively low in the BxPC-3 cell line, a pancreatic cancer cell line without *KRAS* mutation[26], compared to other pancreatic cancer cell lines. Furthermore, our TIMER database analysis revealed that ZDHHC20 expression levels were higher in tumor tissues from pancreatic cancer patients harboring *KRAS* mutations than in those without *KRAS* mutation (Fig. 2B). *KRAS* oncogenic point mutations, which constitutively activate the RAS signaling pathway, affect the regulation of multiple cellular biological processes in pancreatic ductal adenocarcinoma, including cell proliferation, migration, metabolism and autophagy. In a comparison of *KRAS* ON with *KRAS* OFF paired samples in a recent study[27], we found that oncogenic *KRAS* mutations upregulated ZDHHC20 protein expression in pancreatic cancer (Fig. S4A). Hematoxylin and eosin (H&E) staining and immunohistochemical (IHC) staining of tissues from KPC mice indicated that the expression of ZDHHC20 was higher in PanIN tissues than in adjacent normal pancreatic tissues and that it increased further with the progression of pancreatic intraepithelial neoplasias (PanINs) to PDAC (Fig. 2C, D). To further investigate whether *KRAS* mutations result in high expression of ZDHHC20 in pancreatic cancer tissues, we treated pancreatic cancer cells with a KRAS G12D inhibitor. The results of RT–qPCR and western blot analyses showed that KRAS G12D inhibitor treatment significantly downregulated ZDHHC20 expression in pancreatic cancer cells (Figs. 2E and S4B). To confirm the accuracy of this finding and further investigate the mechanism by which KRAS regulates ZDHHC20 expression, we transfected the KRAS G12D plasmid into BxPC-3 (KRAS wild type) and CAPAN-1 (relatively low ZDHHC20 expression) cells. As expected, exogenous expression of KRAS G12D resulted in significantly upregulation of ZDHHC20 (Figs. 2F and S4C). To explore the mechanism underlying this upregulation, we used the KnockTF platform to analyze the transcription factors that target ZDHHC20. STAT3, an important component of the molecular program that is constitutively activated by *KRAS* mutations that drive PDAC progression[28–30], was found to be the transcription factor with the most significant regulatory effect on ZDHHC20 (Fig. S4D). Moreover, knockdown or inhibition of STAT3 significantly reduced ZDHHC20 expression, overexpression of STAT3 resulted in significantly upregulation of ZDHHC20 in pancreatic cancer cells (Figs. 2G, 2H, S4E–S4H). In addition, bioinformatic analysis showed a positive correlation between ZDHHC20 and STAT3 mRNA levels in various cancers, including pancreatic cancer (Fig. S4I, J). ChIP-seq analysis revealed the presence of a binding peak in the transcription initiation region of ZDHHC20 (Fig. 2I). The primers were designed based on the binding peak, and the ChIP–qPCR results demonstrated that STAT3 can bind to the promoter region of ZDHHC20 (Fig. 2J). Moreover, the enrichment of STAT3 at the promoter region of ZDHHC20 was enhanced by the exogenous expression of KRAS G12D (Fig. 2K). The dual-luciferase reporter assay indicated that the binding affinity decreased at the promoter region of ZDHHC20 after treatment with KRAS G12D or STAT3 inhibitor (Fig. 2L and S4K). Importantly, ZDHHC20 upregulation induced by exogenous KRAS G12D could be reversed by STAT3 inhibitors (Fig. 2M). Taken together, these findings indicate that *KRAS* mutations are one of major causative factors for STAT3-ZDHHC20 axis hyperactivation in pancreatic cancer (Fig. 2N).

### ZDHHC20 promotes pancreatic cancer progression in a palmitoylation-dependent manner

Given the abnormal expression of ZDHHC20 in pancreatic cancer tissues, we next explored the biological function of ZDHHC20 in pancreatic cancer cells. ZDHHC20 was knocked down with two different gene-specific shRNAs in PANC-1 and AsPC-1 cells (Figs. 3A, B), which

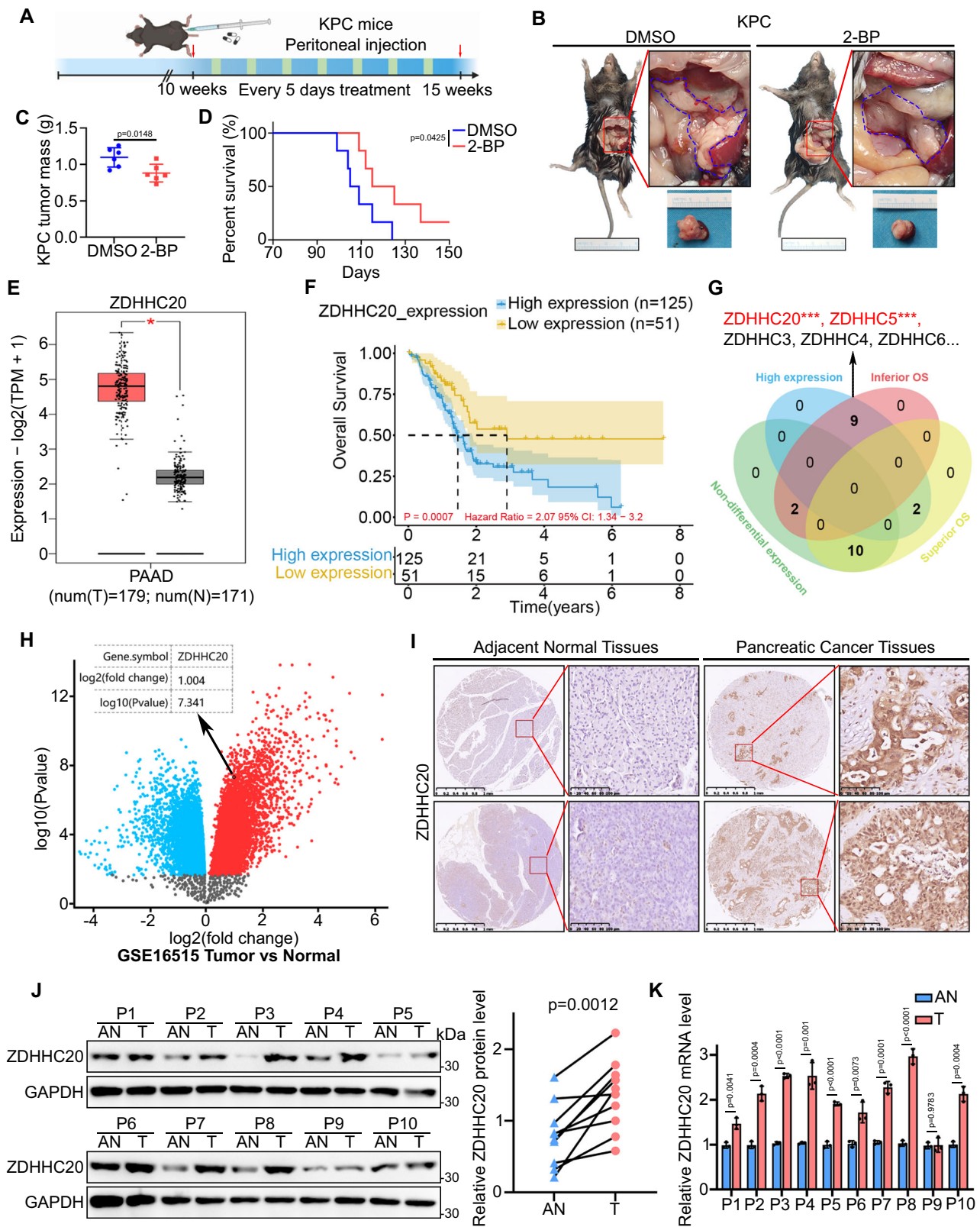

have relatively high expression of ZDHHC20. The CCK8 assay, colony formation assay and Transwell assay results suggested that ZDHHC20 silencing profoundly decreased the proliferation, invasion and migration of these pancreatic cancer cell lines (Figs. 3C–E and S5A). In addition, we found that the knockdown of ZDHHC20 inhibited tumor growth in vivo by employing a cell-derived xenograft (CDX) model (Figs. 3F and S5B–E). Next, we used the KPC mouse model, and

shControl and shZDHHC20 were injected into the mice as planned (Figs. 3G and S5F). Consistent with the in vitro findings, knockdown of ZDHHC20 resulted in a lower tumor weight and smaller pancreatic neoplastic lesion area and, notably, a longer survival time, in KPC mice (Fig. 3H–J). Together, these results indicate that ZDHHC20 functions as a tumor promoter in pancreatic cancer. Then, ZDHHC20 was over-expressed by ectopic transfection of ZDHHC20 expression plasmids

**Fig. 1 | Aberrant ZDHHC20 upregulation predicts unfavorable prognosis in pancreatic cancer. A–D** KPC mice at 10 weeks old, treated with DMSO or 2-BP every five days for 5 weeks, were euthanized, and pancreatic tumor were harvested for gross and histologic characterization ($n$ = 6 mice per group). **A** Schematic diagram for timeline of the treatment in the KPC mouse model, Created with BioRender.com released under a Creative Commons Attribution-NonCommercial-NoDerivs (CC-BY-NC-ND) 4.0 International license. **B** Representative photos of KPC mice treated with DMSO or 2-BP. **C** The comparison of pancreas mass in KPC mice after DMSO or 2-BP treatment. $n$ = 6, two-tailed unpaired $t$ test. **D** Survival analysis of KPC mice (10 weeks old) treated with DMSO or 2-BP every five days for 5 weeks ($n$ = 6 mice per group), Log-rank test. **E** The tissue-wise expression of ZDHHC20 in Pancreatic adenocarcinoma (PAAD) tissues and non-tumor tissues were analyzed by the GEPIA web tool, two-tailed Wilcoxon signed rank test. T Tumor, N Normal, *$P$ < 0.01. The box plots are defined in terms of median, upper quartile, lower quartile and 95% confidence interval. **F** ZDHHC20-related Overall Survival in a pancreatic cancer dataset from TCGA. The log-rank (Mantel-Cox) test for survival analysis. **G** Venn diagram showed the relationship between the expression of palmitoyl acyltransferase family members and patient survival (***$P$ < 0.001) in TCGA. **H** Volcano plot analysis of GSE16515 data sets yielded the same results: ZDHHC20 was significantly upregulated in pancreatic cancer tissues compared to normal tissues. $P$ < 0.001. **I** Representative images of IHC analysis for ZDHHC20 on TMA containing a cohort of pancreatic cancer samples (AN and PDAC:$n$ = 29 biologically independent samples). **J** Western blot analysis for the expression of ZDHHC20 in 10 paired adjacent normal tissues (AN) and PDAC tissues (T) of the same patient. The ImageJ software was used to quantify the protein expression level of ZDHHC20. two-tailed paired $t$ test. **K** RT-qPCR analysis for the expression of ZDHHC20 in paired adjacent normal tissues (AN) and PDAC tissues (T) of the same patient. $n$ = 10 biologically independent samples examined over 3 independent experiments, two-tailed unpaired $t$ test. Data are presented as mean ± SD. Source data are provided as a Source Data file.

into BxPC-3 and CAPAN-1 cells, which have relatively low expression of ZDHHC20 (Fig. S5G). ZDHHC20 overexpression promoted pancreatic cancer cell proliferation, invasion and migration. Intriguingly, 2-BP treatment counteracted and even reversed the promotive effects of ZDHHC20 overexpression on cell proliferation and invasion in vitro (Figs. 3K, 3L, S5H and S5I). Considering the function of ZDHHC20 as a palmitoyl acyltransferase, we wondered whether the oncogenic capacity of ZDHHC20 is dependent on its palmitoylation activity. A previous study reported that two residues, Cys156 and Phe171, in the acyl-binding cavity of ZDHHC20 are essential for its catalytic activity[31]. Therefore, we constructed two ZDHHC20 mutants (C156S and F171A) without significant catalytic activity and then ectopically transfected BxPC-3 and CAPAN-1 cells with plasmids expressing wild-type ZDHHC20 and the two mutants. Expression of the two mutants considerably decreased the catalytic palmitoylation activity of ZDHHC20 and inhibited the proliferation and invasion of pancreatic cancer cells (Figs. 3M, 3N, S5J and S5K). Considering the function of STAT3–ZDHHC20 axis in *KRAS* mutant pancreatic cancer above, we explore the biological effect of these proteins in the CDX model. The knockdown or inhibitor of STAT3 also observably suppress tumor growth in vivo, but the combination with shZDHHC20 don't further enhance this effect. This may suggest that STAT3 plays a promoting role in pancreatic cancer progression, perhaps in part by inducing overexpression of ZDHHC20 (Fig. S5L–Q). Hence, these results suggest that ZDHHC20 may promote pancreatic cancer progression in a manner at least partially dependent on palmitoylation.

## YTHDF3 mediates the oncogenic capacity of ZDHHC20 in pancreatic cancer

Sliver staining for gel electrophoresis of Flag-ZDHHC20-immunoprecipitation in PANC-1 cells was conducted to show potential interactions based on molecular weight (Fig. 4A). To investigate the specific mechanisms by which ZDHHC20 regulates pancreatic cancer progression, we conducted immunoprecipitation–tandem mass spectrometry (IP-MS) to identify potential binding partners of ZDHHC20. The IP-MS assay showed that there are potential interactions between ZDHHC20 and YTHDF3, an $N^6$-methyladenosine (m6A) reader (Fig. 4B, C). It has been reported that YTHDF3 is overexpressed in multiple kinds of tumors and promotes tumor progression, but its regulatory role in pancreatic cancer has not been explored. HA-ZDHHC20 and Flag-YTHDF3 expression plasmids were transfected into 293T cells, and the co-immunoprecipitation (Co-IP) assay showed the interaction of exogenously expressed ZDHHC20 with YTHDF3 (Fig. 4D). Then, we also detected the interaction of ZDHHC20 with YTHDF3, not YTHDF1 or YTHDF2, in PANC-1 and BxPC-3 cells (Figs. 4E, F and S6A). In addition, immunofluorescence confocal microscopy showed that ZDHHC20 and YTHDF3 were colocalized in the cytoplasm (Fig. 4G). Moreover, the co-immunoprecipitation Co-IP assay of YTHDF3 and all ZDHHCs suggested that YTHDF3 was not a target of

any other ZDHHC protein in PANC-1 cells (Fig. S6B). To further confirm the interaction between ZDHHC20 and YTHDF3, the Proximity Ligation Assay (PLA) was conducted (Fig. S6C), and the results indicated the transient interaction of endogenous ZDHHC20 with YTHDF3 in the cytoplasm of PANC-1 cells (Fig. 4H). We also generated recombinant YTHDF3 protein and performed a GST pull-down assay to identify the regions of YTHDF3 modified by ZDHHC20-mediated palmitoylation (Fig. 4I). Our data showed that the C-terminal domain but not the N-terminal domain in YTHDF3 interacted with ZDHHC20 (Fig. 4J). Given that ZDHHC20 acts as a palmitoyl acyltransferase, we speculated that YTHDF3 might be a critical substrate for ZDHHC20. Therefore, we further explored whether ZDHHC20 promotes pancreatic cancer progression via YTHDF3. Knockdown of YTHDF3 inhibited the progression of pancreatic cancer and counteracted and even reversed the promoting effects of ZDHHC20 overexpression on tumor growth and invasion in vitro and in vivo (Figs. 4K–M and S6D–G), implying that YTHDF3 may play a crucial role in mediating the oncogenic capacity of ZDHHC20 in pancreatic cancer.

## ZDHHC20 regulates the palmitoylation of YTHDF3 in pancreatic cancer

To determine whether YTHDF3 is palmitoylated, we detected palmitoylation of endogenous YTHDF3 in pancreatic cancer cells by click chemistry and acyl-biotin exchange (ABE) (Fig. 5A, B). In addition, the omission of HAM treatment abolished the palmitoylation of YTHDF3, and YTHDF3 palmitoylation levels were significantly reduced after treatment with 2-BP, confirming that YTHDF3 is S-palmitoylated through thioester bonds (Fig. 5C). To determine whether ZDHHC20 mediates the palmitoylation of YTHDF3, a set of lentiviral plasmids expressing guide RNAs (gRNAs) targeting ZDHHC20 was constructed for thorough knockout. As expected, knockout of ZDHHC20 significantly reduced palmitoylation of YTHDF3, suggesting that ZDHHC20 is a YTHDF3-palmitoylating enzyme (Fig. 5D). We tried to identify the cysteine residues of YTHDF3 that are S-palmitoylated by ZDHHC20. First, the motif-based prediction tool CSS-palm 4.0 predicted a single palmitoylation site in YTHDF3 (Cys474) in a sequence conserved across multiple species (Fig. 5E). Notably, this uniquely predicted site is located in the C-terminal domain of YTHDF3, also the binding region of ZDHHC20. Subsequently, we mutated Cys474 to serine in the YTHDF3 protein (C474S). Substitution of the Cys474 residue with serine through mutagenesis markedly blocked the palmitoylation of exogenous YTHDF3 (Figs. 5F, G). In line with this finding, reconstitution with the Flag-tagged sgRNA-resistant (r) YTHDF3 C474S mutant in endogenous YTHDF3-knockout (KO) PANC-1 and BxPC-3 cells completely abolished the palmitoylation of YTHDF3 (Fig. 5H, I). In addition, two ZDHHC20 mutants without catalytic activity (C156S and F171A) couldn't mediate the palmitoylation of YTHDF3 as the wild-type (WT) ZDHHC20 (Fig. 5J). Based on the positive effect of the ZDHHC20–YTHDF3 axis on pancreatic cancer progression, we next

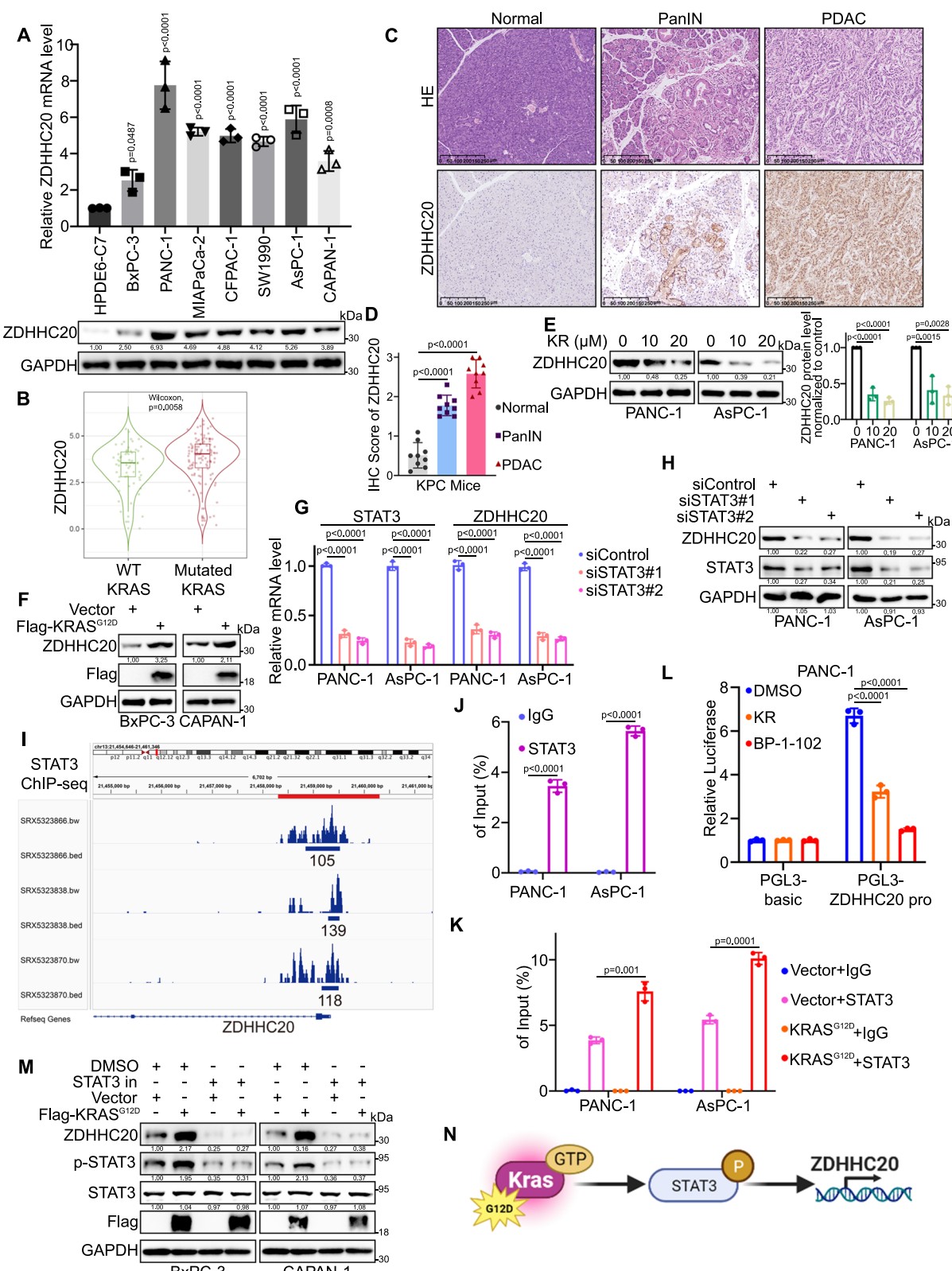

explored whether palmitoylation of Cys474 plays a key role in this process. To this end, we performed a clone formation assay of PANC-1 and AsPC-1 cells expressing YTHDF3 WT and the C474S mutant and found decreased proliferation of pancreatic cancer cells expressing the C474S mutant (Fig. 5K). Similarly, in vivo fluorescence imaging also showed suppression of pancreatic tumor growth in mice implanted with cells expressing the YTHDF3-C474S mutant (Fig. 5L). Taken

together, our data indicate that ZDHHC20-mediated palmitoylation of YTHDF3 on Cys474 promotes pancreatic cancer progression.

## ZDHHC20 suppresses the degradation of YTHDF3 via palmitoylation on Cys474

For a more comprehensive understanding of YTHDF3 palmitoylation, we tried to evaluate the potential effect of YTHDF3 palmitoylation on

**Fig. 2 | KRAS mutations induce the accumulation of ZDHHC20 in pancreatic cancer via STAT3. A** The mRNA and protein levels of ZDHHC20 were examined, $n = 3$ biologically independent experiments, one-way ANOVA. **B** TIMER database analysis showed the different expression of ZDHHC20 in WT-*KRAS* and Mutated-*KRAS* pancreatic cancer $n = 170$, two-tailed Wilcoxon signed rank test. The violin plots are defined in terms of median, upper quartile, lower quartile and 95% confidence interval, and the size of the violin indicates the density of the data distribution. Representative images (**C**) and IHC score (**D**) of ZDHHC20 in normal, PanINs (PanIN 1-3) and pancreatic tissues in KPC mice. $n = 9$, one-way ANOVA. **E** Western blot analysis for the expression of ZDHHC20 in PANC-1 and AsPC-1 cells treated with KRASG12D inhibitor (KR: KRpep-2d), $n = 3$ biologically independent experiments, one-way ANOVA. **F** Western blot analysis for the expression of ZDHHC20 in BxPC-3 and CAPAN-1 cells infected with KRASG12D plasmids. **G** RT-qPCR analysis for the mRNA expression level of ZDHHC20 and STAT3 in PANC-1 and AsPC-1 cells infected with STAT3 siRNAs, $n = 3$ biologically independent experiments, one-way ANOVA. **H** Western blot analysis for the expression of ZDHHC20 and STAT3 in PANC-1 and AsPC-1 cells infected with STAT3 siRNAs.

**I** ChIP-seq data of STAT3 indicated that there were binding peaks in the promoter region of ZDHHC20. **J** ChIP-qPCR assay of ZDHHC20 by using the IgG or STAT3 antibodies in PANC-1 and AsPC-1 cells, $n = 3$ biologically independent experiments, two-tailed unpaired $t$ test. **K** ChIP-qPCR assay of ZDHHC20 by using the IgG or STAT3 antibodies in PANC-1 and AsPC-1 cells infected with Vector/KRASG12D plasmids, $n = 3$ biologically independent experiments, two-tailed unpaired $t$ test. (**L**) The dual-luciferase reporter assay for the promoter region binding affinity of ZDHHC20 in PANC-1 cells treated with KRpep-2d (20 μM) or BP-1-102 (20 μM), $n = 3$ biologically independent experiments, one-way ANOVA. (**M**) Western blot analysis of BxPC-3 and CAPAN-1 cells treated with DMSO/STAT3 inhibitor and infected with Vector/KRASG12D plasmids. **N** Model depicting the mechanism that *KRAS* mutations induce ZDHHC20 upregulation through activation of STAT3 (Created with BioRender.com released under a CC-BY-NC-ND 4.0 International license). Statistical data presented in results (**A, D, E, G, J–L**) show mean values ± SD. Similar results for (**A, E, F, H,** and **M**) panels were obtained in three independent experiments. Source data are provided as a Source Data file.

its subcellular localization. Nuclear/cytosolic fractionation assays showed that YTHDF3 was localized mainly in the cytosol, as previously reported[21]. However, knockout of ZDHHC20 did not relocalize YTHDF3 from the cytosol to the nucleus (Fig. S7A, B). Then, we found that downregulation of ZDHHC20 reduced only the protein level of YTHDF3, while the mRNA level of YTHDF3 was not significantly altered (Fig. 6A, B). As a reversible post-translational modification, palmitoylation regulates protein trafficking, interaction and degradation[11]. We next evaluated whether palmitoylation by ZDHHC20 regulates YTHDF3 degradation. We employed 2-BP to depalmitoylate endogenous YTHDF3 in BxPC-3 and AsPC-1 cells treated with the protein synthesis inhibitor cycloheximide (CHX) and detected the effects of various inhibitors of different degradation pathways (Fig. S7C). 2-BP treatment caused destabilization of YTHDF3, which was rescued by the lysosomal inhibitors $NH_4Cl$, bafilomycin A1, chloroquine and leupeptin but not the proteasomal inhibitor Carfilzomib (Figs. 6C and S7D–F). Furthermore, knockout of ZDHHC20 resulted in destabilization of YTHDF3, which was also rescued by lysosomal inhibitors (Figs. 6D and S7G–I). We next performed immunofluorescence staining of endogenously expressed YTHDF3 in BxPC-3 cells with or without ZDHHC20 knockout and observed more lysosomal localization of YTHDF3 in the ZDHHC20-KO cells (Figs. 6E and S8A). Similarly, western blot analysis of fractionated cellular lysates also revealed that knockout of ZDHHC20 increased the protein level of YTHDF3 in lysosomes (Fig. 6F). Notably, the immunofluorescence staining assay also confirmed that the YTHDF3 C474S mutation induced an increase in the lysosomal localization of exogenous YTHDF3 (Figs. 6G and S8B). The substitution of the Cys474 residue with serine through mutagenesis significantly accelerated the degradation of exogenous YTHDF3 in cells treated with CHX (Figs. 6H and S8C). However, the YTHDF3 WT exhibited instability similar to that of the YTHDF3 C474S in the ZDHHC20-KO cells (Figs. 6I and S8D).

To further explore the potential mechanisms for the decreased lysosomal localization of YTHDF3 after S-palmitoylation, we conducted IP-MS between the YTHDF3 WT and YTHDF3 C474S. The IP-MS assay showed that HSC70, a molecular chaperone recognized soluble cytosolic proteins bearing a KFERQ-like motif and mediated the translocation of the substrate into the lysosome for lumen degradation[32], showed more interactions with the YTHDF3 C474S relative to the YTHDF3 WT (Fig. S8E, F). Analysis of the YTHDF3 amino acid sequence revealed the presence of three potential KFERQ-like motifs, suggesting that YTHDF3 could be a chaperone-mediated autophagy (CMA) substrate (Fig. S8G). Consistently, the Co-IP assays also corroborated more interactions of HSC70 with YTHDF3 C474S (Fig. 6J). Notably, we found that YTHDF3 protein levels were downregulated in ZDHHC20-KO pancreatic cancer cells, and the knockdown of HSC70 or LAMP2A reversed the effects of ZDHHC20 deficiency on

YTHDF3 protein levels in ZDHHC20-KO cells (Fig. S8H, I). In conclusion, ZDHHC20-mediated palmitoylation of YTHDF3-Cys474 suppressed the recognition of YTHDF3 by HSC70, thereby inhibiting its subsequent lysosomal degradation through the CMA pathway (Fig. 6K).

## YTHDF3 stabilizes MYC mRNA in an m6A-dependent manner

To elucidate the mechanism by which the ZDHHC20–YTHDF3 axis promotes the progression of pancreatic cancer, we carried out RNA-seq analysis in PANC-1 cells with ZDHHC20 silencing (Fig. 7A). Principal component analysis (PCA) revealed a striking difference between the shZDHHC20 and control groups (Fig. S9A). Gene Ontology (GO) enrichment analysis also suggested that ZDHHC20 regulated protein trafficking in lysosomes, vesicles and the extracellular space (Fig. S9B), consistent with the previous results. In addition, we analyzed the RIP-seq and RNA-seq data of YTHDF3 from the GSE130173 dataset and found 95 potential direct targets of YTHDF3. Notably, only MYC was consistently regulated by ZDHHC20 and YTHDF3, suggesting that MYC could be the critical target of the ZDHHC20–YTHDF3 axis (Figs. 7B, 7C, S9C, D). Indeed, the RIP assay confirmed the direct binding of YTHDF3 to the MYC transcript (Fig. 7D). Consistent with these findings, YTHDF3 knockdown in BxPC-3 and AsPC-1 cells decreased the mRNA level of MYC, while ectopic expression of YTHDF3 in PANC-1 and CAPAN-1 cells increased the mRNA level of MYC (Fig. 7E, F). Similarly, the regulatory effect of YTHDF3 on the MYC protein level was validated by further western blotting (Fig. 7G, H), suggesting that YTHDF3 palmitoylation is closely associated with MYC expression.

We then examined the mRNA stability of MYC with Actinomycin D treatment. YTHDF3 knockdown in BxPC-3 and AsPC-1 cells significantly reduced the half-life[33] of MYC mRNA (Figs. 7I, J). Next, we tried to explain how YTHDF3 regulates MYC mRNA stability. It has been proved that the coding region instability determinant (CRD), an approximately 250 nucleotide cis-acting element located in the 3′-terminus of the MYC coding region, is critical for MYC mRNA stability[34,35]. Previous studies have reported that HuR and IGF2BP1 (CRD-BP) recognize MYC-CRD and modulate MYC stability. Therefore, we performed immunoprecipitation and surprisingly found that HuR and IGF2BP1 interact with YTHDF3 (Fig. S9E). The data suggested an unreported function of YTHDF3 in recruiting these RNA stabilizers. However, palmitoylation of Cys474 did not seem to affect this function of YTHDF3 (Fig. S9E). In parallel, we knocked out HuR and IGF2BP1 in PANC-1 and AsPC-1 cells (Fig. S9F, G). However, HuR/IGF2BP1 KO did not reverse the enhancement of MYC mRNA stability by YTHDF3 overexpression (Fig. S9H, I). Considering that YTHDF3 functions as an m6A reader to regulate the translation and decay of N6-methyladenosine-modified RNA[22], YTHDF3 may directly

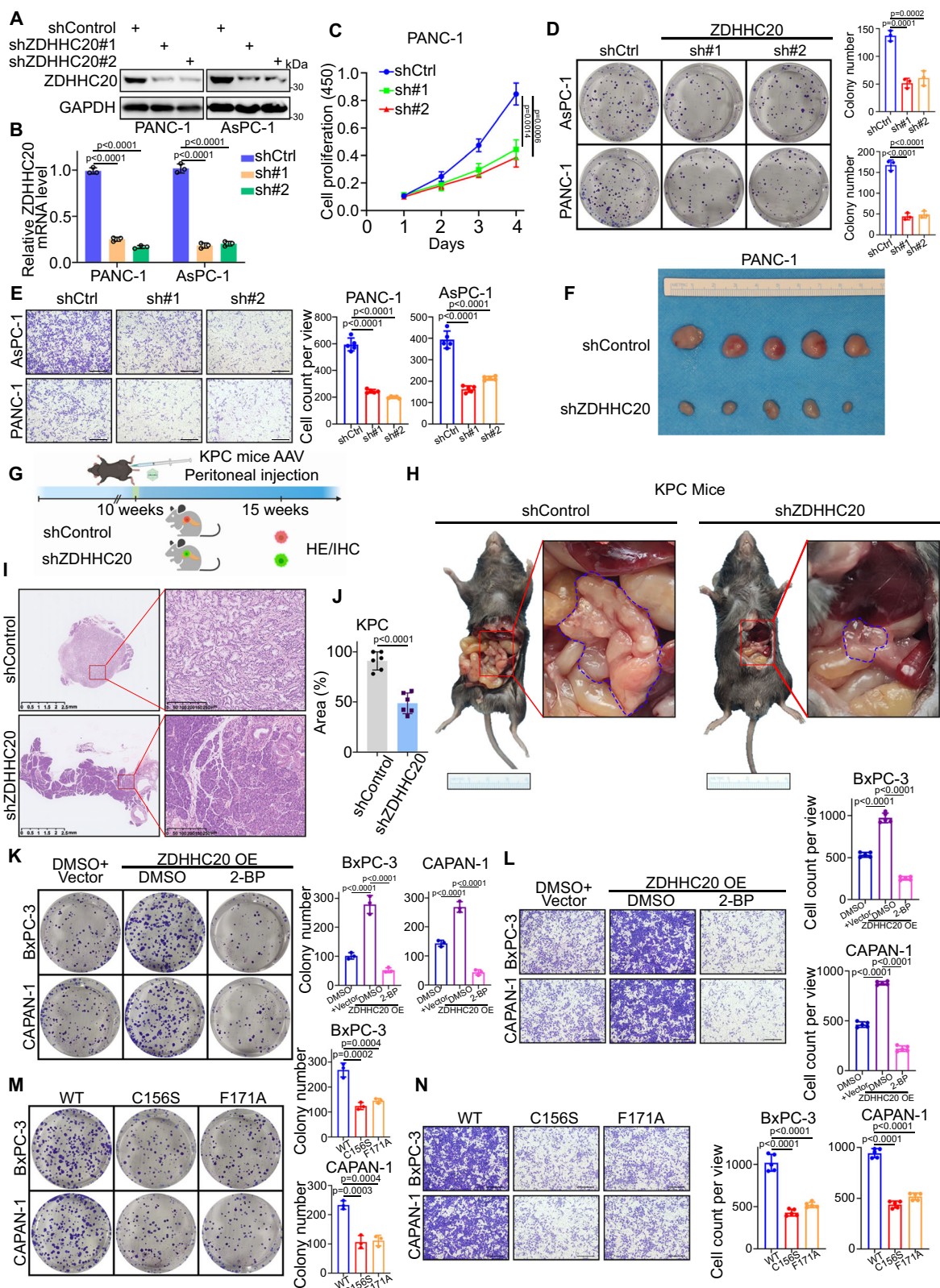

keep MYC mRNA stability. As shown in RIP-seq and MeRIP-seq in the research of YTHDF3, m6A modification accumulated across the MYC transcript, and both the binding peaks and m6A peaks coincided well with the CRD domain (Fig. 7K, L). As expected, we confirmed m6A modification recognized by endogenous YTHDF3 in the MYC CRD in PANC-1 and AsPC-1 cells (Fig. 7M, N). We next mutated all potential m6A sites in the MYC CRD[35,36] and inserted the

corresponding wild-type or mutated sequences into a firefly luciferase (Fluc) reporter plasmid (Fig. S10A, B). Notably, YTHDF3 overexpression significantly increased the Fluc activity of the wild-type reporter but not the mutant reporters (Figs. 7O and S10C). Consistent with these findings, RIP-qPCR analysis showed that YTHDF3 strongly interacted with the wild-type reporter and hardly bound the mutant reporters (Fig. 7P). Interestingly, we found that

**Fig. 3 | ZDHHC20 promotes pancreatic cancer progression in a palmitoylation-dependent manner. A** Western blot analysis for the expression of ZDHHC20 in PANC-1 and AsPC-1 cells infected with ZDHHC20 shRNAs, similar results were obtained in three independent experiments. **B** RT-qPCR analysis for the mRNA expression level of ZDHHC20 in PANC-1 and AsPC-1 cells infected with ZDHHC20 shRNAs, $n = 3$ biologically independent experiments, one-way ANOVA. PANC-1 and AsPC-1 cells infected with lentivirus vectors expressing control or ZDHHC20 specific shRNAs were harvested for CCK-8 assay, $n = 3$ biologically independent experiments, two-way ANOVA (**C**); colony formation assay, $n = 3$ biologically independent experiments, one-way ANOVA (**D**); Transwell invasion assay, scale bars: 100 μm, $n = 5$ biologically independent experiments, one-way ANOVA (**E**). **F** PANC-1 cells infected with lentivirus vectors expressing control or ZDHHC20 specific shRNAs Cells were injected subcutaneously into the nude mice for xenografts assay. **G** Schematic diagram for timeline of the AAV-shControl and AAV-shZDHHC20 treatment in the KPC mouse model (Created with BioRender.com released under a CC-BY-NC-ND 4.0 International license). Representative photos of KPC mice treated with AAV-shControl or AAV-shZDHHC20 (**H**); representative images of H&E staining (**I**); percentage of pancreatic neoplastic area (**J**), $n = 6$ per group, two-tailed unpaired $t$ test. BxPC-3 and CAPAN-1 cells infected with lentivirus vectors expressing control or ZDHHC20 treated with DMSO or 2-BP were harvested for colony formation assay, $n = 3$ biologically independent experiments, one-way ANOVA (**K**); Transwell invasion assay, scale bars: 100 μm, $n = 5$ biologically independent experiments, one-way ANOVA (**L**). BxPC-3 and CAPAN-1 cells infected with lentivirus vectors expressing ZDHHC20 WT/C156F/F174A were harvested for colony formation assay, $n = 3$ biologically independent experiments, one-way ANOVA (**M**); Transwell invasion assay, scale bars: 100 μm, $n = 5$ biologically independent experiments, one-way ANOVA (**N**). Statistical data presented in results (**B, D, E, G, J–N**) show mean values ± SD. Source data are provided as a Source Data file.

YTHDF3-mediated upregulation of MYC mRNA level was rescued by blockade of palmitoylation with 2-BP (Fig. 7F). Thus, we carried out the dual-luciferase reporter assay in pancreatic cancer cells and found that 2-BP treatment or ZDHHC20 knockout significantly downregulated the Fluc activity of the wild-type reporter (Figs. 7Q and S10D). Considering that ZDHHC20-mediated palmitoylation occurs in the YTH domain, which has been reported to function in binding to RNA, we further explored whether palmitoylation regulates the YTHDF3-MYC mRNA interaction. In the case of blocking the lysosomal degradation pathway of YTHDF3 by Barf A1, 2-BP treatment or ZDHHC20 knockout still significantly down-regulated the binding of YTHDF3 to MYC mRNA (Fig. S10E, F). In addition, the RIP-qPCR and MeRIP-qPCR analysis showed that substitution of the Cys474 blocked m6A modification recognized by YTHDF3 in the MYC mRNA (Fig. S10G, H). Taken together, our results reveal that m6A modifications in the MYC CRD are required for YTHDF3-mediated stabilization of MYC mRNA in pancreatic cancer.

## ZDHHC20-mediated palmitoylation of YTHDF3-Cys474 stabilizes MYC mRNA

We next investigated whether ZDHHC20 promotes pancreatic cancer progression by affecting the biological function of MYC. Intriguingly, analyses using the TIMER web tool demonstrated a robust positive association between the mRNA level of MYC and those of ZDHHC20, STAT3 and YTHDF3 (Fig. S11A–C). Consistent with previous findings, we evaluated the expression levels of p-STAT3–ZDHHC20–YTHDF3–MYC axis and Ki67 in PDAC tissue microarrays by IHC staining and observed consistently positive correlations (Figs. 8A and S11D–H). In addition, blockade of palmitoylation by 2-BP significantly reduced the expression of MYC in both a dose-dependent and time-dependent manner (Fig. 8B, C). To confirm the role of ZDHHC20 in MYC expression in PDAC, we evaluated the effect of 2-BP treatment and ZDHHC20 knockout on the mRNA stability of MYC. Our data showed that blockade of YTHDF3 Cys474 palmitoylation decreased the half-life of MYC mRNA (Figs. 8D, E and S12A–D). Further western blotting demonstrated that ectopic ZDHHC20 expression upregulated MYC expression, an effect that was blocked by YTHDF3 knockout (Fig. 8F, G). In addition, the YTHDF3 C474S mutation did not increase MYC expression as YTHDF3 WT did. However, after palmitoylation blockade by 2-BP treatment or ZDHHC20 knockout, there was no significant difference in MYC expression between the C474S and WT groups (Fig. 8H, I). Then, we examined whether ZDHHC20 promotes PDAC progression through MYC. As expected, MYC silencing reversed the promotive effect of ectopic ZDHHC20 expression on PDAC both in vitro and in vivo (Fig. S12E, G). Altogether, our findings suggest that ZDHHC20-mediated palmitoylation of YTHDF3-Cys474 regulates the mRNA stability of MYC to promote PDAC progression.

## Therapeutic blockade of the ZDHHC20-YTHDF3 interaction inhibits pancreatic cancer progression

Our finding that the palmitoylation of YTHDF3-Cys474 mediated by ZDHHC20 stabilizes MYC suggest that inhibition of this specific modification might have therapeutic potential for PDAC through targeting of the ZDHHC20–YTHDF3–MYC signaling axis. Based on the identified palmitoylation motif of YTHDF3, the YTHDF3 (467–481) sequence containing C474 (GFP-Y1) or the C474S mutation (Y2), was fused to the sequence of green fluorescent protein (GFP) to competitively suppress the palmitoylation of endogenous YTHDF3 (Figs. 9A and S13A). Further Click-iT pulldown demonstrated that GFP-Y1 but not GFP-Y2 was palmitoylated (Fig. 9B). Importantly, we found that GFP-Y1 but not GFP-Y2 decreased the palmitoylation of YTHDF3 (Fig. 9C). Consistent with these findings, ectopic expression of GFP-Y1 but not GFP-Y2 significantly reduced the expression of endogenous MYC, and MYC expression was rescued by a selective lysosome inhibitor (Baf-A1) but not a selective proteasome inhibitor (Carfilzomib) (Figs. 9D, S13B and S13C). Generally, these results showed that GFP-Y1 suppressed the palmitoylation and induced the lysosome-dependent degradation of endogenous YTHDF3, thereby significantly decreasing the expression of endogenous MYC.

On the basis of previous results, we designed cell-penetrating peptides (CPPs) to introduce a competitive inhibitor of YTHDF3 palmitoylation into PDAC cells. A CPP was fused to the Y1 (CPPtat-Y1) and Y2 (CPPtat-Y2) sequences to induce efficient molecular transport (Fig. 9E). When PANC-1 and AsPC-1 cells were treated with different concentrations of the CPPtat-Y1 peptide, the endogenous expression of MYC decreased in a dose-dependent manner (Fig. 9F). In contrast, no change in MYC expression was found in cells incubated with the control CPPtat-Y2 peptide (Fig. 9G). Importantly, CPPtat-Y1 treatment significantly inhibited pancreatic cancer cell proliferation and invasion (Fig. S13D-F). In addition, we used the KPC transgenic mouse model of spontaneously initiated pancreatic cancer to further study the effect of the CPPtat-Y1 peptide in vivo. Control CPPtat or CPPtat-Y1 was injected into KPC mice following the relevant protocol (Fig. 9H). Consistent with previous in vitro results, CPPtat-Y1 treatment significantly suppressed PDAC progression and reduced the area of pancreatic lesions in KPC mice (Figs. 9I–K and S13G). And we found that the expression of YTHDF3 and MYC in KPC mouse tumors was significantly decreased after CPPtat-Y1 treatment (Figs. 9L, S13H, I). Notably, the survival time of KPC mice was significantly longer after CPPtat-Y1 treatment, confirming the antitumor effect of CPPtat-Y1 (Fig. 9M).

## Discussion

In recent years, many advances have been made in the functional study of protein palmitoylation, and DHHC protein family member-mediated palmitoylation affects tumor progression, the immune environment and drug resistance through multiple mechanisms[37–39]. In addition, it has been found that palmitoylation of PD-L1 by

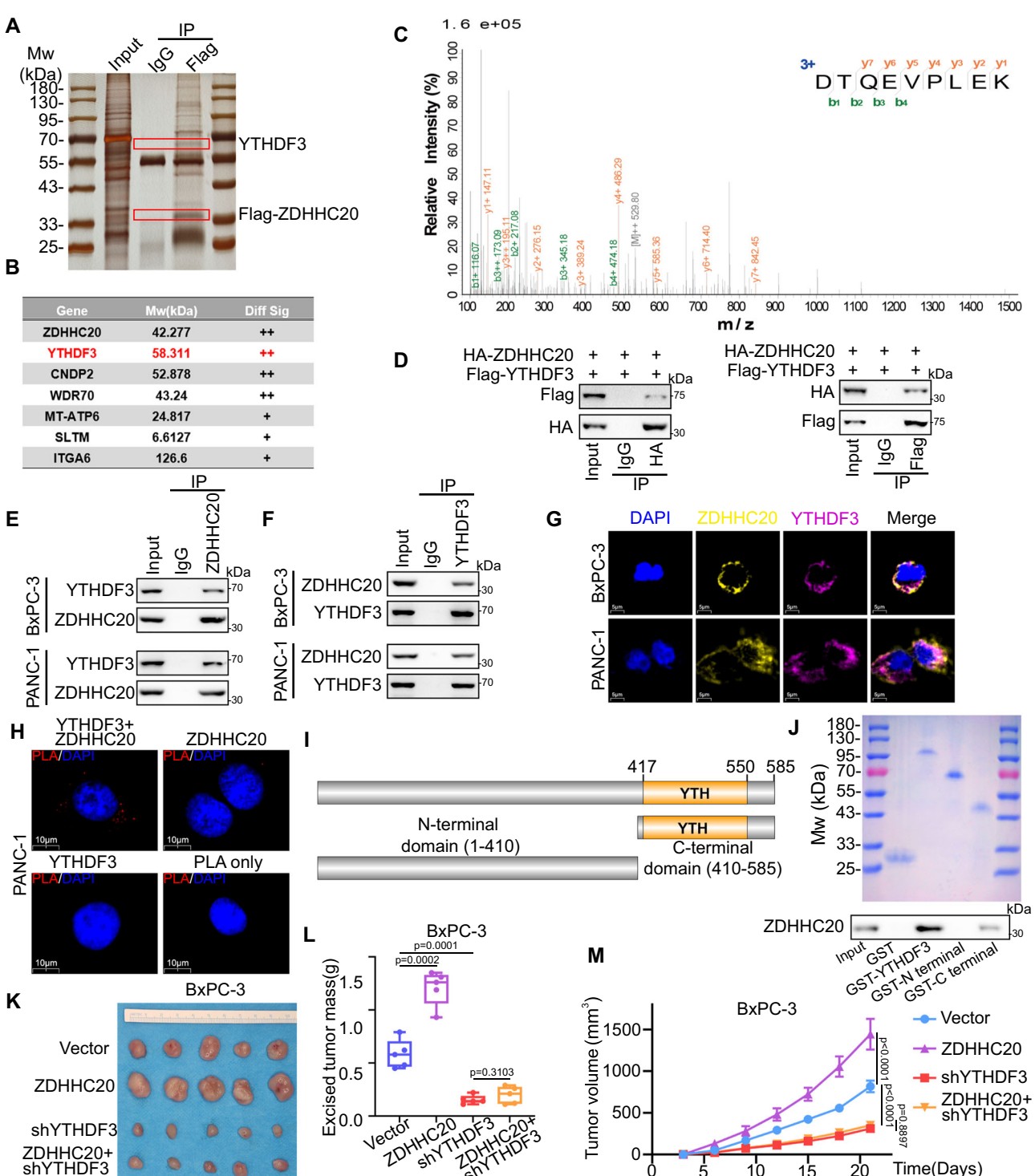

**Fig. 4 | YTHDF3 mediates the oncogenic capacity of ZDHHC20 in pancreatic cancer. A** Gel electrophoresis was conducted after using Flag antibody for Flag-ZDHHC20-immunoprecipitation, then sliver staining was performed. The red frame indicated the YTHDF3 and FLAG-ZDHHC20 predicted based on molecular weight. **B** Mass spectrometry analysis of Flag-ZDHHC20-immunoprecipitates in PANC-1 cell. **C** Mass spectrometry analysis of a peptide derived from Flag-ZDHHC20-immunoprecipitates to show the potential interaction between YTHDF3 and ZDHHC20. **D** Western blot analysis of ectopically expressed Flag-YTHDF3 and HA-ZDHHC20 reciprocally immunoprecipitated by anti-HA and anti-Flag in 293T cells. **E, F** Western blot analysis of endogenous ZDHHC20 and YTHDF3 proteins reciprocally immunoprecipitated by anti-ZDHHC20 and anti-YTHDF3 in PANC-1 and BxPC-3 cells. **G** Immunofluorescence confocal microscopy showed the colocalization of ZDHHC20 and YTHDF3 in PANC-1 and BxPC-3 cells. **H** The proximity ligation assay by using the indicated antibodies to verify the interaction between ZDHHC20 and YTHDF3 in PANC-1 cells. **I** Schematic diagram of YTHDF3 and its truncation mutants. **J** Western blot analysis of ZDHHC20 GST-pulldown by GST-YTHDF3 recombinant. **K–M** Representative images of xenografts assay that BxPC-3 cells was infected with lentivirus and injected subcutaneously into the left flank of nude mice, $n = 5$ biologically independent mice. Tumors were harvested, photographed, and weighed at day 22, one-way ANOVA (**L**); Tumor volumes were measured every 3 days, two-way ANOVA (**M**). The box plots are defined in terms of minima, maxima, centre, bounds of box and whiskers and percentile. Data are presented as mean ± SD. Similar results for (**A, D–H, J**) panels were obtained in three independent experiments. Source data are provided as a Source Data file.

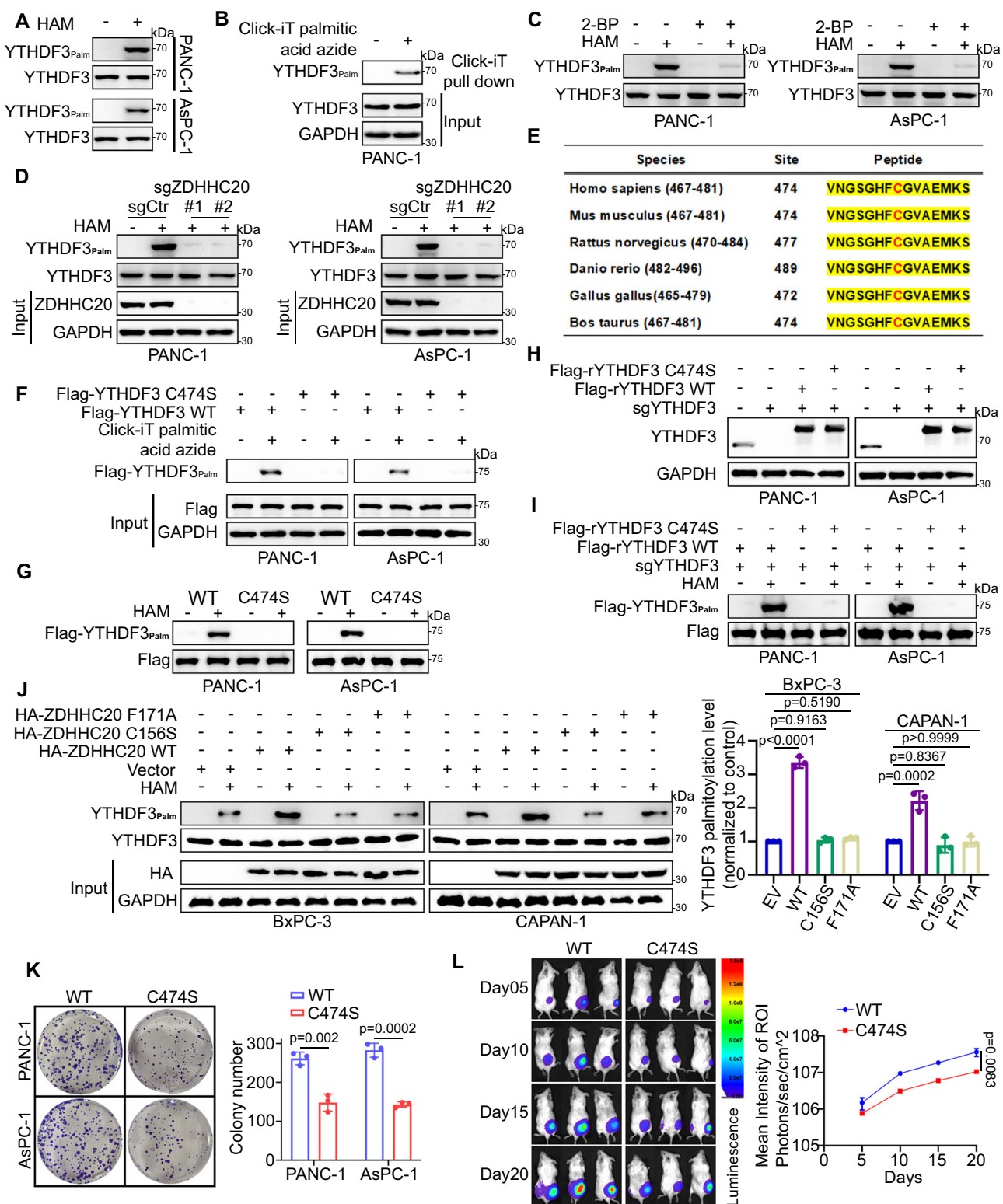

DHHC3 suppresses the monoubiquitination and blocks the degradation of PD-L1, increasing its expression in colorectal cancer and inducing suppression of T-cell cytotoxicity[40]. ZDHHC9-mediated GLUT1 S-palmitoylation, which maintains the plasma membrane localization of GLUT1, is critical for glucose supply during glioblastoma multiforme (GBM) tumorigenesis[17]. It has also been reported that ZDHHC12 promotes the stability of CLDN3 by affecting its cell membrane localization, which in turn promotes ovarian cancer progression[15]. However, the biological function of

palmitoylation in pancreatic cancer progression is still poorly understood. Our findings reveal the regulatory role of ZDHHC20 in the occurrence and development of pancreatic cancer. In our study, *KRAS* mutations could induce ZDHHC20-mediated S-palmitoylation of YTHDF3, thereby suppressing lysosomal localization and degradation of YTHDF3 and regulating MYC mRNA stability to promote PDAC progression. On this basis, we designed a biologically active YTHDF3-derived peptide to competitively inhibit YTHDF3 palmitoylation mediated by ZDHHC20, which in turn downregulated MYC

**Fig. 5 | ZDHHC20 regulates the palmitoylation of YTHDF3 in pancreatic cancer.**
**A** YTHDF3 palmitoylation was detected in PANC-1 and AsPC-1 cells using streptavidin-HRP/anti YTHDF3 after immunoprecipitation with anti-YTHDF3 and ABE assay. **B** PANC-1 cells with or without ALK-C16 treatment were harvested for Click-IT reaction and streptavidin pulldown. **C** YTHDF3 palmitoylation was detected in PANC-1 and AsPC-1 cells with or without 2-BP treatment using streptavidin-HRP/anti YTHDF3 after immunoprecipitation with anti-YTHDF3 and ABE assay. **D** YTHDF3 palmitoylation was detected in PANC-1 and AsPC-1 cells with or without ZDHHC20 knockout using streptavidin-HRP/anti YTHDF3 after immunoprecipitation with anti-YTHDF3 and ABE assay. **E** Prediction of YTHDF3 palmitoylation at Cys474 by the CSS-palm 4.0 and alignment of the similarity of YTHDF3 sequences across multiple species. **F** PANC-1 and AsPC-1 cells infected with Flag-YTHDF3 WT/C474S plasmids with or without ALK-C16 treatment were harvested for Click-IT reaction and streptavidin pulldown. **G** Flag-YTHDF3 palmitoylation was detected in PANC-1 and AsPC-1 cells infected with Flag-YTHDF3 WT/C474S plasmids using streptavidin-HRP/anti YTHDF3 after immunoprecipitation with anti-Flag and ABE assay.
**H** Western blot analysis of PANC-1 and AsPC-1 cells infected with Flag-rYTHDF3 WT/C474S after endogenous YTHDF3 knockout, r: sgRNA-resistant. **I** Western blot analysis of PANC-1 and AsPC-1 cells infected with Flag-rYTHDF3 WT/C474S after endogenous YTHDF3 knockout, using streptavidin-HRP/anti-Flag after immunoprecipitation with anti-Flag and ABE assay. **J** Western blot analysis of PANC-1 and AsPC-1 cells infected with ZDHHC20 WT/C156S/F171A plasmids, $n = 3$ biologically independent experiments, statistical data are presented as mean ± SD, one-way ANOVA. **K** PANC-1 and AsPC-1 cells infected with Flag-YTHDF3 WT/C474S were harvested for colony formation assay, each bar represents the mean ± SD of three independent experiments, two-tailed unpaired $t$ test. **L** PANC-1-luc cells infected with YTHDF3 WT/C474S were harvested for xenografts assay and bioluminescence imaging in vivo. $n = 3$, two-way ANOVA. Statistical data presented in results (**J**–**L**) show mean values ± SD. Similar results for (**A**–**D**, **F**–**J**) panels were obtained in three independent experiments. Source data are provided as a Source Data file.

expression and inhibited the progression of *KRAS* mutant pancreatic cancer (schematic in Fig. 10).

When analyzing ZDHHC20 in pancreatic cancer tissues compared to adjacent nontumor tissues, we found that the protein expression difference of ZDHHC20 by IHC is more significant than the mRNA expression difference from the GSE16515 dataset. Phosphorylation, ubiquitination, and palmitoylation modifications have been reported for other ZDHHCs[41,42], and we conjecture that post-translational modifications of the ZDHHC20 protein in pancreatic cancer may lead to the significance of the mRNA-protein expression difference and may further amplify its carcinogenicity. Meanwhile, when evaluating the prognosis significance of all ZDHHCs in pancreatic cancer, we found great variability in the survival rate of the patients based on the expression of different ZDHHCs. It may suggest the complexity of different ZDHHCs regulating the biological functions of pancreatic cancer[43], and we will also further explore the post-translational modifications and functions of ZDHHCs in pancreatic cancer and try to elucidate this complex regulatory network.

N[6]-methyladenosine, one of the most essential modifications in RNA, has been reported to be involved in many biological processes, especially tumor progression[44–46]. The palmitoylation of YTHDF3 at Cys474 was identified in our study. This posttranslational modification is regulated by ZDHHC20 and maintains the protein stability of YTHDF3, which is essential in the development of pancreatic cancer. These results extend our understanding of the regulatory mechanisms of m6A modification in cancer.

MYC is overexpressed in approximately 70% of malignancies, induces tumor progression and is associated with poor prognosis[47–49]. However, MYC has long been considered an undruggable target due to the lack of a typical active pocket and a fixed conformation[50]. It has been reported that several m6A modification-related proteins (IGF2BPs, etc.)[35,51,52] promote the stability of MYC mRNA and play an essential oncogenic role in various kinds of cancers and likely also in the YTHDF3- recognized, m6A-dependent regulation of MYC mRNA stability in pancreatic cancer. Our study reveals that YTHDF3 plays a role in tumorigenesis by recognizing m6A modifications in MYC mRNA. Excitingly, a possible biologically active YTHDF3-derived peptide was designed to correct the dysregulation of MYC and initially showed some therapeutic potential.

Our study still has some limitations that need to be addressed in future study. First, methods for direct detection of palmitoylation have not been developed, and our study mainly combined Click-IT and ABE approaches to identify palmitoylation modifications of YTHDF3 mediated by ZDHHC20. In addition, palmitoylation is a dynamic and reversible modification, and the balance between palmitoylation and depalmitoylation of YTHDF3 in pancreatic cancer needs further exploration. Because Cys474 is in close vicinity of one the residues of the tryptophan-cage of YTHDF3, the binding surface responsible for the coordination of the N6-methyl moiety of m6A[53], we don't know

whether palmitoylation and C to S mutant may cause a conformational change in the tryptophan cage. To answer these questions, more biochemical, structural, and in silico studies on the YTHDF3 proteins are required. Furthermore, to fully understand the biological role of ZDHHC20 and YTHDF3 in pancreatic cancer, future studies using pancreas-specific deficiency of ZDHHC20 or YTHDF3 KPC mice model are warranted.

In conclusion, ZDHHC20, upregulated by *KRAS*, is abnormally overexpressed and associated with poor prognosis in patients with pancreatic cancer. Dysregulation of ZDHHC20 expression promotes pancreatic cancer progression in a palmitoylation-dependent manner. ZDHHC20 inhibits chaperone-mediated autophagy of YTHDF3 through S-palmitoylation of Cys474, which can result in abnormal accumulation of the oncogenic product MYC and thereby support the malignant phenotypes of cancer cells. These findings also identify the oncogenic roles of YTHDF3 as an m6A reader and highlight the therapeutic potential of targeting the ZDHHC20–YTHDF3–MYC signaling axis in pancreatic cancer.

## Methods
### Cell lines and culture
The pancreatic cancer cell lines (SW1990, AsPC-1, MIAPaCa-2, CFPAC1, PANC-1, CAPAN-1 and BxPC-3), normal human pancreatic duct epithelial cells (HPDE6-C7) and HEK293T were purchased from the ATCC (USA), National Collection of Authenticated Cell Cultures (Shanghai, China) and Procell Life Science&Technology Co., Ltd (Wuhan, China). Their identities were verified by short tandem repeat DNA sequencing analysis. Cells were cultured in DMEM (Gibco, USA) supplemented with 10% fetal bovine serum (FBS, Gibco, USA) and 1% Penicillin–Streptomycin (Thermo Fisher Scientific, USA) at 37 °C in 5% ambient $CO_2$, and tested regularly for mycoplasma-free growth.

### Transient transfection of plasmids
Plasmids used for gene overexpression and silencing were purchased from Shanghai Gene Chem Co., Ltd. Cells were grown in culture plates or dishes at a suitable density for 24 h. Cell transfection with plasmids was carried out using lipofactamine 2000 (#11668019, Thermo Fisher Scientific, USA) according to the manufacture's instructions. The sequences of all short-hairpin RNAs (shRNAs) are listed in Supplementary Table S1.

### Immunohistochemistry (IHC)
Tissue microarrays were purchased from Outdo Biotech (Shanghai, China) (HPanA060CS03). Immunohistochemical analysis was performed with antibodies specific for ZDHHC20 (#TD4335, working dilution 1:500, Abmart), YTHDF3 (#25537-1-AP, working dilution 1:1000, Proteintech), and MYC (#10828-1-AP, working dilution 1:1000, Proteintech). The IHC scores were determined by two independent

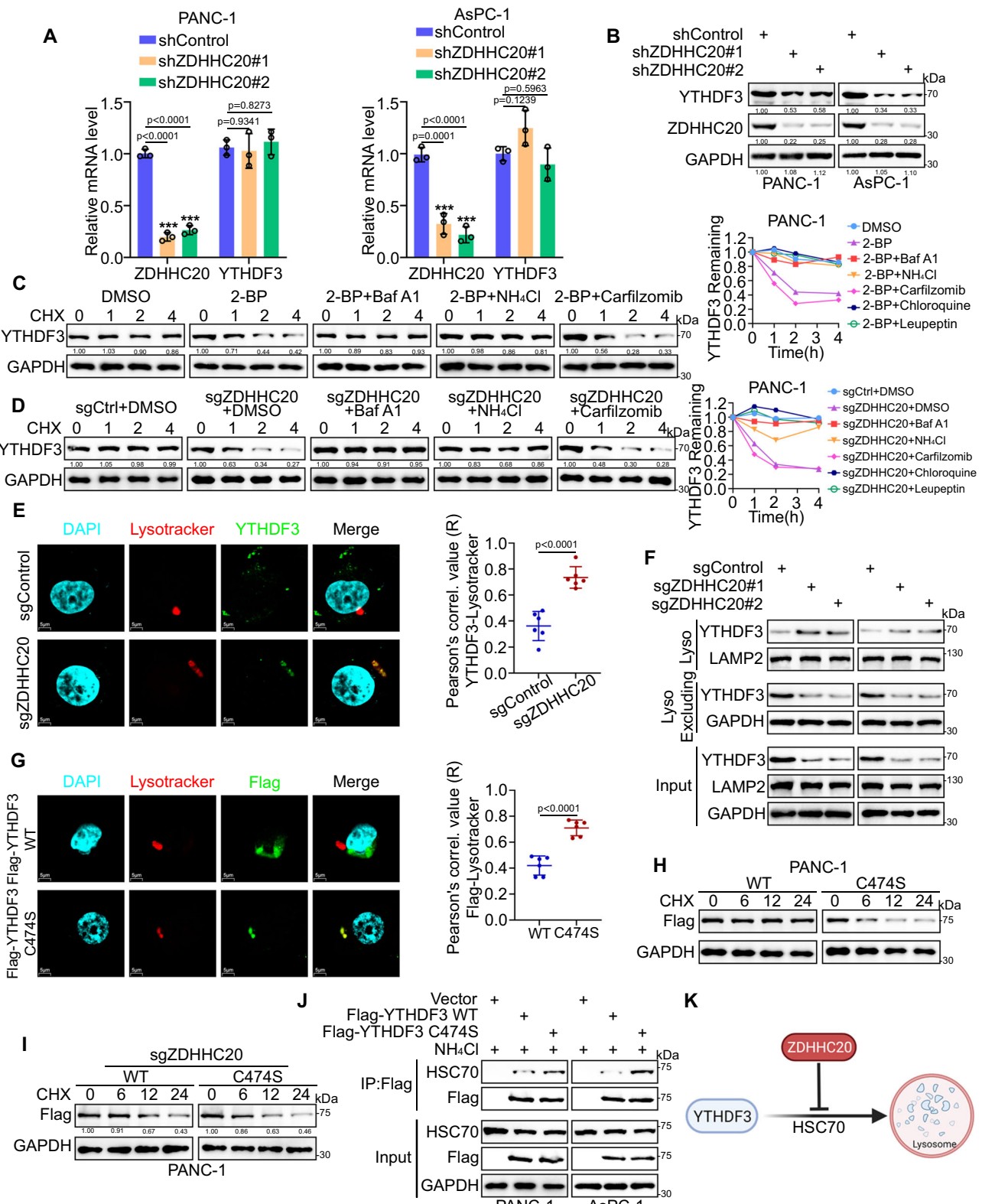

pathologists who were blinded to the patient data. The IHC scores were calculated according to intensity (0/1/2/3) and positive rate (0–100% positive cells). The intensity and positive scores were multiplied to form an IHC score.

**Lentiviral packaging and CRISPR-mediated gene knockout**

The oligonucleotides targeting ZDHHC20 were annealed and subcloned into lenti-CRISPRv2 vectors (Addgene). The sgRNA vectors were then transfected into HEK293T cells, together with pMDG.1 and psPAX2 vectors (Addgene), using Lipofectamine 2000 Reagent (Thermo Fisher Scientific, USA) following the manufacturer's suggested protocols. Media was replaced at 24 h, and the viral supernatant was collected at 72 h post transfection with high-speed ultra-centrifugation at 80,000 g at 4 °C for 2 h. The transduced cells from single guide RNAs were selected with appropriate concentration of puromycin (Thermo Fisher Scientific, USA) for 5–7 days to achieve stable

**Fig. 6 | ZDHHC20 suppresses the degradation of YTHDF3 via palmitoylation on Cys474. A** RT-qPCR analysis for the mRNA expression level of ZDHHC20 and YTHDF3 in PANC-1 and AsPC-1 cells infected with ZDHHC20 shRNAs, $n = 3$ biologically independent experiments, one-way ANOVA. **B** Western blot analysis for the expression of ZDHHC20 and YTHDF3 in PANC-1 and AsPC-1 cells infected with ZDHHC20 shRNAs. **C** Left, the degradation of YTHDF3 in PANC-1 cells treated with or without 2-BP was evaluated by CHX-chase assay in the presence of lysosomal inhibitors (bafilomycin A1 and $NH_4Cl$) and proteasomal inhibitor Carfilzomib. Right, quantification of the intensity determined by the relative level of YTHDF3 remaining. **D** Left, the degradation of YTHDF3 in PANC-1 cells with or without ZDHHC20 knockout was evaluated by CHX-chase assay in the presence of lysosomal inhibitors (bafilomycin A1 and $NH_4Cl$) and proteasomal inhibitor Carfilzomib. Right, quantification of the intensity determined by the relative level of YTHDF3 remaining. **E** Representative images of YTHDF3, Lysotracker and DAPI immunofluorescence staining in PANC-1 cells with or without ZDHHC20 knockout. Scale bar: 5 μm; $n = 6$ biologically independent experiments, two-tailed unpaired $t$ test. **F** Western blot analysis of fractionated cellular lysates evaluated the protein level of YTHDF3 in and outside lysosome in PANC-1 cells with or without ZDHHC20 knockout. **G** Representative images of Flag, Lysotracker and DAPI immunofluorescence staining in PANC-1 cells infected with Flag-YTHDF3 WT/C474S plasmids. Scale bar: 5 μm, $n = 6$ biologically independent experiments, two-tailed unpaired $t$ test. **H** The degradation of Flag-YTHDF3 WT/C474S in PANC-1 cells infected with Flag-YTHDF3 WT/C474S plasmids was evaluated by CHX-chase assay. **I** The degradation of Flag-YTHDF3 WT/C474S in ZDHHC20-KO PANC-1 cells infected with Flag-YTHDF3 WT/C474S plasmids was evaluated by CHX-chase assay. **J** Western blot analysis of ectopically expressed Flag-YTHDF3 WT/C474S and HSC70 reciprocally immunoprecipitated by anti-Flag in PANC-1 and AsPC-1 cells treated with $NH_4Cl$ (20 mM). **K** Model depicting the mechanism that ZDHHC20-mediated palmitoylation of YTHDF3-Cys474 suppress the lysosomal localization and degradation of YTHDF3 (Created with BioRender.com released under a CC-BY-NC-ND 4.0 International license). Statistical data presented in results (**A, E, G**) show mean values ± SD. Similar results for (**B, C, D, F, H, I, J**) panels were obtained in three independent experiments. Source data are provided as a Source Data file.

integration. The sgRNA sequences for CRISPR/Cas9 mediated-gene knockout were presented in Supplementary Table S1.

### Colony formation assay
Pancreatic cancer cells were seeded in 6-well plates (500 cells in 2 mL DMEM with 10% FBS per well). After two weeks, cells were fixed with 4% paraformaldehyde for 30 min, stained with 1% Crystal Violet Staining Solution (#C0121, Beyotime) for another 20 min, and then washed three times with PBS.

### CCK-8 assay
Pancreatic cancer cells were seeded in 96-well plates (3000 cells in 200 μl DMEM containing 10% FBS per well for 5 days). The CCK-8 reagent (#C0037, beyotime, 20 μl) was then added to each well one hour before the end of the incubation period following the manufacturer's instructions, and the optical absorbance of each well at 450 nm was measured using a microplate reader.

### Quantitative real-time PCR (RT-qPCR)
Total RNA was extracted using Trizol reagent (#15596026, Invitrogen, USA), and reverse transcribed into complementary DNA using PrimeScript RT reagent kit (#RR047A, Takara, Japan). Then, TB Green™ Fast qPCR Mix kit (#RR430A, Takara, Japan) was used to carry out real-time quantitative polymerase chain reaction (RT-qPCR). GAPDH served as the reference gene and the $2^{-\Delta\Delta Ct}$ method was used to quantify fold changes. The primer sequences used for RT-qPCR are presented in Supplementary Table S2.

### Western blot assay
Cells were harvested and lysed by modified radio immunoprecipitation assay lysis buffer containing phosphatase inhibitor and 1% protease on ice for 30 min. Then cell lysates were centrifuged at 13400 $g$ for 15 min at 4 °C and collect the supernatants. Protein concentration was determined using a protein quantification kit.

(#P0012S, Beyotime) to make sure the equal amounts of total protein were loaded in each well of SDS-PAGE gels. Protein lysates were separated by SDS-PAGE gels and transferred onto PVDF membranes. Proteins were next incubated with specific antibodies and detected by Enhanced chemiluminescence assay. Antibodies used for western blot are presented in Supplementary Data 1.

### Extraction of lysosomal proteins
Lysosome enrichment was performed following the manufacturer's instructions (Thermo Fisher Scientific, USA). Briefly, $5 \times 10^7$ pancreatic cancer cells were dissociated using the indicated reagents of the kit and were subjected to ultrasonic treatment. The cell lysate was separated by ultra-centrifugation at 145000 $g$ and 4 °C. The samples were analyzed after adding Laemmli buffer by immunoblotting.

### Immunofluorescence assay
Pancreatic cancer cells were incubated with Lyso-Tracker (Thermo Fisher Scientific, USA) for 20 min, fixed with paraformaldehyde for 15 min, and then permeabilized with 0.2% Triton X-100 for 10 min. Cells were incubated with primary antibodies at 4 °C overnight, followed by incubation with fluorescent secondary antibodies for one hour. Cell nuclei were stained with DAPI for 10 min and then were analyzed by confocal microscopy (Andor, Dragonflfly, 633 objective lens). Antibodies used for immunofluorescence are presented in Supplementary Data 1.

### RNA-pulldown assays
Biotin-labelled RNA oligonucleotides containing adenosine or m6A were synthesized by Sangon Biotin (Shanghai, China). The sequence of biotinylated probe was listed in Supplementary Table S2. RNA probes were denatured at 95 °C for 5 min and put on ice immediately. The M-280 streptavidin magnetic beads (Invitrogen, USA) at 25 °C for 2–4 h, and then the cell lysates with Protease/Phosphatase Inhibitor Cocktail and RNase inhibitor added were incubated with MYC probe or oligo probe at 4 °C overnight. The RNA complexes bound to the beads were eluted and extracted and then were quantitatived by qRT-PCR, and the RNA–protein binding mixture was boiled in SDS buffer and the eluted proteins were detected by western blotting.

### Co-immunoprecipitation (Co-IP)
Pancreatic cancer cells were lysed in Western/IP lysis buffer (Beyotime, China) on ice for 30 min, and the lysate was centrifuged at 13400 $g$ and 4 °C for 10 min. The primary antibodies and protein A/G agarose beads (Thermo Fisher Scientific) were added into the supernatant, and samples were incubated overnight at 4 °C. The next day, the beads were collected by centrifugation and washed six times with Western/IP lysis buffer. The beads were boiled for 10 min after resuspending in SDS-PAGE loading buffer and then subjected to western blotting analysis with Secondary antibody Mouse Anti-Rabbit IgG (Light-Chain Specific) monoclonal antibody. Specific antibodies used for Co-IP are presented in Supplementary Data 1.

### Click-iT identification of YTHDF3 palmitoylation
Pancreatic cancer cells were incubated with 100 mM of Click-iT palmitic acidazide for 6 h. After incubation, cells were lysed to extract proteins. Click-iT Protein Reaction Buffer Kit (catalog number C10276; Thermo Fisher Scientific) was used to catalyze the reaction of protein samples with biotin-alkyne. Biotin alkyne-

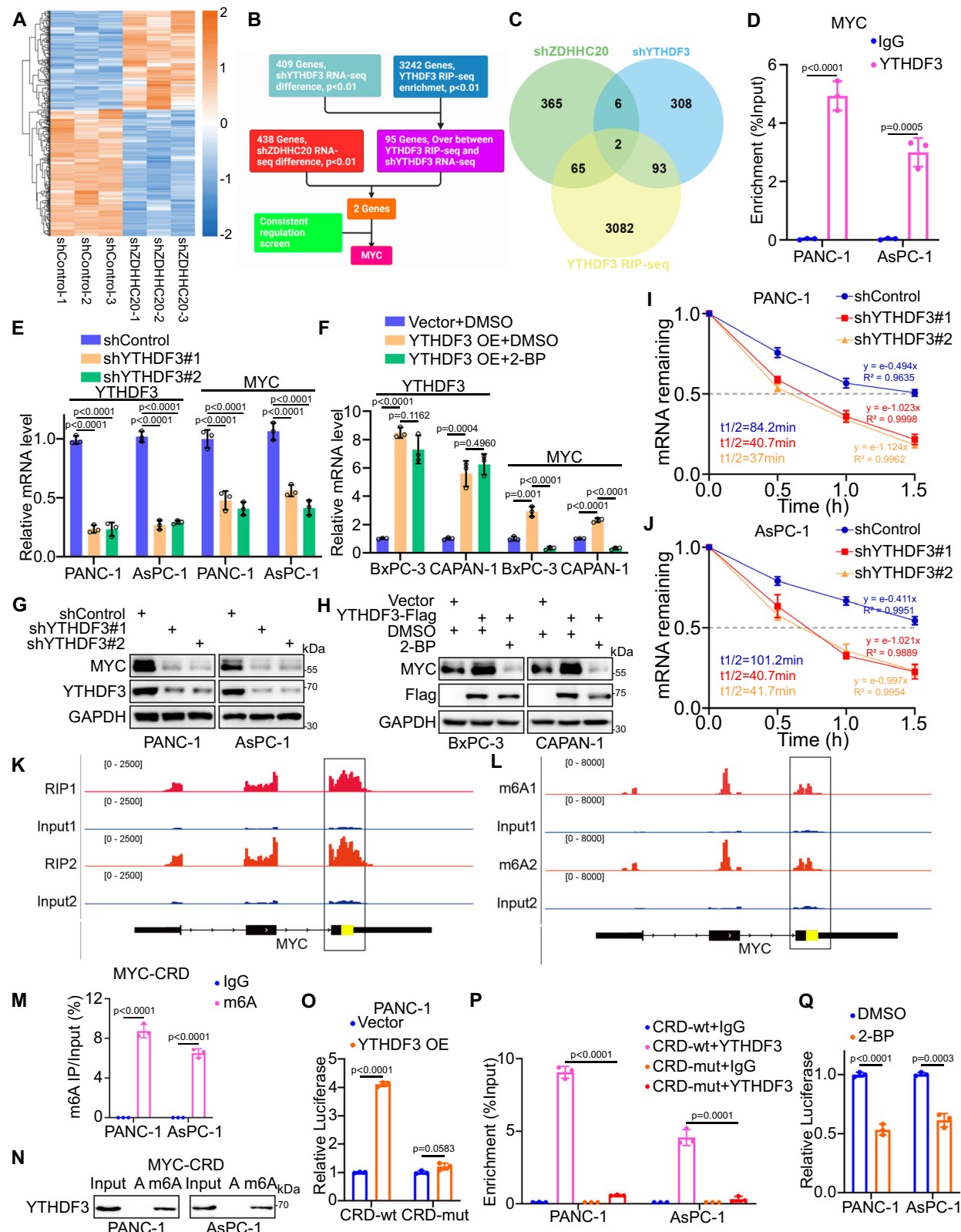

azide–palmitic-protein complex was precipitated with streptavidin, and samples were immunoblotted for YTHDF3.

### Acyl-biotinyl exchange (ABE) assay

The ABE assay was performed using the IP-ABE Palmitoylation Kit For WB (catalog number AM10313; AIMSMASS, Shanghai, China) according to the manufacturer's instructions. Briefly, the procedure includes

blocking, reduction, labelling, elution and detection. Cells were harvested and suspended in lysis buffer followed by incubation with beads and anti-YTHDF3 or anti-Flag overnight at 4 °C. N-ethylmaleimide (NEM) was used to block the unmodified cysteines for 30 min. Then, the beads were washed and incubated with hydroxylamine (HAM) for 1 h at room temperature. Each group was divided into two parts, one including the HAM step (+ HAM) and one omitting the HAM cleavage

**Fig. 7 | YTHDF3 stabilizes MYC mRNA in an m6A-dependent manner.**
**A** Supervised hierarchical clustering analysis of RNA-seq in PANC-1 cells with
ZDHHC20 silencing. Heat-map showed differential mRNA expression of 438 genes
with $p < 0.01$. **B** Schematic workflow of YTHDF3 downstream targets analysis.
**C** Venn diagram showed the targets analysis of the RIP-seq and RNA-seq data of
YTHDF3 from GSE130173 dataset and the RNA-seq of ZDHHC20. RNA-seq under the
criteria that $P < 0.01$ and |Fold Change| ≥1.5. **D** RIP-qPCR assay of MYC by using the
IgG or YTHDF3 antibodies in PANC-1 and AsPC-1 cells. **E** RT-qPCR analysis for the
mRNA expression level of YTHDF3 and MYC in PANC-1 and AsPC-1 cells infected
with YTHDF3 shRNAs. **F** RT-qPCR analysis for the mRNA expression level of YTHDF3
and MYC in PANC-1 and AsPC-1 cells infected with YTHDF3 plasmids, with DMSO/2-
BP treatment. **G** Western blot analysis for the expression of YTHDF3 and MYC in
PANC-1 and AsPC-1 cells infected with YTHDF3 shRNAs. **H** Western blot analysis for
the expression of YTHDF3 and MYC in PANC-1 and AsPC-1 cells infected with
YTHDF3 plasmids, with DMSO/2-BP treatment. The half-life of MYC mRNA in PANC-
1 (**I**) and AsPC-1 (**J**) cells infected with YTHDF3 shRNAs, the half-life of mRNA was
estimated as equation in Method, nonlinear regression analysis. Genome browser
tracks for input and RIP-seq of YTHDF3 (**K**) and meRIP-seq (**L**) data from GSE130173;
Input is indicated in blue and RIP/MeRIP in red, MYC-CRD domain is indicated in
yellow. **M** MeRIP-qPCR assay of MYC-CRD by using the IgG or m6A antibodies in
PANC-1 and AsPC-1 cells. **N** RNA pulldown of endogenous YTHDF3 in PANC-1 and
AsPC-1 cells using synthetic MYC-CRD RNA fragments, with (m6A) or without (**A**)
m6A modifications. **O** Relative firefly luciferase (Fluc) activity of wild-type (CRD-wt)
or mutated (CRD-mut) CRD reporters in PANC-1 cells with ectopically expressed
YTHDF3, Vector as relative control. **P** RIP-qPCR assay of Luc-CRD by using the IgG
or YTHDF3 antibodies in PANC-1 and AsPC-1 cells infected with CRD-wt/CRD-mut
plasmids, IgG as negative control. **Q** Relative Fluc activity of wild-type CRD repor-
ters in PANC-1 and AsPC-1 cells with 2-BP treatment. Results (**D, E, F, I, J, M, O, P, Q**)
are presented as mean ± SD, $n = 3$ biologically independent experiments. $P$ values in
results (**D, M, O, P, Q**) were determined by two-tailed unpaired $t$ test, (**E, F**) by one-
way ANOVA. Similar results for (**G, H, N**) panels were obtained in three independent
experiments. Source data are provided as a Source Data file.

step (-HAM). After washing, the beads were treated with thiol-reactive
biotin molecules for 1 h at room temperature. The beads were boiled
for 10 min after resuspending in SDS-PAGE loading buffer and then
subjected to western blotting analysis with Streptavidin-HRP antibody.

## RNA immunoprecipitation (RIP)
Cells seeded in a 15 cm dish at 70–80% confluency were cross-linked
by ultraviolet light at 254 nm (200 J/cm²), then harvested and lysa-
ted. RNA immunoprecipitation (RIP) assay was performed according
to the instructions of the Magna RIP RNA Binding Protein
Immunoprecipitation Kit (Millipore, USA). The input and co-
immunoprecipitated RNAs were extracted and then quantified by
qRT-PCR. Specific antibodies used for Co-IP are presented in Sup-
plementary Data 1.

## Glutathione S-transferase (GST) pull-down
Escherichia coli BL21 was lysed using muramidase and sonication to
extract GST-fusion proteins (Vector: pET-GST). Glutathione-Sepharose
beads (GE Healthcare Life Sciences, USA) were used to isolate GST-
fusion proteins at 4 °C overnight. The beads were collected and
washed six times. Cells were lysed in Western/IP lysis buffer for 30 min
on ice and then was incubated with the beads at 4 °C overnight. The
beads were washed six times and boiled for 10 min. The bound pro-
teins were subjected to western blotting analysis to analyze the
protein-protein interactions.

## RNA stability assay
Pancreatic cancer cells were seeded into 6-well plates to get 50%
confluency after 24 h. Cells were treated with 5 μg/ml actinomycin D
and collected at indicated time points for RT-qPCR. The half-life of
MYC mRNA was estimated according to a previously published
paper[33]. Since actinomycin D treatment results in transcription stalling,
the change of mRNA concentration at a given time (dC/dt) is propor-
tional to the constant of mRNA decay (K) and mRNA concentration (C),
leading to the following equation:

$$dC/dt = -KC \tag{1}$$

Thus the mRNA degradation rate K was estimated by:

$$\ln(C/C_0) = -Kt \tag{2}$$

To calculate the mRNA half-life (t1/2), when 50% of mRNA is
decayed (ie. $C/C0 = 1/2$), the equation was:

$$\ln(1/2) = -Kt_{1/2} \tag{3}$$

From where:

$$t_{1/2} = (\ln 2)/K \tag{4}$$

## Dual-luciferase reporter assay
The DNA fragments of wild-type and mutant CRD were synthesized
and cloned into the XhoI site of pMIR-REPORT vector (Ambion, Austin,
TX) to constructed CRD firefly luciferase reporters. The CRD firefly
luciferase reporter plasmids (pMIR-CRD-WT and pMIR-CRD-mut,
respectively) and renilla luciferase reporter control plasmids (pRL-TK)
were cotransfected with or without YTHDF3 expression vectors using
Lipofactamine 2000 for 48 h. Luciferase activity equals to the ratio
Fireflfy luciferase/Renilla luciferase.

## Ethics approval
This study was approved by the institutional research ethics commit-
tee of Tongji Medical College, Huazhong University of Science and
Technology, and written informed consent was obtained from all
patients prior to the investigation. All animal experiments were per-
formed in strict accordance with the recommendations in the guide for
the care and Use of laboratory animals of Tongji Medical College. The
licence was issued by the Animal Use and Care Committees at Tongji
Medical College, Huazhong University of Science and Technology
(IACUC Number 2728).

## Mice xenograft models
All animal experiment procedures were approved by the Ethics Com-
mittee of Tongji Medical College, Huazhong University of Science and
Technology (IACUC NO.2728). Mice were monitored daily except
weekends for signs of disease progression. Moribund animals such as
the subcutaneous tumor reaching a size larger than 2 cm in diameter,
were sacrificed as mandated by the IACUC protocol. The BALB/c-nu
mice (4 weeks old, male) and M-NSG (NOD-Prkdcscid IL2rgem1/Smoc,
male, 4 weeks old) mice were purchased from Vitalriver (Beijing,
China) and housed under pathogen-free conditions for one week
before the experiments. Pancreatic cancer cells ($5 \times 10^6$) infected with
different lentivirus were dispersed in 100 μL PBS and inoculated sub-
cutaneously into the left dorsal flank of mice. The subcutaneous tumor
volume was measured using a caliper every three days and estimated
as follows: tumor volume (mm³)= (width)² × length × 1/2. All sub-
cutaneous xenografts were excised to weight after all mice were
euthanized. Tumor tissues were taken subcutaneously for subsequent
immune phenotyping by IHC. Corresponding antibodies are listed in
Supplementary Table S2. Tumor burdens were monitored and quan-
tified by Bioluminescence imaging (BLI) on Lago X Imaging System
(Spectral Instruments Imaging, USA). BLI signals were shown as the
average radiance and acquired by Living Image software (Amiview).

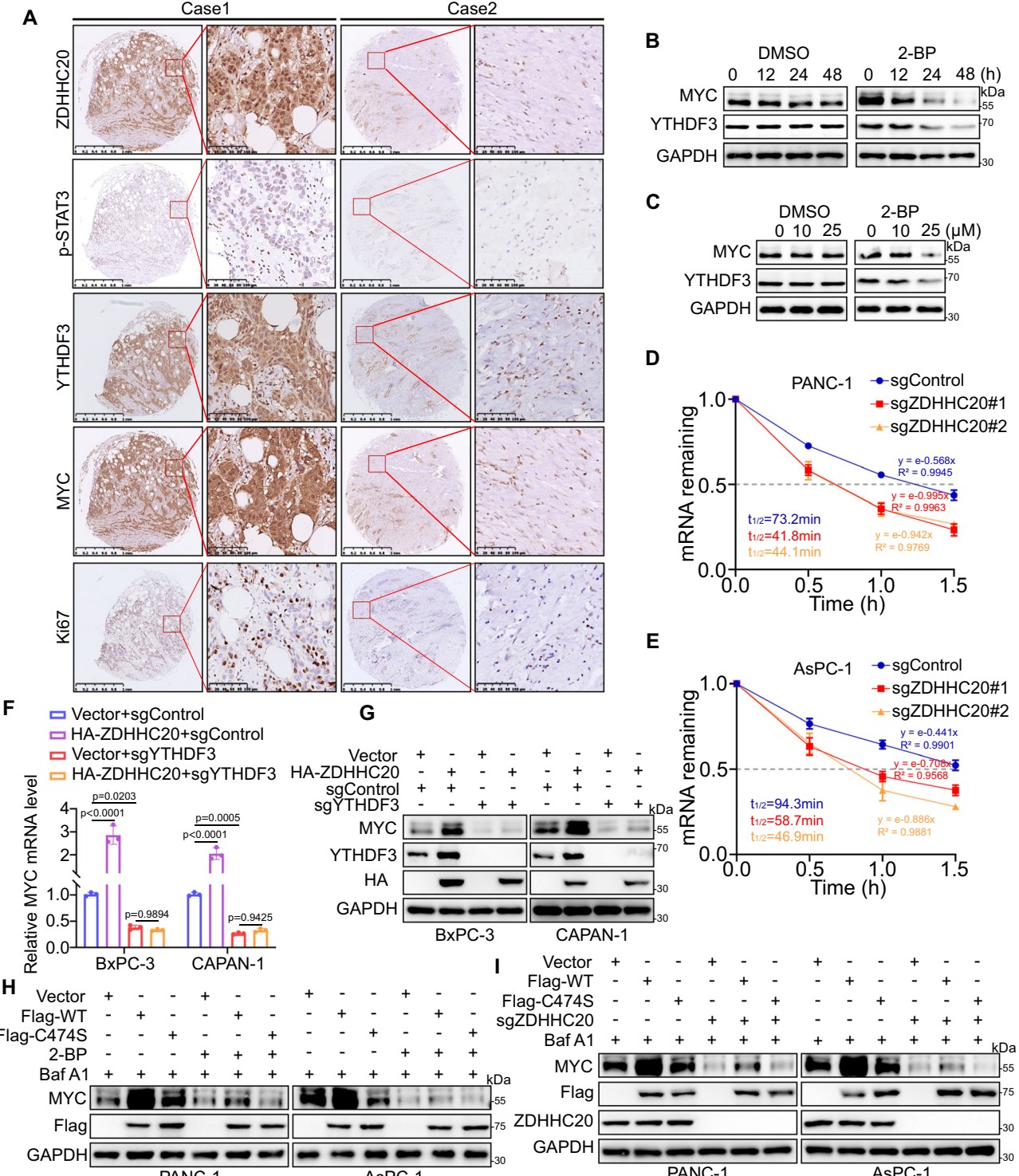

**Fig. 8 | ZDHHC20-mediated palmitoylation of YTHDF3-Cys474 stabilizes MYC mRNA. A** Representative images of IHC analysis for ZDHHC20, p-STAT3, YTHDF3, MYC and Ki67 on TMA containing a cohort of pancreatic cancer samples (*n* = 29). **B** Western blot analysis for the expression of YTHDF3 and MYC in PANC-1 cells with DMSO/2-BP treatment for 0, 12, 24, 48 h. **C** Western blot analysis for the expression of YTHDF3 and MYC in PANC-1 cells with DMSO/2-BP treatment for 0, 10, 25 µM. **D**, **E** The half-life of MYC mRNA in PANC-1 and AsPC-1 cells infected with ZDHHC20 knockout was evaluated by actinomycin D treatment for different time. Each bar represents the mean ± SD of three independent experiments. **F** RT-qPCR analysis for the mRNA expression level of MYC in BxPC-3 and CAPAN-1 cells infected with HA-ZDHHC20 plasmids, with or without YTHDF3 knockout, *n* = 3

biologically independent experiments, one-way ANOVA, statistical data are presented as mean ± SD. **G** Western blot analysis for the expression of HA, YTHDF3 and MYC in BxPC-3 and CAPAN-1 cells infected with HA-ZDHHC20 plasmids, with or without YTHDF3 knockout. **H** Under Barf A1 treatment, western blot analysis for the expression of Flag-YTHDF3 and MYC in PANC-1 and AsPC-1 cells infected with Vector, Flag-YTHDF3 WT/C474S plasmids, with or without 2-BP treatment. **I** Under Barf A1 treatment, western blot analysis for the expression of Flag-YTHDF3 and MYC in PANC-1 and AsPC-1 cells infected with Vector, Flag-YTHDF3 WT/C474S plasmids, with or without ZDHHC20 knockout. Data are presented as mean ± SD. Similar results for (**B**, **C**, **G**–**I**) panels were obtained in three independent experiments. Source data are provided as a Source Data file.

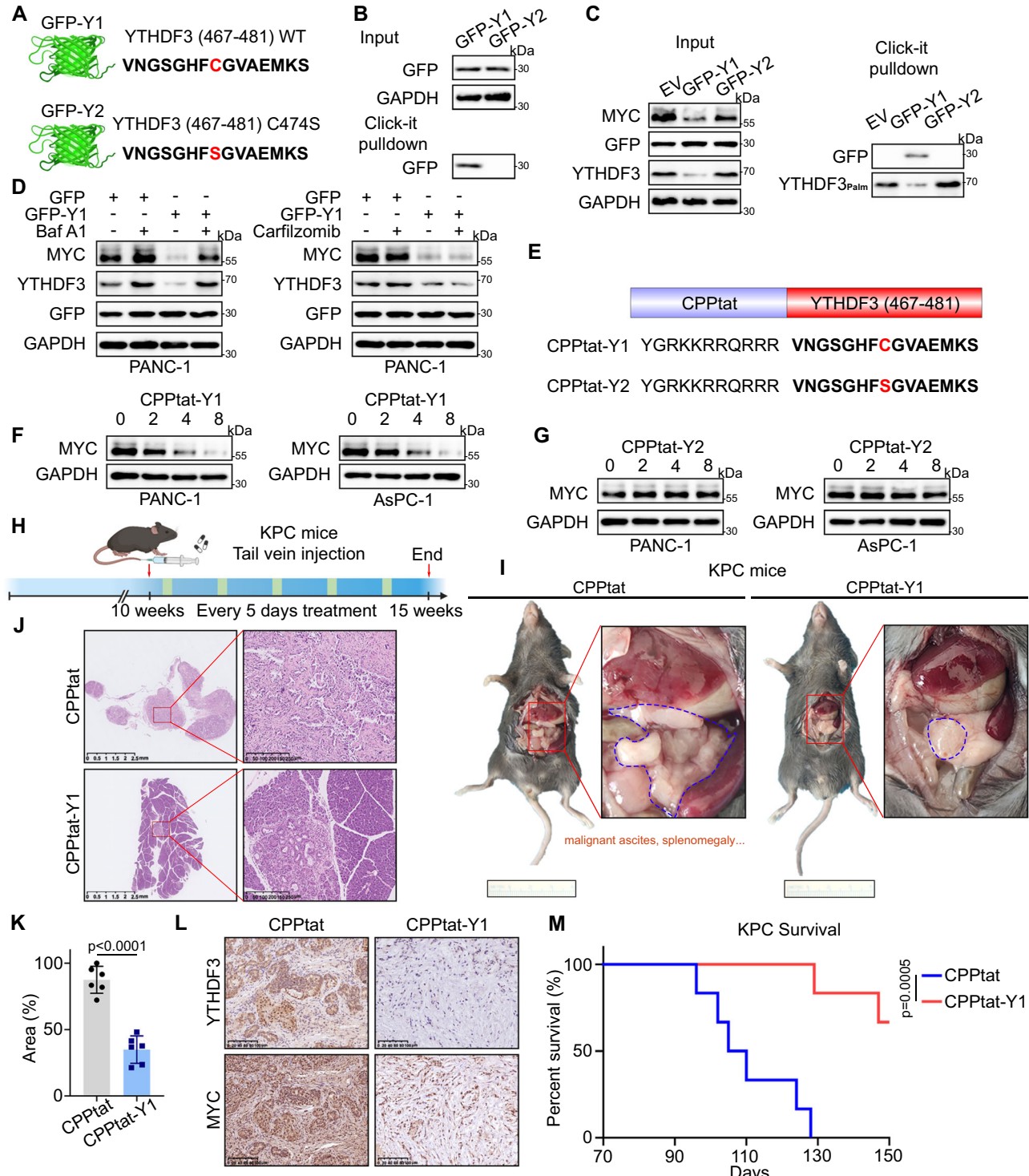

**Fig. 9 | Therapeutic blockade of the ZDHHC20-YTHDF3 interaction inhibits pancreatic cancer progression. A** Schematic of the fusion motif GFP-Y1, including GFP and YTHDF3 (467–481), or C474S mutant (GFP-Y2). **B** PANC-1 cells infected with GFP-Y1 and GFP-Y2 were harvested for Click-IT reaction and streptavidin pulldown to detect the palmitoylation of exogenous GFP. **C** PANC-1 cells infected with Vector, GFP-Y1 and GFP-Y2 were harvested for Click-IT reaction and streptavidin pulldown to detect the palmitoylation of endogenous YTHDF3. **D** Western blot analysis for the expression of GFP, YTHDF3 and MYC in PANC-1 cells infected with GFP-Vector or GFP-Y1 plasmids, with or without Baf A1 and Carfilzomib treatment. **E** Schematic of CPPtat-Y1 and CPPtat-Y2 peptides. The different residues are shown in red. **F** Western blot analysis for the expression of MYC in PANC-1 and AsPC-1 cells treated with CPPtat-Y1 for 0, 2, 4, 8 μM. **G** Western blot analysis for the

expression of MYC in PANC-1 and AsPC-1 cells treated with CPPtat-Y2 for 0, 2, 4, 8 μM. **H** Schematic diagram for timeline of CPPtat and CPPtat-Y1 treatment in the KPC mouse model (Created with BioRender.com released under a CC-BY-NC-ND 4.0 International license). **I**–**L** Representative photos of KPC mice treated with CPPtat and CPPtat-Y1 ; representative images of H&E staining ; percentage of pancreatic neoplastic area, $n = 6$ per group, two-tailed unpaired $t$ test; representative images of IHC staining (L). (M) Survival analysis of KPC mice (10 weeks old) treated with CPPtat and CPPtat-Y1 every one week for 5 weeks (100 mg/kg; $n = 6$ mice per group), Log-rank test. Data are presented as mean ± SD. Similar results for (**B**–**D**, **F**, **G**) panels were obtained in three independent experiments. Source data are provided as a Source Data file.

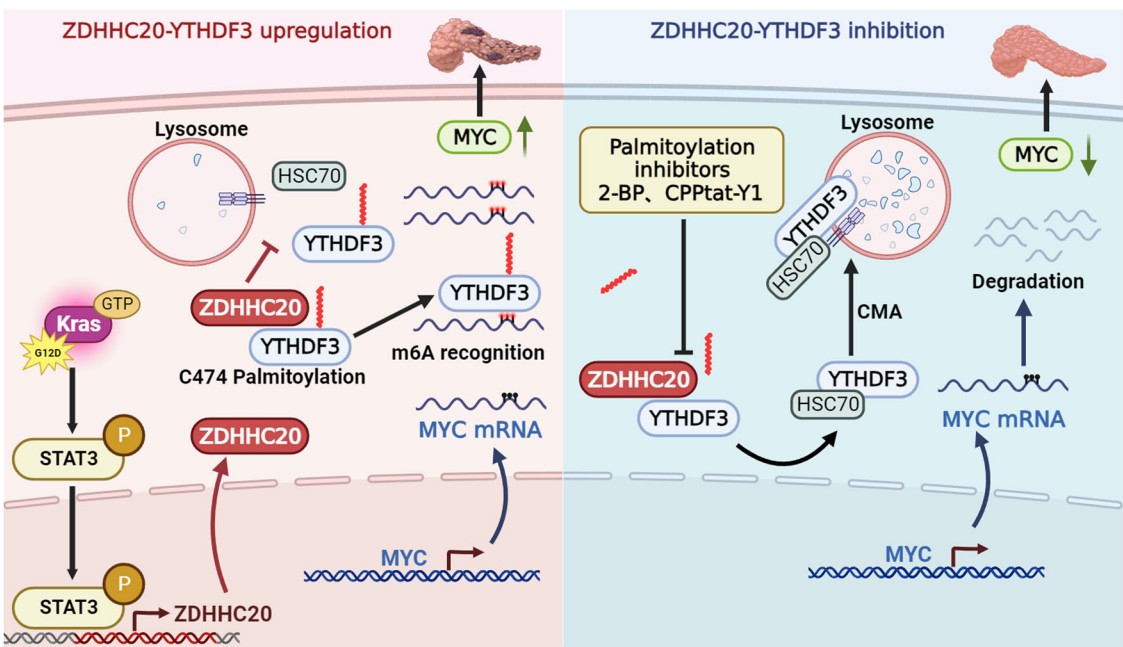

**Fig. 10 | ZDHHC20-mediated S-palmitoylation of YTHDF3 stabilizes MYC mRNA to promote pancreatic cancer progression.** *KRAS* mutations induce the accumulation of ZDHHC20 in pancreatic cancer through activation of STAT3. ZDHHC20 inhibits lysosomal localization and degradation of YTHDF3 through S-palmitoylation of Cys474, which can result in abnormal accumulation of the oncogenic product MYC and thereby support the malignant phenotypes of cancer cells. Targeting palmitoylation using a pharmacological inhibitor (2-BP) or competitive inhibitory peptide (CPPtat-Y1) inhibit YTHDF3 palmitoylation mediated by ZDHHC20, which in turn downregulate MYC expression and inhibited the progression of *KRAS* mutant pancreatic cancer (Created with BioRender.com released under a CC-BY-NC-ND 4.0 International license).

## Spontaneous pancreatic cancer model

The KPC (*LSL-KrasG12D/+; LSL-TrpS3R172H/+; Pdx-1-Cre*, 8 weeks old, sex-matched) transgenic mice were purchased from MODEL ORGANISMS Inc. (Shanghai, China) and housed under pathogen-free conditions. AAV8-shZDHHC20 virus was purchased from Shandong WZ Biotech (Jinan, Shandong, China). The virus was intraperitoneally injected 100 µL per mouse at a concentration of $6 \times 10^9$ vector genomes per microliter. CPPtat/ CPPtat-Y1 (100 mg/kg) or DMSO/2-BP (40 mg/kg) were injected via the tail vein every five days for a total of five times from 10 weeks old. Mice were euthanized, and the pancreas was collected after 5 weeks.

## Human tissue samples

A collection of 10 frozen PDAC and corresponding non-tumorous surrounding pancreas, from first line surgically treated patients, were used to study at the mRNA and protein level the expression of ZDHHC20 in tissues. PDAC specimens were collected at the Wuhan Union Hospital. Institutional Review Board approval was obtained at the local Ethical Committee (No. [2020] IEC-J (030)). Informed consent was obtained from the participants both for the use of their samples and to publish information that identified individuals.

## Data mining and bioinformatics analysis

GEPIA (http://gepia2.cancer-pku.cn) and TCGA-PAAD were used for the expression profile of palmitoyl acyltransferases in pancreatic cancer. (GEPIA Expression Analysis; Box Plots; Gene A: ZDHHCs; Datasets: PAAD).

TIMER2.0 (http://timer.comp-genomics.org/timer/) Gene_Corr module was to explore the correlation between STAT3, ZDHHC20, YTHDF3 and MYC in various cancer types. TIMER2.0 Gene_Mutation module compares the differential ZDHHC20 expression between different mutation status of *KRAS* (Mutated Gene: *KRAS*; Gene Expression: ZDHHC20).

KNOCKTF2.0 (https://bio.liclab.net/KnockTFv2/) database, provided comprehensive transcription (co-)factors knockdown/knockout dataset resource across multiple tissue/cell types of different species, was used to analyze the potential transcription (co-)factors that most significantly regulate ZDHHC20 transcription (Search by Target Gene; Species: Homo sapiens; Gene Name Type: Gene Symbol; Gene Name: ZDHHC20; Fold Change: 2).

Gene Ontology (GO) enrichment analysis result of the RNA-seq was drawn by an online platform OmicShare (https://www.omicshare.com/), under the default instructions.

## Analysis of publicly available RIP-Seq and MeRIP-Seq

To study how m6A reader YTHDF3 regulates MYC mRNA stability, the published RIP-Seq and MeRIP-Seq involved in this study were downloaded from GSE130173. Reads were analysed and aligned to the reference genome by Hisat2 software (v2.0.4). IGV v2.9.0 was used for analysis and visualization of RIP-Seq and MeRIP-Seq data.

## LC-MS/MS

The bead samples obtained from the immunoprecipitation experiment were washed three times with precooled PBS to remove the remaining detergent. Then, bead samples were incubated in reaction buffer (1% SDC; 100 mM Tris-HCl, pH 8.5; 10 mM TCEP; 40 mM CAA) at 60 °C for 1 h for protein denaturation, cysteine reduction and alkylation. The eluates were diluted with an equal volume of H2O and subjected to trypsin digestion overnight at 37 °C by adding 1 µg of trypsin. The peptide was purified using homemade SDB desalting columns. The eluate was vacuum dried and stored at −20 °C for later use. LC-MS/MS data acquisition was carried out on a Q Exactive HF mass spectrometer coupled with UltiMate 3000 RSLCnano system. For DDA mode analysis, each scan cycle is consisted of one full-scan mass spectrum (R = 60 K, AGC = 3e6, max IT = 25 ms, scan range = 350–1500 m/z) followed by 20 MS/MS events (R = 15 K, AGC = 1e5, max IT = 50 ms). HCD collision energy

was set to 27. Isolation window for precursor selection was set to 1.4 Da. Former target ion exclusion was set for 24 s. MS raw data were analyzed with MaxQuant (V2.0.1) using the Andromeda database search algorithm. Spectra files were searched against the UniProt Human proteome database and target protein file using the following parameters: type standard. Variable modifications, Oxidation (M) & Acetyl (Protein N-term); Fixed modifications, Carbamidomethyl (C); Digestion, Trypsin/P. The MS1 match tolerance was set as 20 ppm for the first search and 4.5 ppm for the main search; the MS2 tolerance was set as 20 ppm. The mode of match between runs was on. Search results were filtered with 1% FDR at both protein and peptide levels. Information for ZDHHC20-binding proteins analyzed by IP/MS assays is provided as Supplementary Data 2. Information for Flag-YTHDF3 WT/C474S immunoprecipitates proteins analyzed by IP/MS assays is provided as Supplementary Data 3. The protein identification via mass spectrometry (MS) was performed by SpecAlly Life Technology Co., Ltd, Wuhan, China.

## Plasmid construction
Lentiviral vectors plasmids were constructed by GENECHEM Biotech at Shanghai, China (http://genechem.bioon.com.cn/). Briefly, the expression vectors encoding ZDHHC20, YTHDF3, KRAS G12D, STAT3 and GFP-Y were generated by inserting synthesized complementary DNAs into the GV493 or GV492 vector. The ZDHHC20 C156S/F171A and YTHDF3 C474S mutants were generated by site-directed mutagenesis PCR reaction using platinum QuikChange site-directed mutagenesis kit (Stratagene) according to the manufacturer's instructions. All plasmids were sequenced to confirm whether the designed mutation was present, without any other unwanted mutation.

## Methylated RNA immunoprecipitation (MeRIP) assay
The MeRIP assay was performed using the EpiQuik ™ CUT&RUN m6A RNA Enrichment (MeRIP) Kit according to the manufacturer's instructions. Briefly, immunocapture solution was prepared by adding the reagents to 0.2 ml PCR tubes and rotating at room temperature for 90 min. 10 μl of NDE (Nuclear Digestion Enhancer) and 2 μl of CEM (Cleavage Enzyme Mix) were added to each tube and incubated at room temperature for 4 min. Tubes were placed on the magnetic device until the solution turned clear. The supernatant was then collected and discarded. Samples were then washed three times with 150 μl of Wash Buffer, and once with 150 μl of protein digestion buffer. Then, 20 μl of Protein Digestion Solution was added to samples before mixing and incubating at 55 °C for 15 min in a thermocycler. RNA Purification Solution and 100% ethanol were then applied to the samples and vortexed to resuspend and wash the RNA Binding Beads. Resuspended beads were then treated with 13 μl of Elution Buffer and incubated at room temperature for 5 min to release the RNA from the beads. 13 μl from each sample was transferred to a new 0.2 ml PCR tube for immediate use or store at −20 °C.

## ChIP-qPCR assay
ChIP-qPCR assay in pancreatic cancer cells was performed with Pierce ChIP Kit, Agarose (Cat. #26156, Thermo Fisher Scientific, Waltham, MA, USA) according to the manufacturer's instructions. Briefly, crosslinking was performed by 1% formalin, and Nuclease (ChIP Grade) was used to fragment the DNA. The following antibodies were utilized to immunoprecipitate crosslinked protein-DNA complexes: rabbit anti-STAT3 (10253-2-AP, Protein tech) and Rabbit IgG (A7016, Beyotime). Eluted DNA fragments were analyzed by qPCR. ChIP-seq analysis revealed the presence of a STAT3 binding peak in the transcription initiation region of ZDHHC20 and the primers are listed in Supplementary Table S2.

## Transwell assay
Pancreatic cancer cells were cultured into the 24-well Transwell plates with 8.0 μm pores (Corning Costar, USA) with precoated Matrigel (BD, USA; diluted 1: 8). Briefly, $5 \times 10^4$ pancreatic cancer cells were seeded into the top chamber. After cultivation for 24 h, the membrane was collected and stained with the Crystal Violet Solution (Solarbio, China). A cotton swab was used to remove those cells that did not migrate or invade through the pores. The migrating cells were counted and photographed by a microscope from five different microscopic fields per well.

## Statistical analysis
All data were statistically analyzed using GraphPad Prism software (version 8.0). Statistical significance was determined by unpaired Student's t test (when comparing two experimental groups), one-way and two-way ANOVA tests as appropriate (when comparing more than two experimental groups). The data are represented as mean ± standard deviation (SD). All statistical details of the experiments were mentioned in the figure legends. All tests were two-sided; the P value less than 0.05 was considered statistically significant (*$P < 0.05$; **$P < 0.01$; ***$P < 0.001$; ****$P < 0.0001$; ns: not significant).

## Reporting summary
Further information on research design is available in the Nature Portfolio Reporting Summary linked to this article.

## Data availability
The RNA-seq data generated in this study have been deposited in the NCBI Gene Expression Omnibus (GEO) under accession code GSE235516. All mass spectrometry raw data generated in this study have been deposited to the ProteomeXchange via the PRIDE partner repository with the dataset identifiers PXD046541 and PXD043292. The previous published data of GEO, GSE16515 and GSE130173 were used in this paper. All data needed to evaluate the conclusions in the paper are present in the paper and/or the Supplementary Materials. Source data are provided with this paper.

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

## Acknowledgements

We would like to thank all researchers for their contributions. This study was supported by grants from the National Natural Science Foundation of China (No. 82073321 (X.J.), No.82272910 (X.J.), No. 82100190 (L.T.), No. 82303826 (H.Z.), China Postdoctoral Science Foundation (2023M731220, H.Z.), Excellent Youth Foundation of Hunan Scientific Committee (Grant No. 2022JJ10092, X. J.), and Central South University Innovation-Driven Research Programme (Grant No.2023CXQD059, X.J.). Thanks for the technical support by the Huazhong University of Science & Technology Analytical & Testing center, Medical sub-center.

## Author contributions

Study concept and experimental design: H.Z., X.J.; Conducting experiments: H.Z., L.T., X.H.; Data analysis and interpretation: H.Z., L.T., Y.S., X.H., Z.W., K.J., X.J.; Supervision: H.Z., L.T., Y.S., X.H., Z.W., K.J., X.J.; Writing—original draft: H.Z., L.T., X.J.; Writing—review & editing: H.Z., X.J.; All authors contributed to reading and editing the manuscript.

## Competing interests

The authors declare no competing interests.
