## [Peer Review File · Nature Communications]

ZDHHC20-Mediated S-Palmitoylation of YTHDF3 Stabilizes MYC mRNA to Promote Pancreatic Cancer ProgressionREVIEWER COMMENTS

Reviewer #1 (Remarks to the Author):

In “ZDHHC20-Mediated S-Palmitoylation of YTHDF3 Stabilizes MYC mRNA to Promote Pancreatic Cancer Progression”, Zhang and colleagues explored the effects of palmitoylation of YTHDF3 on the stability of c-Myc RNA, and the effects of this process on the progression of pancreatic cancer. They also identified a potential pharmacological strategy to treat pancreatic cancer, by using a peptide that mimics YTHDF3 and “sequester” its palmitoylation.

The manuscript is overall well written and clear. The findings are exciting and represent important advances of significance to specialists within various fields such as pancreatic cancer, cancer pharmacology, RNA, RNA-binding proteins, and epitranscriptomic fields. I personally found very exciting that the authors describes the first post-transcriptional modification of an m6A-reader: with their work they pave the way to a new point of view on the analysis of the canonical and non-canonical m6A readers. Moreover, the pharmacological approach is very promising and absolutely worth the attention of the scientific community. I do recommend the publication of this work, after addressing a series of major and minor issues.

First, the Methods section lacks the description of a series of techniques used in the work:

1. All the bioinformatics analyses (GEPIA, TCGA, RIP-seq, TIMER2.0 and more) are not detailed in the methods. These must be detailed in the Methods and, to help the readers who are not familiar with these software/servers, explain concisely their scope.
2. The ChIP-qPCR method must be described.
3. The mass spectrometry sample preparation, MS/MS, and analysis description is wholly absent, and it must be added.
4. The cloning/mutagenesis of various plasmids is mentioned throughout the manuscript, and no methodological description is provided in the paper. This must be added.
5. The Me-RIP experiments are not described in the Methods, and this must be added.

Following, my comments on the results section.

1. Aberrant ZDHHC20 upregulation predicts unfavorable prognosis in pancreatic cancer. In Figure 1B, the tumor area is highlighted within blue dotted lines. Can please the authors explain how the tumor was assessed (visually, and on what basis)? Were metastasis observed? Were these considered?

Figure S1A reports the expression of all known palmitoyl acyltransferases in pancreatic cancer based on GEPIA. Statistics is provided, but no error bars, which must be added. Figure S2/1E reports more on this matter; there is not a legend, so it is impossible for the reader to appreciate the meaning of this figure. The figures also report numerosity of T and N, which I assume mean tumor and normal, but a legend will definitely help to understand. Similarly, Figure S3 does not report a full legend. I assumed that orange is “high expression” and green “low expression” of ZDHHC, respectively. This should be clearly explained. More importantly, this figure shows a great variability in the survival rate of the patients, based on the expression of the different ZHDDs. For example, ZDHHC20 high expression in pancreatic cancer tissue seems to correlate with low survival, but the trend is the opposite with ZDHHC5, C6, C7, and various more. Can the authors discuss this? Do the different palmitoylated proteins share palmitoylation targets? Have the authors checked if YTHDF3 is target of any other ZDHHC protein? Do the authors think that the negative prognosis always depends on the YTHDF3 palmitoylation, no matter what ZDHHC is overexpressed.

Figure 1K is missing the statistics (asterisks) on the figure.

In line 119, the authors state that KRAS is low in BxPC-3 cell line; a reference is needed to support that.

2. KRAS mutations induce the accumulation of ZDHHC20 in pancreatic cancer via STAT3.

Figure 2C reports on the expression of the ZDHHC20 protein in different PDAC lines. It is not clear how these data were obtained (western blots? IHC?) and the statistics are missing.

Figure 2G shows the expression of ZDHHC20 PDAC lines treated with a KRAS inhibitor.

Particularly for AsPC-1, the change in the signal is not very much. Can the author add a quantification of the biological/technical repeats and show the statistics to corroborate the decrease in ZDHHC20 stated in this section of the manuscript? Figure 2H reports on the RT-qPCR analysis for the mRNA of ZDHHC20 in PDAC lines infected with KRASG12D plasmids. Could the authors state how the ZDHHC20 mRNA levels in these cell lines and vector-transfected cell lines relate, to show better the effect of the overexpression (as it stands, it is not clear if the reader can assume that the endogenous mRNA levels are the same as in the vector-transfected cells)?

Figure S4A reports the prediction of the transcription factors that target ZDHHC20 based on the KnockTF platform. Not being familiar with this platform, I found this panel quite complex to understand. It is unclear whether the fold change entails protein or mRNA and whether this is normalized to the knockdown, silencing, or inhibition of ZDHHC20. Moreover, why is this defined as a prediction? Is it not based on collected data? In the prediction, seven proteins show a great fold change: what was the rationale to exclude 6 out of 7? Were the other proteins tested too?

The western blots in Figures S4C and 2K are missing the quantification and relative statistics. Particularly, with BP-1-102 treatment in AsPC-1, the change seems quite minor (and the GAPDH signal decreases too), so the quantification will be beneficial.

Figure S4D/E reports the correlation between STAT3 and ZDHHC20 in various cancer types calculated with TIMER2.0. The authors must provide a detailed explanation on how these data were generated. Additionally, can the authors comment on whether the observed correlation is specific to pancreatic cancer?

Figures 2L/2M report on the ChIP-seq data. It is difficult for me to comment on this panel as the method section is missing. I am assuming that the immunoprecipitation was performed with an antibody against STAT3, but this must be detailed in the Methods.

Figure 2N must report a proper quantification.

3. ZDHHC20 promotes pancreatic cancer progression in a palmitoylation-dependent manner.

Figures 3C-3E. The transwell assay is not detailed in the methods. Although a commercial kit-based, this must be described.

4. YTHDF3 mediates the oncogenic capacity of ZDHHC20 in pancreatic cancer

Figure 4A shows the silver staining of the gel electrophoresis after Flag-ZDHHC20-immunoprecipitation. The bands of YTHDF3 and FLAG-ZDHHC20 are indicated based exclusively on MW, but this is just a hypothetical position, as no western blot is provided.

The caption should be rephrased stating that these are just predicted bands.

Figure 4C reports the MS of a peptide derived from Flag-ZDHHC20-immunoprecipitates belonging to YTHDF3. I am very concerned about this panel as the sequence reported herein is shared with the paralogs YTHDF1 and YTHDF2 (See below – sequences highlighted in lower case).

>sp|Q9BYJ9|YTHD1_HUMAN YTH domain-containing family protein 1 OS=Homo sapiens OX=9606 GN=YTHDF1 PE=1 SV=1

MSATSVDTRTKGQDNKVGQNGSLHQKDTVHDNDFEPYLTGQSNQSNYSYPSMSDPYLYSSY
YPPSIGFPYSLNEAPWSTAGDPPIPYLTTYGQLSNGDHHFMHDAVFGQPGGLGNNIYQHRF
NFFPENPAFSAWGTSGSQGQQTQSSAYGSSYTYPPSSLGGTVVDGQPGFHSDTLSKAPG

MNSLEQGMVGLKIGDVSSSAVKTVGSVSSVALTGVLSGNGGTNVNMPVSKPTSWAAIAS
KPAKPQPKMKTSGPVMGGGLPPPIKHNM DIGTWDNKGPVPKAPVPPQAPSPQAAPQP
QQVAQPLPAQP PALAQPQYQSPQQPPQTRWVAPRNRNAAFGQSGGAGSDSNSPGNVQP
NSAPSVESHVLEKLKAAHSYNPKFEFENLKSGRVFIKSYSEDDIHRSIKYSIWCSTEHGN
KRLDSAFRCMSSKGPVYLLFSVNGSGHFVCGVAEMKSPVDYGTSA G VWSQDKWKGKFDV
QWIFVKDVPNNQLRHIRLENNDNKPVTNSRdtqevplekAKQVLKI ISSYKHTTSIFDDFAHYEK
RQEEEEVVRKERQSRNKQ

>sp|Q9Y5A9|YTHD2_HUMAN YTH domain-containing family protein 2 OS=Homo sapiens
OX=9606 GN=YTHDF2 PE=1 SV=2

MSASSLLEQRPKGQGNKVQNGSVHQKDG LND DDFEPYLS PQARPNNAYTAMSDSYLPSY
YPSIGFSYSLGEAAWSTGGDTAMPYLTSYGQLSNGEPHFLPDAMFGQP GALGSTPFLGQ
HGFNFFPSGIDFSAWGNSSQGGQSTQSSGYSSNYAYAPSSLGGAMIDGQSAFANETLNKA
PGMNTIDQGMAALKLGSTEVASNVPKVVGSAVGS GSITSNIVASNSLPPATIAPPKPASWAD
IASKPAKQQPKLKTNGIAGSSLPPIKHNM DIGTWDNKGPVAKAPSQALVQNIGQPTQG
SPQPVGQQANN SPPVAQASVGQQTQLPPPPPPQAQLSVQQQAAQPTRWVAPRNRGSG
FGHNGVDGNGVGGSSQAGSGSTPSEHPVLEKLRSINNYNPKDFDWNLKHGRVFIKSYSE
DDIHRSIKYN IWCSTEHGNKRLDAA YRSMNGKGPVYLLFSVNGSGHFVCGVAEMKSAVDYN
TCAGVWSQDKWKGRFDVRWIFVKDVPNSQLRHIRLENNENKPVTNSRdtqevplekAKQVLKI
IASYKHTTSIFDDFSHYEK RQEEEE SVKKERQGRGK

How can the authors unequivocally state that the interaction is with YTHDF3 (from the MS data)? Also, the description of these data is way too concise, it must be rephrased so that a better understanding is ensured.

5. ZDHHC20 regulates the palmitoylation of YTHDF3 in pancreatic cancer.

Figure 5A shows the results obtained with a commercial kit which is not detailed in the Methods. The description must be added.

At page 9, lines 235-236, the authors state that “this uniquely predicted site is also located in the binding region of ZDHHC20 and YTHDF3.” Based on what evidence they believe that the peptide reported is part of an interaction surface? This is not supported by any of the data reported in the manuscript.

6. ZDHHC20 suppresses the degradation of YTHDF3 via palmitoylation on Cys474. Figure 6B reports a western blot: signal quantification is missing, and it must be added.

Figure 6H reports representative images of cancer cells infected with Flag-YTHDF3 WT/C474S plasmids in IF. Are these CHX-treated cells? If not, why is the signal of YTHDF3 in lysosome similar to the one of sgZDHHC20? Does it mean that YTHDF3 does get palmitoylated by other enzymes? This should be detailed.

7. YTHDF3-mediated m6A modification regulates MYC mRNA stability.

I believe that Figure 7F is missing the labelling of YTHDF3 and c-Myc (similar to 7E).

Generally speaking, this is the weakest link in the manuscript. Although the authors provide a good characterisation of the binding of YTHDF3 to m6A-RNA (this was already well known in the literature anyway), the overall connection between YTHDF3 palmitoylation and m6A binding is not fully explored and discussed. Based on 7F, the inhibition of the palmitoylation of YTHDF3 leads to less binding of c-MYC RNA, as recapitulated by the C474S mutant in RIP-seq and MeRIP-Seq assays, suggesting a connection between ZHDCC-dependent palmitoylation and c-MYC RNA binding, not necessarily with m6A-RNA. In the luciferase assay, the DRACH motifs are completely altered (A-to-T) and it is known from structural data that the YTH requires an A in that position to bind RNA, independently from the presence of the methylation (Nucleic Acids Res. 2014 Dec 16;42(22):13911-9, and similar studies). To robustly state the connection with m6A-RNA, the authors must repeat the luciferase assay with a fully methylated RNA, with and without 2-BP treatment.

In addition, the authors must consider the fact that C474 is in close vicinity of one the residues of the tryptophan-cage of YTHDF3, the binding surface responsible for the

coordination of the N6-methyl moiety of m6A (see J. Chem. Inf. Model. 2020, 60, 12). Hence, two (maybe more?) scenarios are possible: i) the palmitoylation is inducing a conformational change that increases the RNA overall binding properties; ii) the mutation of C474 into serine might be altering the conformation of the tryptophan-cage and partially disrupting the ability to bind m6A in a manner that is completely independent from the palmitoylation. All considered, the claim that “m6A modifications in the MYC CRD are required for YTHDF3-mediated stabilization of MYC mRNA in pancreatic cancer” requires more structural data that are not herein provided. As I totally understand that a biophysical characterisation of this process is not in the scope of the paper, I think the authors should just include a more detailed description of the possible scenarios and all these limitations of their characterisation in their discussion.

Following, a list of minor typos that require attention.

Page 6, line 131. PanIN missing acronym.

Page 7, line 167. “we found that knockdown of ZDHHC20”, should be amended into “the knockdown”.

Page 8, line 196. “N6-methyladenosine” must be amended with italic N and superscript 6 (see <https://en.wikipedia.org/wiki/N6-Methyladenosine>).

Page 11, line 294. “ZDHHC20-YTHDF3 axis”, the hyphen should be changed into an en dash.

Page 14, line 358. “ZDHHC20-YTHDF3-MYC signaling axis”, the hyphens should be changed into an en dashes (and in the rest of the paper).

Page 19, line 520: the word gene is repeated.

Page 21, line 574: “100mM”, a space should be added between the number and the units.

Page 22, line 582: “500µg², as above.

Page 22, line 584: extra spaces should be removed.

Page 22, line 59: “200 J/cm²”, cm² should be reported with the 2 as superscript.

Page 22, line 593: the word “quantitaved” should be replaced with “quantitated” or “quantified”.

Page 23, line 607: “Pancreatic cancer cells with were seeded”, the word “with” seems to be a typo.

Page 23, RNA stability assay section: the - signs should be swapped into en dash signs (–).

Page 23, line 610: “according to previously published paper” Should be amended into “according to a previously published paper”.

Page 23, line 613: “leading to following equation” should be amended into “leading to the following equation”.

Page 24, line 641: “mm³”, 3 should be reported as superscript.

Page 24, line 655: “40mg/kg”, a space should be added between the number and the units.

Page 25, line 661: “Student’s t test”, t should be italicized (also in the rest of the manuscript).

Page 25, line 662: “one-way and two-way ANOVA test”, the word test should be plural.

Page 25, line 665: “P value”, the correct way should be p italicized (therefore throughout the whole manuscript).

Page 31, line 871: “n =6”, the space should be removed.

Page 34, line 896: “n =9”, the space should be removed.

Page 41, line 1010. “(G)The degradation”, space is missing.

Page 43, line 1022. “p<0.01.” spaces are missing.

Page 43, line 1025. “|Fold Change |”, there is an extra space.

Page 48, line 1088. “0, 2, 4. 8µM.”, the point should be swapped into a comma.

Page 48, line 1089. “0, 2, 4. 8µM”, as above.

Page 48, line 1095. “100mg/kg”, a space should be added between the number and the units.

Throughout the whole manuscript, the symbol of degrees is sometimes reported with a space after the number, some other times without the space; this should be made homogenous throughout the whole manuscript.

There is a mixed use of Oxford comma e non-Oxford comma that should be amended.

Reviewer #2 (Remarks to the Author):

In this manuscript, Zhang and co-authors identified a novel palmitoylation regulation pathway in PDAC wherein ZDHHC20 promotes S-Palmitoylation of YTHDF3 at Cys474 and inhibits YTHDF3 lysosomal degradation, the stabilized YTHDF3 then promotes m6A modification of MYC mRNA CRD, resulting in the stabilization of MYC mRNA. The data is substantial and generally supporting the main conclusion. Overall, this is an important finding demonstrating the the role of deregulated Ras-ZDHHC20-YTHDF3-MYC signaling axis in PDAC. I have a few minor concerns:

1. Although the upstream signal is well characterized it would be great if the authors could further characterize the mechanism underlying the m6A modification of MYC mRNA CRD regulation of MYC mRNA stability. Does this affect CRD-BP and/or HuR binding and regulation of MYC mRNA stability?
2. Statistics is missing in some of the panels such as Fig. 2C.
3. PLA assays need single antibody staining as controls (Fig. 4H)
4. Typos: Lane 165: "increased" should be "decreased"; Lane 489: "Polybrene" should be "purimycin". Please check carefully.

Reviewer #3 (Remarks to the Author):

The manuscript by Xin Jin and colleagues described zDHHC20 plays an important role in pancreatic cancer development. zDHHC20 catalyzes the palmitoylation of YTHDF3 to stabilize the expression of YTHDF3 which further stabilize the mRNA of Myc. It is appropriate to publish in this journal after several issues are addressed:

1. In Figure 2, the author showed that KRas G12D mutant regulates zDHHC20 level through STAT3. Did you see more enrichment of STAT3 at the promoter region of zDHHC20 when activated by KRas mutant? Also, when treated with KRas mutant inhibitor or STAT3 inhibitor, does the binding affinity decrease at the promoter region?
2. In Figure 3, why all the KPC mouse model related experiments only contains one shRNA data?
3. In Figure 5, zDHHC20 catalytic active and inactive mutant should be put back to see the palmitoylation of YTHDF3 is dependent on zDHHC20 enzyme catalytic activity.
4. In Figure 6, Does YTHDF3 WT and C474S still show degradation differences in zDHHC20 KO cells?

Reviewer #4 (Remarks to the Author):

In the present study, the authors demonstrated that KRAS mutations could induce ZDHHC20-mediated S-palmitoylation of YTHDF3, thereby suppressing lysosomal

localization and degradation of YTHDF3 and regulating MYC mRNA stability to promote PDAC progression. Based on these findings, the authors designed a biologically active YTHDF3-derived peptide to competitively inhibit YTHDF3 palmitoylation mediated by ZDHHC20, which in turn downregulated MYC expression and inhibited the progression of KRAS mutant pancreatic cancer. The experiments are well designed and the data are well organized. In my opinion, this is an interesting study and is suitable for publication in Nature Communications after the authors address the following issues.

1. The authors proposed that ZDHHC20 inhibited the degradation of YTHDF3 via decreasing its lysosomal localization. What are the potential mechanisms for the decreased lysosomal localization of YTHDF3 after S-palmitoylation?
2. In Fig. 6E and 6H, quantitative analysis of the co-localization of YTHDF3 with lysotracker is recommended. In addition, the lysotracker is misspelled in Fig. 6E and 6H.
3. In Fig. 6H, the data can not support the conclusion that YTHDF3 C474S mutation induced an increase in the lysosomal localization of exogenous YTHDF3. As it seems that the flag doesn't co-localize with lysotracker in C474S group.
4. It has been reported that ZDHHC20 promotes pancreatic cancer metastasis. Did the ZDHHC20-YTHDF3-MYC signaling axis promote pancreatic cancer metastasis?
5. In Fig. 7F, the annotation of YTHDF3 and MYC is missed.
6. In Fig. 1, the layout of the graph needs to be further optimized, for example, the order of C and D.

Reviewer #5 (Remarks to the Author):

In the current study, Zhang and colleagues present an elegant model where mutant KRas regulates the expression of the palmitoylation enzyme ZDHHC20 in a process that requires the activation of STAT3. By performing bioinformatic analyses of publicly available datasets from PAAD, TCGA, the authors show that ZDHHC20 is associated with worse prognosis in PDAC patients. This epidemiological data is supported by their experiments using mouse models of pancreatic cancer, where they show that ZDHHC20 is required for the progression of PDAC. Mechanistically, the oncogenic capacity of ZDHHC20 is mediated by ZDHHC20 palmitoylation of a conserved residue of YTHDF3 (Cysteine 474 in mouse and human), a Methyladenosine RNA Binding Protein F3, with an unknown role in PDAC progression. Furthermore, the authors claim that "ZDHHC20-mediated palmitoylation of YTHDF3-474 suppressed the lysosomal localization and degradation of YTHDF3". When drilling down into the molecular mechanism, the authors found that ZDHHC20 regulates the stability of Myc mRNA, whose activation mechanism is summarised in the last section of the paper.

This elegant study provides an interesting association between ZDHHC20, YTHDF3 and MYC during PDAC progression. However, the current study must be improved to be suitable for publication in Nature Communications.

Major points

a) the majority of the results in the paper are based on in vitro experiments. Does the pancreas-specific deficiency of ZDHHC20 or YTHDF3 reproduce the effect observed in the cell lines when injected into the mice? Or does pharmacological manipulation of the STAT3, ZDHHC20 axis reduce tumour progression?

b) suboptimal demonstration of the role of STAT3, ZDHHC20 and YTHDF3 in the progression of PDAC.

- 1) Does STAT3 overexpression upregulate ZDHHC20?
- 2) Does ZDHHC20 overexpression affect KRas and/or STAT3 signaling?
- 3) Although BxPC-3 cells do not have mutant KRas, the authors observed the same effect in BxPC-3 as in PanC1, which suggests that the role of KRas does not have a genuine role in activating the axis ZDHHC20-STAT3. The authors should clarify this.
- 4) The mechanistic link with the lysosomal degradation pathway needs to be supported by more and more precise data.
- 5) The author should demonstrate that in PDAC samples, there is a correlation between ZDHHC20, STAT3, and YTHDF3 staining in cells high in MYC levels.

Minor points

Figure 1

- E) Which is the gene displayed in the graph in e)?
- H) the Log₂ fold change for ZDHHC20 is only 1.004, whereas the protein expression by IHC is much larger than this. The author should comment on the discussion since it calls for a post-transcriptional modification of ZDHHC20.
- K) Graph in K lacks the statistical test and does not provide the number of replicates in each condition.

Figure 2

- C) The graph lacks the statistical test and does not provide the number of replicates in each condition.
- E) Please define which PanINs (1,2,3) were included in the analyses.
- F) It would be interesting to know if the authors show a similar downregulation of ZDHHC20 upon treatment with inhibitors for other kinases.
- M) Please add the genomic sequence of the amplified region (distance from TSS).

Figure 4

- A) which cell line was used to perform the IP-MS? Provide more details of the number of peptides identified in the analyses and the enrichment of the bait protein vs the others.
- H) The image is blurry and not clear. Please provide the negative control of the PLA experiment. PLA is known to give a lot of false positive results.

Figure 6

- C) Repeat the same experiments using other Lysosomotropic drugs such as Chloroquine and Leupeptine. Perform LysoTracker staining and quantify by cytometry. Lysosomotropic drugs can activate or reduce lysosomal activity depending on the dose and length of the treatment.
- E) LysoTracker is not a sensitive marker for Lysosomes by IF. Repeat the IF using LAMP-1 or LAMP-2, which works well by IF. We have tried this antibody, which works well in human cells (CST, D2D11).

Figure 8

Since the role of MYC in pancreatic cancer is commonly associated with the hyperproliferation of tumour cells, please confirm that the tumours that are high for ZDHHC20, YTHDF3 and MYC are also higher for Ki67.

SF8

- B) Provide more information about the tool used to generate this graph. If a public website, please add the link.

References

The authors may consider adding the following references

role of MYC in pancreatic homeostasis and PDAC initiation (PMID: 28159836).

Palmitoyl transferases act as novel drug targets for pancreatic cancer (PMID: 37038141).

IP-MS of protein in total pancreas lysates (PMID: 37353485).

In summary, in my opinion, this study is suitable for publication in Nature Communication but requires a major revision.

REVIEWER COMMENTS

Reviewer #1 (Remarks to the Author):

In “ZDHHC20-Mediated S-Palmitoylation of YTHDF3 Stabilizes MYC mRNA to Promote Pancreatic Cancer Progression”, Zhang and colleagues explored the effects of palmitoylation of YTHDF3 on the stability of c-Myc RNA, and the effects of this process on the progression of pancreatic cancer. They also identified a potential pharmacological strategy to treat pancreatic cancer, by using a peptide that mimics YTHDF3 and “sequester” its palmitoylation.

The manuscript is overall well written and clear. The findings are exciting and represent important advances of significance to specialists within various fields such as pancreatic cancer, cancer pharmacology, RNA, RNA-binding proteins, and epitranscriptomic fields. I personally found very exciting that the authors describes the first post-transcriptional modification of an m6A-reader: with their work they pave the way to a new point of view on the analysis of the canonical and non-canonical m6A readers. Moreover, the pharmacological approach is very promising and absolutely worth the attention of the scientific community. I do recommend the publication of this work, after addressing a series of major and minor issues.

Authors' Response: Authors' Response: We would like to express our great appreciation to you for the in-depth reading of the manuscript and for giving constructive suggestions which are valuable for improving our paper. We learned a lot from the comments and tried our best to revise the manuscript according to the comments.

First, the Methods section lacks the description of a series of techniques used in the work:

1. All the bioinformatics analyses (GEPIA, TCGA, RIP-seq, TIMER2.0 and more) are not detailed in the methods. These must be detailed in the Methods and, to help the readers who are not familiar with these software/servers, explain concisely their scope.

Authors' Response: Special thanks for these comments. All the bioinformatics analyses are detailed in the Methods as below.

“Data mining and bioinformatics analysis

GEPIA (<http://gepia2.cancer-pku.cn>) and TCGA-PAAD were used for the expression profile of palmitoyl acyltransferases and patient survival in pancreatic cancer. (GEPIA Expression Analysis; Box Plots; Gene A: ZDHHCs; Datasets: PAAD).

TIMER2.0 (<http://timer.comp-genomics.org/timer/>) Gene_Corr module was to explore the

correlation between STAT3, ZDHHC20, YTHDF3 and MYC in various cancer types. TIMER2.0 Gene_Mutation module compares the differential ZDHHC20 expression between different mutation status of KRAS (Mutated Gene: KRAS; Gene Expression: ZDHHC20). KNOCKTF2.0 (<https://bio.liclab.net/KnockTFv2/>) database, provided comprehensive transcription (co-)factors knockdown/knockout dataset resource across multiple tissue/cell types of different species, was used to analyze the potential transcription (co-)factors that most significantly regulate ZDHHC20 transcription (Search by Target Gene; Species: Homo sapiens; Gene Name Type: Gene Symbol; Gene Name: ZDHHC20; Fold Change: 2).”

2. The ChIP-qPCR method must be described.

Authors’ Response: We really appreciate for this thoughtful comment. The ChIP-qPCR method was provided as below.

“CHIP-qPCR assay

CHIP-qPCR assay in pancreatic cancer cells was performed with Pierce ChIP Kit, Agarose (Cat. #26156, Thermo Fisher Scientific, Waltham, MA, USA) according to the manufacturer’s instructions. Briefly, crosslinking was performed by 1% formalin, and Nuclease (ChIP Grade) was used to fragment the DNA. The following antibodies were utilized to immunoprecipitate crosslinked protein-DNA complexes: rabbit anti-STAT3 (10253-2-AP, Protein tech) and Rabbit IgG (A7016, Beyotime). Eluted DNA fragments were analyzed by qPCR. ChIP-seq analysis revealed the presence of a STAT3 binding peak in the transcription initiation region of ZDHHC20 and the primers are listed in Supplementary Table S2.”

3. The mass spectrometry sample preparation, MS/MS, and analysis description is wholly absent, and it must be added.

Authors’ Response: Thank you for this comment. We have added the information of the mass spectrometry sample preparation, MS/MS, and analysis as below.

“LC-MS/MS

The bead samples obtained from the immunoprecipitation experiment were washed three times with precooled PBS to remove the remaining detergent. Then, bead samples were incubated in reaction buffer (1% SDC; 100 mM Tris-HCl, pH 8.5; 10 mM TCEP; 40 mM CAA) at 60 °C for 1 h for protein denaturation, cysteine reduction and alkylation. The eluates were diluted with an equal volume of H₂O and subjected to trypsin digestion overnight at 37 °C by adding 1 µg of trypsin. The peptide was purified using homemade SDB desalting columns. The eluate was vacuum dried and stored at -20 °C for later use. LC-MS/MS data acquisition was carried out on a Q Exactive HF mass spectrometer coupled with UltiMate 3000 RSLCnano system. MS raw data were analyzed with MaxQuant (V2.0.1) using the

Andromeda database search algorithm. Spectra files were searched against the UniProt Human proteome database and target protein file using the following parameters: type standard. The protein identification via mass spectrometry (MS) was performed by SpecAlly Life Technology Co., Ltd, Wuhan, China.”

4. The cloning/mutagenesis of various plasmids is mentioned throughout the manuscript, and no methodological description is provided in the paper. This must be added.

Authors’ Response: We really appreciate for this thoughtful comment. We have added the information of methodological description as below.

“Plasmid construction

Lentiviral vectors plasmids were constructed by GENECHM Biotech at Shanghai, China (<http://genechem.bioon.com.cn/>). Briefly, the expression vectors encoding ZDHHC20, YTHDF3, KRAS G12D, STAT3 and GFP-Y were generated by inserting synthesized complementary DNAs into the GV493 or GV492 vector. The ZDHHC20 C156S/F171A and YTHDF3 C474S mutants were generated by site-directed mutagenesis PCR reaction using platinum QuikChange site-directed mutagenesis kit (Stratagene) according to the manufacturer’s instructions. All plasmids were sequenced to confirm whether the designed mutation was present, without any other unwanted mutation.”

5. The Me-RIP experiments are not described in the Methods, and this must be added.

Authors’ Response: Special thanks for this comment. We have added the information of the Me-RIP experiments in the Methods as below.

“Methylated RNA Immunoprecipitation (MeRIP) assay

The MERIP assay was performed using the EpiQuik™ CUT&RUN m6A RNA Enrichment (MeRIP) Kit according to the manufacturer’s instructions. Briefly, immunocapture solution was prepared by adding the reagents to 0.2 ml PCR tubes and rotating at room temperature for 90 min. 10 µl of NDE (Nuclear Digestion Enhancer) and 2 µl of CEM (Cleavage Enzyme Mix) were added to each tube and incubated at room temperature for 4 min. Tubes were placed on the magnetic device until the solution turned clear. The supernatant was then collected and discarded. Samples were then washed three times with 150 µl of Wash Buffer, and once with 150 µl of protein digestion buffer. Then, 20 µl of Protein Digestion Solution was added to samples before mixing and incubating at 55 °C for 15 min in a thermocycler. RNA Purification Solution and 100% ethanol were then applied to the samples and vortexed to resuspend and wash the RNA Binding Beads. Resuspended beads were then treated with 13 µl of Elution Buffer and incubated at room temperature for 5 min to release the RNA from the

beads. 13 μ l from each sample was transferred to a new 0.2 ml PCR tube for immediate use or store at -20°C .”

Following, my comments on the results section.

1. Aberrant ZDHHC20 upregulation predicts unfavorable prognosis in pancreatic cancer.

1.1 In Figure 1B, the tumor area is highlighted within blue dotted lines. Can please the authors explain how the tumor was assessed (visually, and on what basis)? Were metastasis observed? Were these considered?

Authors' Response: We really appreciate for this thoughtful comment. After dissecting the KPC mice, we observed the obvious boundary of mouse tumors by visual observation and mark these areas within blue dotted lines in Figure 1B.

In the preliminary experiment, we found that some KPC mice began to develop severe ascites, cachexia and even death after 15 weeks of treatment with control alone. Therefore, considering animal ethical concerns, we chose week 15 as time endpoint of our study. While we did not observe obvious tumor metastasis in other organs (e.g., liver, lungs, as following HE staining shows) in the KPC mouse model, perhaps because our experiments were terminated at 15 weeks when tumor metastasis had not yet occurred.

1.2 Figure S1A reports the expression of all known palmitoyl acyltransferases in pancreatic cancer based on GEPIA. Statistics is provided, but no error bars, which must be added. Figure S2/1E reports more on this matter; there is not a legend, so it is impossible for the reader to appreciate the meaning of this figure. The figures also report numerosity of T and N, which I assume mean tumor and normal, but a legend will definitely help to understand.

Authors' Response: Thank you for this comment. We have added the error bars in Figure S1A. Additionally, we have added the legend for Figure S2 and Figure 1E.

Figure S1A

“Figure 1 (E) The tissue-wise expression of ZDHHC20 in Pancreatic adenocarcinoma (PAAD) tissues and non-tumor tissues were analyzed by the GEPIA web tool. T: Tumor; N: Normal; $P < 0.01^*$.”

Figure S1 (A) Boxplot of the expression of all known palmitoyl acyltransferases in pancreatic cancer based on the GEPIA web server (T: Tumor; N: Normal).”

1.3 Similarly, Figure S3 does not report a full legend. I assumed that orange is “high expression” and green “low expression” of ZDHHC, respectively. This should be clearly explained. More importantly, this figure shows a great variability in the survival rate of the patients, based on the expression of the different ZHDDs. For example, ZDHHC20 high expression in pancreatic cancer tissue seems to correlate with low survival, but the trend is the opposite with ZDHHC5, C6, C7, and various more. Can the authors discuss this? Do the different palmitoylated proteins share palmitoylation targets? Have the authors checked if YTHDF3 is target of any other ZDHHC protein? Do the authors think that the negative prognosis always depends on the YTHDF3 palmitoylation, no matter what ZDHHC is overexpressed.

Authors' Response: Thank you for this comment. We have adjusted the annotation format in Figure S3 and supplemented responding figure legends. The cutoff was taken based on the best p-value, orange represents "high expression" of ZDHHC, and green represents "low expression" of ZDHHC.

“Figure S3(A) Survival analysis of all known palmitoyl acyltransferases in pancreatic cancer in a pancreatic cancer dataset from TCGA (Orange: ZDHHCs high expression; Green: ZDHHCs low expression).”

As for the great variability in the survival rate of the patients based on the expression of the

different ZDHHCs, we also supplement the discussion of this phenomenon as below.

“Meanwhile, when evaluating the prognosis significance of all ZDHHCs in pancreatic cancer, we found great variability in the survival rate of the patients based on the expression of different ZDHHCs. It may suggest the complexity of different ZDHHCs regulating the biological functions of pancreatic cancer¹, and we will also further explore the post-translational modifications and functions of ZDHHCs in pancreatic cancer and try to elucidate this complex regulatory network.”

It has been reported in previous studies that some proteins have indeed been found to be modified by different ZDHHCs, such as STING². Therefore, we also examined the binding of other ZDHHC molecules to YTHDF3 and the data showed that no other ZDHHCs were found to bind to YTHDF3 in pancreatic cancer cells except for ZDHHC20 (Figure 4H). Because we only found that ZDHHC20 can mediate palmitoylation at the YTHDF3-C474 locus, which in turn stabilized c-Myc and promoted pancreatic cancer progression. Follow-up experiments are needed to further explore whether palmitoylation at other sites of YTHDF3 exists and how this relates to pancreatic cancer progression as well as prognosis.

Figure S3

Figure 4H

1.4 Figure 1K is missing the statistics (asterisks) on the figure.

Authors' Response: Thank you for this comment. We have added the statistics (asterisks) on Figure 1K.

Figure 1K

1.5 In line 119, the authors state that KRAS is low in BxPC-3 cell line; a reference is needed to support that.

Authors' Response: Special thanks for this comment. As shown in Figure 2A, our experimental results showed that the expression of ZDHHC20 was lower in BxPC-3 than in other pancreatic cancer cell lines. Moreover, it has been reported that BxPC-3 is a KRAS wild-type pancreatic cancer cell line, and we have added the relevant reference.

Figure 2A

2. KRAS mutations induce the accumulation of ZDHHC20 in pancreatic cancer via STAT3.

2.1 Figure 2C reports on the expression of the ZDHHC20 protein in different PDAC lines. It is not clear how these data were obtained (western blots? IHC?) and the statistics are missing.

Authors' Response: Thank you for this comment. Recently, Yao W, et al reported the comparison of KRAS-ON with KRAS-OFF paired samples identified 196 upregulated proteins, including ZDHHC20³. We presented and cited the expression ratio of ZDHHC20 in KRAS ON/OFF pancreatic cancer cells from that study. We apologize for choosing an inappropriate graphic to display the results and have corrected it in Figure S4A.

A

Accession	Gene	AK10965	AK196
		Ratio_KRAS ON/OFF	
Q5Y5T1	ZDHHC20	1.799	1.513

Figure S4A

2.2 Figure 2G shows the expression of ZDHHC20 PDAC lines treated with a KRAS inhibitor. Particularly for AsPC-1, the change in the signal is not very much. Can the author add a quantification of the biological/technical repeats and show the statistics to corroborate the decrease in ZDHHC20 stated in this section of the manuscript? Figure 2H reports on the RT-qPCR analysis for the mRNA of ZDHHC20 in PDAC lines infected with KRASG12D plasmids. Could the authors state how the ZDHHC20 mRNA levels in these cell lines and vector-transfected cell lines relate, to show better the effect of the overexpression (as it stands, it is not clear if the reader can assume that the endogenous mRNA levels are the same as in the vector-transfected cells)?

Authors' Response: We really appreciate for this thoughtful comment. The previous Figure 2G was quantified as well as statistically analyzed (Figure 2E), and the expression of ZDHHC20 was indeed down-regulated in a dose-dependent manner in AsPC-1 cells treated with KRAS inhibitor. Additionally, the RT-qPCR analysis of ZDHHC20 mRNA in KRASG12D plasmid-infected PDAC cell lines was also re-performed, and we found no difference in ZDHHC20 mRNA levels between untreated cells and vectored cells (Figure S4C).

Figure 2E

Technical repeats in AsPC-1

Figure S4C

2.3 Figure S4A reports the prediction of the transcription factors that target ZDHHC20 based on the KnockTF platform. Not being familiar with this platform, I found this panel quite complex to understand. It is unclear whether the fold change entails protein or mRNA and whether this is normalized to the knockdown, silencing, or inhibition of ZDHHC20. Moreover, why is this defined as a prediction? Is it not based on collected data? In the prediction, seven proteins show a great fold change: what was the rationale to exclude 6 out of 7? Were the other proteins tested too?

Authors' Response: Thank you for this comment. We are very sorry for our negligence and we have supplemented more details in the method description, as follows. As for the word “predict”, we misstated this in the manuscript and have corrected as below.

“KNOCKTF2.0 (<https://bio.liclab.net/KnockTFv2/>) database, provided comprehensive transcription (co-)factors knockdown/knockout dataset resource across multiple tissue/cell types of different species, was used to analyze the potential transcription (co-)factors that most significantly regulate ZDHHC20 transcription (Search by Target Gene; Species: Homo sapiens; Gene Name Type: Gene Symbol; Gene Name: ZDHHC20; Fold Change: 2).”

“To explore the mechanism underlying this upregulation, we used the KnockTF platform to analyze the transcription factors that target ZDHHC20.”

STAT3 was found to be the transcription factor with the most significant regulatory effect on ZDHHC20 mRNA. Multiple ChIP-seq analyses showed conspicuous and consistent binding peaks, while other transcription factors did not have such significant binding peaks. Thus, we choose STAT3 and the ChIP-qPCR results demonstrated that STAT3 can bind to the promoter region of ZDHHC20.

2.4 The western blots in Figures S4C and 2K are missing the quantification and relative statistics. Particularly, with BP-1-102 treatment in AsPC-1, the change seems quite minor (and the GAPDH signal decreases too), so the quantification will be beneficial. Figure S4D/E reports the correlation between STAT3 and ZDHHC20 in various cancer types calculated with TIMER2.0. The authors must provide a detailed explanation on how these data were generated. Additionally, can the authors comment on whether the observed correlation is specific to pancreatic cancer?

Authors' Response: Special thanks for this comment. We supplemented quantification with statistics relative to GAPDH in the western blot of previous Figure S4C and 2K (Figure 2H and S4F). Quantitative analysis from triplicate experiments in AsPC-1 showed that BP-1-102 treatment significantly downregulated ZDHHC20 expression. As for Figure S4D/E, the method description of the correlation analysis with TIMER2.0 is also provided in the Methods, as follows.

“TIMER2.0 (<http://timer.comp-genomics.org/timer/>) Gene_Corr module was to explore the correlation between STAT3, ZDHHC20, YTHDF3 and MYC in various cancer types.”

The results of TIMER2.0 analysis suggest that the correlation between STAT3 and ZDHHC20 may be widespread in a variety of tumors. We also found that STAT3 knockdown could downregulate ZDHHC20 in lung and colon cancer cells, and this remains to be further explored in future study.

Figure 2H and S4F

Technical repeats in AsPC-1

2.5 Figures 2L/2M report on the ChIP-seq data. It is difficult for me to comment on this panel as the method section is missing. I am assuming that the immunoprecipitation was performed with an antibody against STAT3, but this must be detailed in the Methods.

Authors' Response: Thank the Reviewer for pointing this out. We are very sorry for our negligence of method description and have added the description of ChIP-qPCR assay in the Methods as below.

"ChIP-qPCR assay in pancreatic cancer cells was performed with Pierce ChIP Kit, Agarose (Cat. #26156, Thermo Fisher Scientific, Waltham, MA, USA) according to the manufacturer's instructions. Briefly, crosslinking was performed by 1% formalin, and Nuclease (ChIP Grade) was used to fragment the DNA. The following antibodies were utilized to immunoprecipitate crosslinked protein-DNA complexes: rabbit anti-STAT3 (10253-2-AP, Protein tech) and Rabbit IgG (A7016, Beyotime). Eluted DNA fragments were analyzed by qPCR. ChIP-seq analysis revealed the presence of a STAT3 binding peak in the transcription initiation region of ZDHHC20 and the primers are listed in Supplementary Table S2."

2.6 Figure 2N must report a proper quantification.

Authors' Response: Thank you for this comment. We have added the quantification for previous Figure 2N.

Figure 2M

3. ZDHHC20 promotes pancreatic cancer progression in a palmitoylation-dependent manner.

3.1 Figures 3C-3E. The transwell assay is not detailed in the methods. Although a commercial kit-based, this must be described.

Authors' Response: We really appreciate for this thoughtful comment. We have added the description of transwell assay in the Methods as below.

“Pancreatic cancer cells were cultured into the 24-well Transwell plates with 8.0 μm pores (Corning Costar, USA) with precoated Matrigel (BD, USA; diluted 1: 8). Briefly, 5×10^4 pancreatic cancer cells were seeded into the top chamber. After cultivation for 24 hours, the membrane was collected and stained with the Crystal Violet Solution (Solarbio, China). A cotton swab was used to remove those cells that did not migrate or invade through the pores. The migrating cells were counted and photographed by a microscope from five different microscopic fields per well.”

4. YTHDF3 mediates the oncogenic capacity of ZDHHC20 in pancreatic cancer

4.1 Figure 4A shows the silver staining of the gel electrophoresis after Flag-ZDHHC20-immunoprecipitation. The bands of YTHDF3 and FLAG-ZDHHC20 are indicated based exclusively on MW, but this is just a hypothetical position, as no western blot is provided. The caption should be rephrased stating that these are just predicted bands.

Authors' Response: We are very sorry for our incorrect writing. As Reviewer suggested that we rephrased the caption of Figure 4A and stated that these are just predicted bands as below.

“Figure 4 (A) Gel electrophoresis was conducted after using Flag antibody for Flag-ZDHHC20-immunoprecipitation, then silver staining was performed. The red frame indicated the YTHDF3 and FLAG-ZDHHC20 predicted based on molecular weight.”

4.2 Figure 4C reports the MS of a peptide derived from Flag-ZDHHC20-immunoprecipitates belonging to YTHDF3. I am very concerned about this panel as the sequence reported herein is shared with the paralogs YTHDF1 and YTHDF2 (See below – sequences highlighted in lower case).

>sp|Q9BYJ9|YTHD1_HUMAN YTH domain-containing family protein 1 OS=Homo sapiens OX=9606 GN=YTHDF1 PE=1 SV=1

MSATSVDTRTKGQDNKVQNGSLHQKDTVHDNDFEPYLTGQSNQSNYSYPSMSDPYL
SSYYPPSIGFPYSLNEAPWSTAGDPPIPYLTTYGQLSNGDHHFMHDAVFGQPGGLGNN
IYQHRFNFFPENPAFSAWGTSGSQGQQTQSSAYGSSYTYPPSSLGGTVVDGQPGFHSD
TLKAPGMNSLEQGMVGLKIGDVSSAVKTVGSSVSSVALTGVLSGNGGTNVNMPV
SKPTSWAAIASKPAKQPKMKTKSGPVMGGGLPPPIKHNM DIGTWDNKGPVPAKPV
PQQAPSPQAAPQPQQAQPLPAQPPALAQPYQSPQPPQTRWVAPRNRNAAFGQSG
GAGSDSNSPGNVQPNAPSVEHPVLEKLKAAHSYNPKFEFENLKSGRVFIKSYSED
DIHRSIKYSIWCSTEHGKRLDSAFRCMSSKGPVYLLFSVNGSGHFHFCGVAEMKSPVD
YGTSAQVWSQDKWKGFQVWIFVKDVPNNQLRHIRLENNNDNKPV TNSRdtqevplek
AKQVLKIISSYKHTTSIFDDFAHYEKRQEEEEVVRKERQSRNKQ

>sp|Q9Y5A9|YTHD2_HUMAN YTH domain-containing family protein 2 OS=Homo sapiens OX=9606 GN=YTHDF2 PE=1 SV=2

MSASSLLEQRPKGQGNKVQNGSVHQKDGLNDDDFEPYLSQARPNNAYTAMSDSYL
PSYYSYPSIGFSYSLGEAAWSTGGDTAMPYLTSYGQLSNGEPHFLPDAMFGQPGALGST
PFLGQHGFNFFPSGIDFSAWGNNSSQGQSTQSSGYSSNYAYAPSSLGGAMIDGQSAFA
NETLNKAPGMNTIDQGMAALKLGSTEVASNVPKVVGSVAVGSGSITSNIVASNSLPPATI
APPKPASWADIASKPAKQPKLTKNGIAGSSLPPPIKHNM DIGTWDNKGPVAKAPS
QALVQNIQPTQGSPPVGGQANNSPPVAQASVGGQQTQPLPPPPQPAQLSVQQQAA
QPTRWVAPRNRGSGFGHNGVDGNGVGSQAGSGSTPSEPHVLEKLRSINNYNPKDF
DWNLKHGRVFIKSYSEDDIHRSIKYNWCSTEHGKRLDAAAYRSMNGKGPVYLLFS
VNGSGHFHFCGVAEMKSAVDYNTCAGVWSQDKWKGRFDVRWIFVKDVPNSQLRHIRL
ENNENKPV TNSRdtqevplekAKQVLKIIASYKHTTSIFDDFSHYEKRQEEEEVVKKERQG
RGK

How can the authors unequivocally state that the interaction is with YTHDF3 (from the MS data)? Also, the description of these data is way too concise, it must be rephrased so that a better understanding is ensured.

Authors' Response: We thank the Reviewer for this great question. It is really true as Reviewer suggested that the peptides sequence reported herein is shared with the paralogs

YTHDF1 and YTHDF2. From the search file, protein YTHDF3, YTHDF1 and YTHDF2 have shared peptides, and are grouped in one protein group. In addition, when multiple matches were found, the best scoring protein (YTHDF3) were presented in the 'Leading Razor Protein' column. Therefore, we carried out the western blot analysis of YTHDF1, YTHDF2 and Flag-ZDHHC20 proteins reciprocally immunoprecipitated by anti- Flag and in PANC-1 and BxPC-3 cells as below. And combined with the positive anti-YTHDF3 WB experiment, it is adequate to make a conclusion that YTHDF3 is target interactor of ZDHHC20. Meanwhile we have re-written the description of the IP/MS and co-immunoprecipitation (Co-IP) results, as follows.

“Sliver staining for gel electrophoresis of Flag-ZDHHC20-immunoprecipitation in PANC-1 cells was conducted to show potential interactions based on molecular weight (Figure 4A). The IP-MS assay showed that there are potential interactions between ZDHHC20 and YTHDF3, an N6-methyladenosine (m6A) reader (Figure 4B-4C). It has been reported that YTHDF3 is overexpressed in multiple kinds of tumors and promotes tumor progression, but its regulatory role in pancreatic cancer has not been explored. HA-ZDHHC20 and Flag-YTHDF3 expression plasmids were transfected into 293T cells, and the co-immunoprecipitation (Co-IP) assay showed the interaction of exogenously expressed ZDHHC20 with YTHDF3 (Figure 4D). Then, we also detected the interaction of ZDHHC20 with YTHDF3, not YTHDF1 or YTHDF2, in PANC-1 and BxPC-3 cells (Figure 4E, 4F and S6A).”

Figure S6A

5. ZDHHC20 regulates the palmitoylation of YTHDF3 in pancreatic cancer.

5.1 Figure 5A shows the results obtained with a commercial kit which is not detailed in the Methods. The description must be added.

Authors' Response: Thank you for this comment. We have added the description of Acyl-biotinyl exchange (ABE) assay in the Methods as below.

“Acyl-biotinyl exchange (ABE) assay

The ABE assay was performed using the IP-ABE Palmitoylation Kit For WB (catalog number AM10313; AIMSMASS, Shanghai, China) according to the manufacturer’s instructions. Briefly, the procedure includes blocking, reduction, labelling, elution and detection. Cells were harvested and suspended in lysis buffer followed by incubation with beads and anti-YTHDF3 or anti-Flag overnight at 4°C. N-ethylmaleimide (NEM) was used to block the unmodified cysteines for 30 min. Then, the beads were washed and incubated with hydroxylamine (HAM) for 1 h at room temperature. Each group was divided into two parts, one including the HAM step (+HAM) and one omitting the HAM cleavage step (-HAM). After washing, the beads were treated with thiol-reactive biotin molecules for 1 h at room temperature. The beads were boiled for 10 minutes after resuspending in SDS-PAGE loading buffer and then subjected to western blotting analysis with Streptavidin-HRP antibody.”

5.2 At page 9, lines 235-236, the authors state that “this uniquely predicted site is also located in the binding region of ZDHHC20 and YTHDF3.” Based on what evidence they believe that the peptide reported is part of an interaction surface? This is not supported by any of the data reported in the manuscript.

Authors’ Response: We thank the Reviewer for this great question. Our results showed that the Cys474 was located in the C-terminal domain of YTHDF3, which was the binding region with ZDHHC20 in Figure 4K. We apologize for the misdescription and have reworded the result description as follows.

“Notably, this uniquely predicted site is located in the C-terminal domain of YTHDF3, also the binding region of ZDHHC20.”

6. ZDHHC20 suppresses the degradation of YTHDF3 via palmitoylation on Cys474. 6.1 Figure 6B reports a western blot: signal quantification is missing, and it must be added.

Authors’ Response: Thank you for this comment. We have added the signal quantification for western blot of Figure 6B.

Figure 6B

6.2 Figure 6H reports representative images of cancer cells infected with Flag-YTHDF3 WT/C474S plasmids in IF. Are these CHX-treated cells? If not, why is the signal of YTHDF3 in lysosome similar to the one of sgZDHHC20? Does it mean that YTHDF3 does get palmitoylated by other enzymes? This should be detailed.

Authors' Response: We really appreciate for this thoughtful comment. We feel very sorry for the mistake in Figure 6H. There cells were not CHX-treated, and the top and bottom two sets of previous Figure 6H were inverted previously. We have corrected the figure as below. And the co-immunoprecipitation assay showed that YTHDF3 only interacted with ZDHHC20 but not with other ZDHHCs in pancreatic cancer cells.

Figure 6G

Figure 4H

7. YTHDF3-mediated m6A modification regulates MYC mRNA stability.

7.1 I believe that Figure 7F is missing the labelling of YTHDF3 and c-Myc (similar to 7E). Generally speaking, this is the weakest link in the manuscript. Although the authors provide a good characterisation of the binding of YTHDF3 to m6A-RNA (this was already well known in the literature anyway), the overall connection between YTHDF3 palmitoylation and m6A binding is not fully explored and discussed. Based on 7F, the inhibition of the palmitoylation of YTHDF3 leads to less binding of c-MYC RNA, as recapitulated by the C474S mutant in RIP-seq and MeRIP-Seq assays, suggesting a connection between ZHDCC-dependent

palmitoylation and c-MYC RNA binding, not necessarily with m6A-RNA. In the luciferase assay, the DRACH motifs are completely altered (A-to-T) and it is known from structural data that the YTH requires an A in that position to bind RNA, independently from the presence of the methylation (Nucleic Acids Res. 2014 Dec 16;42(22):13911-9, and similar studies). To robustly state the connection with m6A-RNA, the authors must repeat the luciferase assay with a fully methylated RNA, with and without 2-BP treatment.

Authors' Response: Thank you for this comment. We have added the labeling of YTHDF3 and c-Myc in Figure 7F.

Figure 7F

We would like to express our sincere thanks to the Reviewer for the constructive and positive comments. Supplementation and optimization of the original scheme were carried out according to the Reviewer's suggestion. YTHDF3 which contains a YTH domain has increased affinities for the methylated compared to the non-methylated form of the same RNA target sequence. And the completely mutant (A-to-T) DRACH motifs may reduce the affinity of YTHDF3 for c-MYC mRNA. With this in mind, we supplemented the wild-type luciferase experiments with and without 2-BP treatment or ZDHHC20 knockout, and found that 2-BP treatment or ZDHHC20 knockout significantly downregulated the Fluc activity of the wild-type reporter (Fig. 7Q and S9H).

Figure 7Q and S9H

7.2 In addition, the authors must consider the fact that C474 is in close vicinity of one the residues of the tryptophan-cage of YTHDF3, the binding surface responsible for the coordination of the N6-methyl moiety of m6A (see J. Chem. Inf. Model. 2020, 60, 12). Hence, two (maybe more?) scenarios are possible: i) the palmitoylation is inducing a conformational change that increases the RNA overall binding properties; ii) the mutation of C474 into serine might be altering the conformation of the tryptophan-cage and partially disrupting the ability to bind m6A in a manner that is completely independent from the palmitoylation. All considered, the claim that “m6A modifications in the MYC CRD are required for YTHDF3-mediated stabilization of MYC mRNA in pancreatic cancer” requires more structural data that are not herein provided. As I totally understand that a biophysical characterisation of this process is not in the scope of the paper, I think the authors should just include a more detailed description of the possible scenarios and all these limitations of their characterisation in their discussion.

Authors' Response: Thank you for this comment. As the Reviewer said, more structural data are required to explore the effect of palmitoylation on the tryptophan conformation of YTHDF3. Regrettably we were not able to elucidate these possible scenarios, and we add these limitations in our discussion.

“Because Cys474 is in close vicinity of one the residues of the tryptophan-cage of YTHDF3, the binding surface responsible for the coordination of the N6-methyl moiety of m6A ⁴, we don't know whether palmitoylation and C to S mutant may cause a conformational change in the tryptophan cage. To answer these questions, more biochemical, structural, and in silico studies on the YTHDF3 proteins are required.”

Following, a list of minor typos that require attention.

Page 6, line 131. PanIN missing acronym.

Page 7, line 167. “we found that knockdown of ZDHHC20”, should be amended into “the knockdown”.

Page 8, line 196. “N6-methyladenosine” must be amended with italic N and superscript 6 (see <https://en.wikipedia.org/wiki/N6-Methyladenosine>).

Page 11, line 294. “ZDHHC20-YTHDF3 axis”, the hyphen should be changed into an en dash.

Page 14, line 358. “ZDHHC20-YTHDF3-MYC signaling axis”, the hyphens should be changed into an en dashes (and in the rest of the paper).

Page 19, line 520: the word gene is repeated.

Page 21, line 574: “100mM”, a space should be added between the number and the units.

Page 22, line 582: “500µg², as above.

Page 22, line 584: extra spaces should be removed.

Page 22, line 59: “200 J/cm²”, cm² should be reported with the 2 as superscript.

Page 22, line 593: the word “quantitaved” should be replaced with “quantitated” or “quantified”.

Page 23, line 607: “Pancreatic cancer cells with were seeded”, the word “with” seems to be a typo.

Page 23, RNA stability assay section: the - signs should be swapped into en dash signs (–).

Page 23, line 610: “according to previously published paper” Should be amended into “according to a previously published paper”.

Page 23, line 613: “leading to following equation” should be amended into “leading to the following equation”.

Page 24, line 641: “mm³”, 3 should be reported as superscript.

Page 24, line 655: “40mg/kg”, a space should be added between the number and the units.

Page 25, line 661: “Student’s t test”, t should be italicized (also in the rest of the manuscript).

Page 25, line 662: “one-way and two-way ANOVA test”, the word test should be plural.

Page 25, line 665: “P value”, the correct way should be p italicized (therefore throughout the whole manuscript).

Page 31, line 871: “n =6”, the space should be removed.

Page 34, line 896: “n =9”, the space should be removed.

Page 41, line 1010. “(G)The degradation”, space is missing.

Page 43, line 1022. “p<0.01.” spaces are missing.

Page 43, line 1025. “|Fold Change |”, there is an extra space.

Page 48, line 1088. “0, 2, 4. 8µM.”, the point should be swapped into a comma.

Page 48, line 1089. “0, 2, 4. 8µM”, as above.

Page 48, line 1095. “100mg/kg”, a space should be added between the number and the units.

Throughout the whole manuscript, the symbol of degrees is sometimes reported with a space after the number, some other times without the space; this should be made homogenous throughout the whole manuscript.

There is a mixed use of Oxford comma e non-Oxford comma that should be amended.

Authors’ Response: We really appreciate for these thoughtful comments. We have made modification in corresponding parts as the Reviewer suggested.

Special thanks to you for your good comments.

Reviewer #2 (Remarks to the Author):

In this manuscript, Zhang and co-authors identified a novel palmitoylation regulation pathway in PDAC wherein ZDHHC20 promotes S-Palmitoylation of YTHDF3 at Cys474 and inhibits YTHDF3 lysosomal degradation, the stabilized YTHDF3 then promotes m6A modification of MYC mRNA CRD, resulting in the stabilization of MYC mRNA. The data is substantial and generally supporting the main conclusion. Overall, this is an important finding demonstrating the the role of deregulated Ras-ZDHHC20-YTHDF3-MYC signaling axis in PDAC. I have a few minor concerns:

Authors' Response: We sincerely appreciate the time and effort that you dedicated to providing feedback on our manuscript and are grateful for the insightful comments and valuable improvements to our paper.

1. Although the upstream signal is well characterized it would be great if the authors could further characterize the mechanism underlying the m6A modification of MYC mRNA CRD regulation of MYC mRNA stability. Does this affect CRD-BP and/or HuR binding and regulation of MYC mRNA stability?

Authors' Response: Special thanks to the reviewer for these constructive comments, and we performed immunoprecipitation and were pleasantly surprised to find that HuR and IGF2BP1 (CRD-BP) interact with YTHDF3. The data suggested an unreported function of YTHDF3 in recruiting these RNA stabilizers. However, palmitoylation of Cys474 did not seem to affect this function of YTHDF3. In parallel, we knocked out HuR and IGF2BP1 in PANC-1 cells. But HuR/IGF2BP1 KO did not reverse the enhancement of MYC mRNA stability by YTHDF3 overexpression. Considering that YTHDF3 functions as an m6A reader to regulate translation and decay of N6-methyladenosine-modified RNA⁵, YTHDF3 may directly regulate MYC mRNA stability. More studies on the potential role of YTHDF3 in mRNA stabilization and RNA binding are also needed in the future.

2. Statistics is missing in some of the panels such as Figure 2C.

Authors' Response: Thank you for this comment. Recently Yao W, et al reported the comparison of KRAS-ON with KRAS-OFF paired samples identified 196 upregulated proteins, including ZDHHC20³. We presented and cited the expression ratio of ZDHHC20 in KRAS ON/OFF pancreatic cancer cells from that study. We apologize for choosing an inappropriate graphic to display the results and have corrected it in Figure S4A.

A

Accession	Gene	AK10965	AK196
		Ratio_KRAS ON/OFF	
Q5Y5T1	ZDHHC20	1.799	1.513

Figure S4A

3. PLA assays need single antibody staining as controls (Figure 4H)

Authors' Response: Special thanks for this comment. We have added the single antibody staining as controls in PLA assays (Figure 4I)

Figure 4I

4. Typos: Lane 165: "increased" should be "decreased"; Lane 489: "Polybrene" should be "puromycin". Please check carefully.

Authors' Response: Thank you for this comment. We have corrected this in corresponding parts. "The CCK8 assay, colony formation assay and Transwell assay results suggested that ZDHHC20 silencing profoundly decreased the proliferation, invasion and migration of these pancreatic cancer cell lines (Fig. 3C-3E and S5A)." (Lane 171)

"The transduced cells from single guide RNAs were selected with appropriate concentration of puromycin (Thermo Fisher Scientific, USA) for 5-7 days to achieve stable integration." (Lane 551)

Special thanks to you for your good comments.

Reviewer #3 (Remarks to the Author):

The manuscript by Xin Jin and colleagues described zDHHC20 plays an important role in pancreatic cancer development. zDHHC20 catalyzes the palmitoylation of YTHDF3 to stabilize the expression of YTHDF3 which further stabilize the mRNA of Myc. It is appropriate to publish in this journal after several issues are addressed:

Authors' Response: We thank the reviewer for recognizing the significance of our finding. We also thank the reviewer for the insightful suggestions that have helped us improve our manuscript significantly.

1. In Figure 2, the author showed that KRas G12D mutant regulates zDHHC20 level through

STAT3. Did you see more enrichment of STAT3 at the promoter region of zDHHC20 when activated by KRas mutant? Also, when treated with KRas mutant inhibitor or STAT3 inhibitor, does the binding affinity decrease at the promoter region?

Authors' Response: We would like to express our sincere thanks to the Reviewer for the constructive and positive comments. Supplementation and optimization of the original scheme were carried out according to the Reviewer's suggestion. The ChIP-qPCR results demonstrated that the enrichment of STAT3 at the promoter region of zDHHC20 was enhanced by the exogenous expression of KRAS G12D (Figure 2K). The dual-luciferase reporter assay indicated that the binding affinity decrease at the promoter region of ZDHHC20 after treatment with KRAS G12D or STAT3 inhibitor (Figure 2L and S4K).

Figure 2K

Figure 2L and S4K

2. In Figure 3, why all the KPC mouse model related experiments only contains one shRNA data?

Authors' Response: Special thanks for this comment. It is really true as Reviewer suggested that animal experiments should contain one more shRNAs. In fact, we designed two shRNAs that can effectively knock down mZDHHC20. Considering the difficulty of breeding, we chose one with the most significant knockdown effect to conduct experiments in the KPC mouse model. Meanwhile we also explored the biological functions of ZDHHC20 in pancreatic cancer progression by human-derived pancreatic cancer cell lines with cell-derived xenograft model. In future experiments, we must select multiple shRNAs to improve the rigor

of animal experiments.

3. In Figure 5, zDHHC20 catalytic active and inactive mutant should be put back to see the palmitoylation of YTHDF3 is dependent on zDHHC20 enzyme catalytic activity.

Authors' Response: Thank you very much for your comments and suggestions. It is really true as Reviewer suggested that the enzyme catalytic activity ZDHHC20 should be considered in the detection of YTHDF3 palmitoylation assays. Consistent with the previous results, two ZDHHC20 mutants without catalytic activity (C156S and F171A) couldn't mediate the palmitoylation of YTHDF3 as the wild-type (WT) ZDHHC20 (Figure 5J).

Figure 5J

4. In Figure 6, Does YTHDF3 WT and C474S still show degradation differences in zDHHC20 KO cells?

Authors' Response: We would like to express our sincere thanks to the Reviewer for the constructive and positive comments. As Reviewer suggested that we carried out CHX-chase assay to see the stabilities of Flag-YTHDF3 WT/C474S in PANC-1 and AsPC-1 cells infected with these mutant plasmids. While the YTHDF3 WT exhibited degradation similar to that of the YTHDF3 C474S in the ZDHHC20-KO cells (Figure 6I and S8D).

Figure 6I and S8D

Special thanks to you for your good comments.

Reviewer #4 (Remarks to the Author):

In the present study, the authors demonstrated that KRAS mutations could induce ZDHHC20-mediated S-palmitoylation of YTHDF3, thereby suppressing lysosomal

localization and degradation of YTHDF3 and regulating MYC mRNA stability to promote PDAC progression. Based on these findings, the authors designed a biologically active YTHDF3-derived peptide to competitively inhibit YTHDF3 palmitoylation mediated by ZDHHC20, which in turn downregulated MYC expression and inhibited the progression of KRAS mutant pancreatic cancer. The experiments are well designed and the data are well organized. In my opinion, this is an interesting study and is suitable for publication in Nature Communications after the authors address the following issues.

Authors' Response: We sincerely appreciate the time and effort that you dedicated to providing feedback on our manuscript and are grateful for the insightful comments and valuable improvements to our paper.

1. The authors proposed that ZDHHC20 inhibited the degradation of YTHDF3 via decreasing its lysosomal localization. What are the potential mechanisms for the decreased lysosomal localization of YTHDF3 after S-palmitoylation?

Authors' Response: We would like to express our sincere thanks to the Reviewer for the constructive and positive comments. To further explore the potential mechanisms for the decreased lysosomal localization of YTHDF3 after S-palmitoylation, we conducted IP-MS between the YTHDF3 WT and YTHDF3 C474S. The IP-MS assay showed that HSC70, a molecular chaperone recognized soluble cytosolic proteins bearing a KFERQ-like motif and mediated the translocation of the substrate into the lysosome for lumen degradation⁶, showed more interactions with the YTHDF3 C474S relative to the YTHDF3 WT (Figure S8E and S8F). Analysis of the YTHDF3 amino acid sequence revealed the presence of three potential KFERQ-like motifs, suggesting that YTHDF3 could be a chaperone-mediated autophagy (CMA) substrate (Figure S8G). Consistently, the Co-IP assays also corroborated more interactions of HSC70 with YTHDF3 C474S (Figure 6J). Notably, we found that YTHDF3 protein levels were downregulated in ZDHHC20-KO pancreatic cancer cells, and the knockdown of HSC70 or LAMP2A reversed the effects of ZDHHC20 deficiency on YTHDF3 protein levels in ZDHHC20-KO cells (Figure S8H and S8I). In conclusion, ZDHHC20-mediated palmitoylation of YTHDF3-Cys474 suppressed the recognition of YTHDF3 by HSC70, thereby inhibiting its subsequent lysosomal degradation through the CMA pathway.

Figure 6J and S8E-S8I

2. In Figure 6E and 6H, quantitative analysis of the co-localization of YTHDF3 with lysotracker is recommended. In addition, the lysotracker is misspelled in Figure 6E and 6H.
 Authors' Response: We thank the Reviewer for mentioning this point. We are very sorry for our negligence and have corrected the lysotracker in previous Figure 6E and 6H. Quantitative analysis of co-localization of YTHDF3 with Lysotracker was added as suggested by the reviewers.

Figure 6E and 6G

3. In Figure 6H, the data can not support the conclusion that YTHDF3 C474S mutation induced an increase in the lysosomal localization of exogenous YTHDF3. As is seems that the flag doesn't co-localize with lysotracker in C474S group.

Authors' Response: Thank you for this comment. We feel very sorry for the mistake in Figure 6H. The top and bottom two sets of Figure 6H were inverted previously. We have corrected this as below and have added the quantitative analysis of the co-localization of YTHDF3 with lysotracker.

Figure 6G

4. It has been reported that ZDHHC20 promotes pancreatic cancer metastasis. Did the ZDHHC20-YTHDF3-MYC signaling axis promote pancreatic cancer metastasis?

Authors' Response: We thank the Reviewer for raising this great comment. The transwell assay demonstrated that ZDHHC20 could promote the invasion of pancreatic cancer cells in vitro. In the preliminary experiment, we found that some KPC mice began to develop severe ascites, cachexia and even death after 15 weeks of treatment with control alone. Therefore, considering animal ethical concerns, we chose week 15 as time endpoint of our study. While we did not observe tumor metastasis in other organs (e.g., liver, lungs, as following HE staining shows) in the KPC mouse model, perhaps because our experiments were terminated at 15 weeks, when tumor metastasis had not yet occurred. In addition, MYC has been reported to be closely related to tumor metastasis, and it will be further explored whether ZDHHC20-YTHDF3-MYC signaling axis promote in vivo pancreatic cancer metastasis in future studies.

5. In Figure 7F, the annotation of YTHDF3 and MYC is missed.

Authors' Response: Thank you for this comment. We are very sorry for our negligence and have added the annotation of YTHDF3 and MYC in Figure 7F.

Figure 7F

6. In Figure 1, the layout of the graph needs to be further optimized, for example, the order of C and D.

Authors' Response: We thank the Reviewer for pointing this out. The layout of the graph has been corrected in the revised manuscript.

Figure 1C and 1D

Special thanks to you for your good comments.

Reviewer #5 (Remarks to the Author):

In the current study, Zhang and colleagues present an elegant model where mutant KRas regulates the expression of the palmitoylation enzyme ZDHHC20 in a process that requires the activation of STAT3. By performing bioinformatic analyses of publicly available datasets from PAAD, TCGA, the authors show that ZDHHC20 is associated with worse prognosis in PDAC patients. This epidemiological data is supported by their experiments using mouse models of pancreatic cancer, where they show that ZDHHC20 is required for the progression

of PDAC. Mechanistically, the oncogenic capacity of ZDHHC20 is mediated by ZDHHC20 palmitoylation of a conserved residue of YTHDF3 (Cysteine 474 in mouse and human), a Methyladenosine RNA Binding Protein F3, with an unknown role in PDAC progression. Furthermore, the authors claim that "ZDHHC20-mediated palmitoylation of YTHDF3-474 suppressed the lysosomal localization and degradation of YTHDF3". When drilling down into the molecular mechanism, the authors found that ZDHHC20 regulates the stability of Myc mRNA, whose activation mechanism is summarised in the last section of the paper. This elegant study provides an interesting association between ZDHHC20, YTHDF3 and MYC during PDAC progression. However, the current study must be improved to be suitable for publication in Nature Communications. In summary, in my opinion, this study is suitable for publication in Nature Communication but requires a major revision.

Authors' Response: On behalf of my co-authors, we thank you very much for giving us an opportunity to revise our manuscript. We appreciate you very much for the positive and constructive comments on our manuscript. Those comments are valuable and helpful for revising and improving our paper, as well as the important guiding significance to our future research.

Major points

(a) the majority of the results in the paper are based on in vitro experiments. Does the pancreas-specific deficiency of ZDHHC20 or YTHDF3 reproduce the effect observed in the cell lines when injected into the mice? Or does pharmacological manipulation of the STAT3, ZDHHC20 axis reduce tumour progression?

Authors' Response: We would like to express our sincere thanks to the Reviewer for the constructive and positive comments. And we agree that it is important to construct pancreas-specific deficiency of ZDHHC20 or YTHDF3 PDAC mice model to examine the biological effect of these proteins in pancreatic cancer. In fact, our research team has spent fee about two years constructing pancreas-specific defective ZDHHC20 KPC mice. But unfortunately, we have not succeeded until now because of such multiple mutations and extremely difficult breeding. And we also supplemented this limitation in our discussion. Therefore, we used AAV-shZDHHC20 for ZDHHC20 knockdown in KPC mice and cell-derived xenograft (CDX) model to explore the biological function of ZDHHC20 in pancreatic cancer. Despite the difficulties, we will try to establish the pancreatic-specific ZDHHC20-deficient KPC mouse models in future studies.

“Furthermore, to fully understand the biological role of ZDHHC20 and YTHDF3 in pancreatic cancer, future studies using pancreas-specific deficiency of ZDHHC20 or

YTHDF3 KPC mice model are warranted.”

For pharmacological manipulation of the STAT3–ZDHHC20 axis, we employed the CDX model to see the specific effects in pancreatic cancer. The knockdown or inhibitor of STAT3 also observably suppress tumor growth in vivo, but the combination with shZDHHC20 can't further enhance this effect. This may suggest that STAT3 plays a promoting role in pancreatic cancer progression, perhaps in part by inducing overexpression of ZDHHC20.

Figure S5L-S5Q

(b) suboptimal demonstration of the role of STAT3, ZDHHC20 and YTHDF3 in the progression of PDAC.

(1) Does STAT3 overexpression upregulate ZDHHC20?

Authors' Response: We thank the Reviewer for raising this great point. As suggested, we carried out RT-qPCR and western blot analysis in BxPC-3 and CAPAN-1 cells. And the results showed that overexpression of STAT3 resulted in significantly upregulation of ZDHHC20 in pancreatic cancer cells (Figure S4G and S4H).

Figure S4G and S4H

(2) Does ZDHHC20 overexpression affect KRas and/or STAT3 signaling?

Authors' Response: Thanks for these valuable comments. As Reviewer suggested, we carried out the western blot in BxPC-3 and CAPAN-1 cells with ZDHHC20 overexpression. The data indicated that ZDHHC20 overexpression didn't affect the expression of KRAS and STAT3, but slightly increased the phosphorylation of STAT3. This phenomenon might suggest that there is a potential feedback regulation between STAT and ZDHHC20, which might be one of the reasons why ZDHHC20 is overexpressed in pancreatic cancer. According to the recent findings of Bowen Hu et al, YTHDF3 enhanced the stability and translation of m6A-modified EGFR mRNA and stimulated hepatocellular carcinoma progression via the YTHDF3/m6A-EGFR/STAT3 and EMT pathways⁷. In addition, PI3K heterodimers have been reported to bind palmitoylated EGFR, modified by ZDHHC20, to activate PI3K-AKT signaling in non-small cell lung cancer⁸. Combined with these findings, we hypothesize that there is a potential positive feedback regulation between ZDHHC20 and STAT3 in pancreatic cancer, perhaps mediated by EGFR. However, we are very sorry that we have not yet investigated the exact mechanism. More research work is needed to support this possible conjecture in the future.

(3) Although BxPC-3 cells do not have mutant KRas, the authors observed the same effect in BxPC-3 as in PanC1, which suggests that the role of KRas does not have a genuine role in activating the axis ZDHHC20-STAT3. The authors should clarify this.

Authors' Response: We thank the Reviewer for this great question. It is really true as Reviewer mentioned that BxPC-3 cells without KRAS mutant also presented higher expression of ZDHHC20 than that in normal pancreatic epithelial cells. Our results only demonstrated that KRAS mutations were one of major causative factors for STAT3-ZDHHC20 axis hyperactivation in pancreatic cancer. And we have clarified this in the manuscript.

“Taken together, these findings indicate that KRAS mutations are one of major causative factors for STAT3-ZDHHC20 axis hyperactivation in pancreatic cancer.”

(4) The mechanistic link with the lysosomal degradation pathway needs to be supported by more and more precise data.

Authors' Response: We would like to express our sincere thanks to the Reviewer for the constructive and positive comments. To further explore the potential mechanisms for the decreased lysosomal localization of YTHDF3 after S-palmitoylation, we conducted IP-MS between the YTHDF3 WT and YTHDF3 C474S. The IP-MS assay showed that HSC70, a molecular chaperone recognized soluble cytosolic proteins bearing a KFERQ-like motif and mediated the translocation of the substrate into the lysosome for lumen degradation⁶, showed more interactions with the YTHDF3 C474S relative to the YTHDF3 WT (Figure S8E and S8F). Analysis of the YTHDF3 amino acid sequence revealed the presence of three potential KFERQ-like motifs, suggesting that YTHDF3 could be a chaperone-mediated autophagy (CMA) substrate (Figure S8G). Consistently, the Co-IP assays also corroborated more interactions of HSC70 with YTHDF3 C474S (Figure 6J). Notably, we found that YTHDF3 protein levels were downregulated in ZDHHC20-KO pancreatic cancer cells, and the knockdown of HSC70 or LAMP2A reversed the effects of ZDHHC20 deficiency on YTHDF3 protein levels in ZDHHC20-KO cells (Figure S8H and S8I). In conclusion, ZDHHC20-mediated palmitoylation of YTHDF3-Cys474 suppressed the recognition of YTHDF3 by HSC70, thereby inhibiting its subsequent lysosomal degradation through the CMA pathway.

Figure 6J and S8E-S8I

(5) The author should demonstrate that in PDAC samples, there is a correlation between ZDHHC20, STAT3, and YTHDF3 staining in cells high in MYC levels.

Authors' Response: Many thanks for providing these references and suggestions. We have supplemented the corresponding results and the data demonstrate that there is a correlation between ZDHHC20, p-STAT3, and YTHDF3 staining in cells with high MYC levels in PDAC samples, as follows.

Figure 8A and S10D-S10F

Minor points

Figure 1E Which is the gene displayed in the graph in e)?

Authors' Response: Thank you for this comment. We are very sorry for our negligence of the title in this graph and have added it in Figure 1E.

Figure 1E

Figure 1H the Log₂ fold change for ZDHHC20 is only 1.004, whereas the protein expression by IHC is much larger than this. The author should comment on the discussion since it calls for a post-transcriptional modification of ZDHHC20.

Authors' Response: We really appreciate for these thoughtful comments. As Reviewer suggested, we comment on the potential post-transcriptional modification of ZDHHC20 in the discussion, as follows.

“When analyzing ZDHHC20 in pancreatic cancer tissues compared to adjacent nontumor tissues, we found that the protein expression difference of ZDHHC20 by IHC is more significant than the mRNA expression difference from the GSE16515 dataset. Phosphorylation, ubiquitination, and palmitoylation modifications have been reported for other ZDHHCs^{9,10}, and we conjecture that post-translational modifications of the ZDHHC20 protein in pancreatic cancer may lead to the significance of the mRNA-protein expression difference and may further amplify its carcinogenicity. Meanwhile, when evaluating the prognosis significance of all ZDHHCs in pancreatic cancer, we found great variability in the survival rate of the patients based on the expression of different ZDHHCs. It may suggest the complexity of different ZDHHCs regulating the biological functions of pancreatic cancer¹,

and we will also further explore the post-translational modifications and functions of ZDHHCs in pancreatic cancer and try to elucidate this complex regulatory network.”

Figure 1K Graph in K lacks the statistical test and does not provide the number of replicates in each condition.

Authors’ Response: Thank you for this comment. We have added the statistical test and the number of replicates in Figure 1K.

Figure 1K

Figure 2C The graph lacks the statistical test and does not provide the number of replicates in each condition.

Authors’ Response: We really appreciate for this thoughtful comment. Recently Yao W, et al reported the comparison of KRAS-ON with KRAS-OFF paired samples identified 196 upregulated proteins, including ZDHHC20³. We presented and cited the expression ratio of ZDHHC20 in KRAS ON/OFF pancreatic cancer cells from that study. We apologize for choosing an inappropriate graphic to display the results and have corrected it in Figure S4A.

A

Accession	Gene	AK10965	AK196
		Ratio_KRAS ON/OFF	
Q5Y5T1	ZDHHC20	1.799	1.513

Figure S4A

Figure 2E) Please define which PanINs (1,2,3) were included in the analyses.

Authors’ Response: Thank the Reviewer for mentioning this point. We evaluated the IHC

score of ZDHHC20 in normal, PanINs (PanIN 1-3) and pancreatic tissues in KPC mice. And we have defined it and rephrased the caption of Figure 2D.

“Figure 2 (D) IHC score of ZDHHC20 in normal, PanINs (PanIN 1-3) and pancreatic tissues in KPC mice. n=9, unpaired t test.”

Figure 2F) It would be interesting to know if the authors show a similar downregulation of ZDHHC20 upon treatment with inhibitors for other kinases.

Authors’ Response: We thank the Reviewer for the great suggestion. Considering this suggestion, we have detected the level of ZDHHC20 in PANC-1 and AsPC-1 cells with inhibitors for other kinases. Interestingly, the western blot presented a similar downregulation of ZDHHC20 upon ERK inhibitor (Trametinib) and AKT inhibitor (MK2206) treatment, as follows. And the potential regulatory mechanisms and biological phenomena involved will be further explored in our future studies.

Figure 2M) Please add the genomic sequence of the amplified region (distance from TSS).

Authors’ Response: We thank the Reviewer for pointing this out. The primers were designed based on the binding peak, and the ChIP–qPCR results demonstrated that STAT3 can bind to the promoter region of ZDHHC20. And we added the genomic sequence of the amplified region (-670 bp, +932 bp, from TSS), as follow.

```
>hg38_dna    range=chr13:21458700-21460300    5'pad=0    3'pad=0    strand=+
repeatMasking=none
```

```
ACAACCTAAACCTTCTAAGAAGCTCGACTACTCCCGTTCTGCACGTTCCCTTCCCAG
AAGGATGCAGACCCCGGTGTCGGCGGGAACCAGGAAGCTCTTGGGGCCTCCCT
GGGCCAGACCTGCACCCGGAAGCCGCGCAAGGCTTCAGGAGCCCCAGGTCGC
CCCAAGGCCGCGCCGACGATTGAACAAAGCGGCGGGTGCAGCAACCTCGGCGC
TTCCGAGCGCGTTCGGGGCCGACGCCCCGGCGTCCGGCGCTGGCTCCAGAAAGG
CGGCGGGTGTGGGGCGCAGAGGCCTATCTCGCGCCCTAGCCGCGGCCCGCGCCCC
GCCGACGTCCTCCGGGGACGGTACTCACACACGCAGAGCTCCACCACGTACGCGTA
GTAGGACCAGACGACCACGAAGGTGATGAAGAGCACCGGCACCCAGCCCACGAC
GCGCTGGCAGCAGCGCCACAGCGTCCAGGGCGCCATGTTCCGCTGGCGGCTGCCG
AGCCCCGCGTCCACCGTTCTGGGGAGCGCGGGAGCCCCGCGACGGTGACTCG
GACGCTCCAGGCGGCTGCTGGTCCAGCTCCCCCGCTCCGAGGCAGGACTTGTGG
GAGCAAAAGTCCGAGGCGCCGCGGGACTCCTCCCGTCCCGGCGCCCAGCCAGG
CCCCGCCCCACCCACATCCCGTCGCCCCCTCCCCCGCGTCCGTCCTCCCGCCAG
```

CCCGCCCGCCGCGGGCCGCGGGCCGCCCCGGTCTCCGCGTCGTTGGGGCGGGGT
TTCCGCGGCCGGGGAGGCGGTGAGGGGGCGGGGCCGACGCTCGGGACCGCCCAT
GGGGCGTGGCGGGGCTGGATGCGGACAGGCTGGGCGGACGCGGGAGGCCGGGG
AAGGAGAGGCGAAAAACACGCCCTTGATGGATGAGAGGGACAGGGGATAGCTTT
TGTGGATCGAGACTCAATAGGGGAGGGAGACTATGGAAATGATTCCTTAGCTGATT
TGTGGATCTTGGCCCTCTCTGTGTACAGAACTCGCTCCACTGGGGTTCCCTGGGACA
TTGAGTAGTTGGGGCACTGTTTCAAACATGACAAGGCCATTCCCCACTCAGTGTGC
GAAGAATGCACATTGTTAGAAGGCTCCTCCTGTGATTTTTATGTTTCGTCCATCTTGG
CGGAACCCCTCCCTTTCTTGAAGATGGAACGGTAGATACCTGTAATTCGGGTAGA
AGCCATGTAGGGTCCATCAGTTTGATTCGTAACCCAAATTTAAGATTGAGTAGCGA
GAAGGAAAGGGCATGTAAATGACCCTCTCCTTGCCCCGCAAACGACTAAGGAAAT
TTAAAGAATTTATTTACAACACTAGAGGTTGCTAAGCAATGACGATAATCATATAGAAC
CCAGAAGCATCACTGTGAAGACTTCACAATGCAAGGTCAGACGCATTAGAAGGGA
TTCCAGACCGCAATCTTCATACCAGTTCCTGCTACTAGCATTAGTTTTGGTTAGAA
TGAATTTTTGTGTTTGTTTTTTGAACAGGGTCTTCTCGCTCTGTCCGCAAGCTG
GAGTGCAGTGGCACGACCTGGGCTCATTAAAGCCTCGACCTCCCTGCTAAAGCAC
TCC

Figure 4A) which cell line was used to perform the IP-MS? Provide more details of the number of peptides identified in the analyses and the enrichment of the bait protein vs the others.

Authors' Response: Thank you for this comment. We are very sorry for our negligence and have added this description in the caption. This mass spectrometry analysis of Flag-ZDHHC20-immunoprecipitates was carried out in PANC-1 cells. More details of our IP-MS have been supplemented in Table S4 and S5.

Figure 4H) The image is blurry and not clear. Please provide the negative control of the PLA experiment. PLA is known to give a lot of false positive results.

Authors' Response: Thanks for these valuable comments. We agree that assays such as PLA experiment could easily have false positives and have added the negative control of the PLA experiment and improved the picture clarity.

Figure 4I

Figure 6C) Repeat the same experiments using other Lysosomotropic drugs such as Chloroquine and Leupeptine. Perform LysoTracker staining and quantify by cytometry. Lysosomotropic drugs can activate or reduce lysosomal activity depending on the dose and length of the treatment.

Authors' Response: Many thanks for providing these references and suggestions. As the Reviewer suggested, we repeated the same experiment with Chloroquine and Leupeptin and obtained similar results with NH₄Cl and bafilomycin A1. 2-BP treatment or knockout of ZDHHC20 caused destabilization of YTHDF3, which was rescued by these lysosomal inhibitors. And we also carried out the LysoTracker staining to see lysosomal activity. These Lysosomotropic drugs significantly suppress the fluorescence intensity of LysoTracker, except for Leupeptin. It is not clear to us whether this phenomenon with leupeptin is due to its different pharmacological mechanism from that of other lysosomal inhibitors.

Figure S7C-S7I

Figure 6E) LysoTracker is not a sensitive marker for Lysosomes by IF. Repeat the IF using LAMP-1 or LAMP-2, which works well by IF. We have tried this antibody, which works well in human cells (CST, D2D11).

Authors' Response: We thank the Reviewer for these helpful suggestions. As the Reviewer suggested, we repeated the immunofluorescence staining assay with anti-LAMP1 (truly well). Consistently, the ZDHHC20 knockout or C474S mutant upregulated the co-location of YTHDF and LAMP1.

Figure S8A and S8B

Figure 8 Since the role of MYC in pancreatic cancer is commonly associated with the hyperproliferation of tumour cells, please confirm that the tumours that are high for ZDHHC20, YTHDF3 and MYC are also higher for Ki67.

Authors' Response: We thank the Reviewer for raising these excellent points. We evaluated the expression levels of ZDHHC20–YTHDF3–MYC axis and Ki67 in PDAC tissue microarrays by IHC staining and observed consistently positive correlations (Figure 8A sand S10D).

Figure 8A and S10D

SF8 F) Provide more information about the tool used to generate this graph. If a public website, please add the link.

Authors' Response: Thank you for this comment. This graph which showed the Gene Ontology (GO) enrichment analysis result of the ZDHHC20 RNA-seq was drawn by an online platform OmicShare under the default instructions (<https://www.omicshare.com/>). And we added this description in Supplementary Material and Methods.

“Gene Ontology (GO) enrichment analysis result of the RNA-seq was drawn by an online platform OmicShare (<https://www.omicshare.com/>), under the default instructions.”

References

The authors may consider adding the following references

role of MYC in pancreatic homeostasis and PDAC initiation (PMID: 28159836).

Palmitoyl transferases act as novel drug targets for pancreatic cancer (PMID: 37038141).

IP-MS of protein in total pancreas lysates (PMID: 37353485).

Authors' Response: Thank you for this comment. We have cited these references in corresponding part as the Reviewer suggested.

Special thanks to you for your good comments.

References

1. Lin Z, Lv Z, Liu X, Huang K. Palmitoyl transferases act as novel drug targets for pancreatic cancer. *J Transl Med* 21, 249 (2023).
2. Mukai K, et al. Activation of STING requires palmitoylation at the Golgi. *Nat Commun* 7, 11932 (2016).
3. Yao W, et al. Syndecan 1 is a critical mediator of macropinocytosis in pancreatic cancer. *Nature* 568, 410-414 (2019).
4. Li Y, Bedi RK, Moroz-Omori EV, Caflisch A. Structural and Dynamic Insights into Redundant Function of YTHDF Proteins. *J Chem Inf Model* 60, 5932-5935 (2020).
5. Ni W, et al. Long noncoding RNA GAS5 inhibits progression of colorectal cancer by interacting with and triggering YAP phosphorylation and degradation and is negatively regulated by the m(6)A reader YTHDF3. *Mol Cancer* 18, 143 (2019).
6. Kaushik S, Cuervo AM. The coming of age of chaperone-mediated autophagy. *Nat Rev Mol Cell Biol* 19, 365-381 (2018).
7. Hu B, Gao J, Shi J, Wen P, Guo W, Zhang S. m(6) A reader YTHDF3 triggers the progression of hepatocellular carcinoma through the YTHDF3/m(6) A-EGFR/STAT3 axis and EMT. *Mol Carcinog* 62, 1599-1614 (2023).

8. Kharbanda A, Walter DM, Gudiel AA, Schek N, Feldser DM, Witze ES. Blocking EGFR palmitoylation suppresses PI3K signaling and mutant KRAS lung tumorigenesis. *Sci Signal* 13, (2020).
9. Sun Y, et al. AMPK Phosphorylates ZDHHC13 to Increase MC1R Activity and Suppress Melanomagenesis. *Cancer Res* 83, 1062-1073 (2023).
10. Abrami L, et al. Identification and dynamics of the human ZDHHC16-ZDHHC6 palmitoylation cascade. *Elife* 6, (2017).

REVIEWER COMMENTS

Reviewer #2 (Remarks to the Author):

The authors have satisfactorily addressed all my concerns.

I would suggest the authors include the HuR and IGF2BP1 data in rebuttal in the manuscript as supplementary data.

Reviewer #3 (Remarks to the Author):

The authors have addressed all of my previous questions, and greatly improved the manuscript. I am now supporting the publication of this paper.

Reviewer #4 (Remarks to the Author):

The authors have addressed my concerns. I recommend acceptance of the manuscript for publication in Nature Communications.

Reviewer #5 (Remarks to the Author):

The newest version of the MS provides sufficient data to answer my comments. I believe the MS is ready to be published in Nature Communications. Thank you to Nature Communications for giving me the opportunity to review this work. Congratulations to the authors.

Reviewer #6 (Replacing Reviewer #1, Remarks to the Author):

Zhang and colleagues have successfully addressed most of the reviewer's original concerns. However, one comment still remains questionable to this reviewer. In addition, this reviewer has a few minor comments.

1. Response to the original comment #1.3: During revision, the authors conducted immunoprecipitation experiments using a variety of ZDHHCs (Fig. 4H) and concluded that YTHDF3 was not a target of any other ZDHHC proteins in PANC-1 cells (Fig. 4H). The present data look great. However, the western blotting showing comparable levels of immunoprecipitations of ZDHHC proteins is missing.

2. Response to the original comment #7: As mentioned in the original comment, the overall connection between YTHDF3 palmitoylation and m6A binding is not fully explored and discussed. The revised manuscript still lacks direct evidence addressing the question. In addition, as pointed out in the original comment #7.2, alternative interpretation is still possible. Although the structural reanalysis is outside of the scope of this manuscript, the authors should show additional experimental evidence supporting their model proposed in Fig. 10. In summary, one important question remains: YTHDF3 palmitoylation increases m6A MYC mRNA. Is this caused by either protein stabilization of YTHDF3 after palmitoylation, or an increased binding ability of YTHDF3 to m6A MYC mRNA after palmitoylation, or both?

3. Throughout the manuscript, the authors described “YTHDF3-mediated m6A modification”. This description is misleading, because YTHDF3 is a reader protein for m6A and thus this protein is not involved in m6A modification. The sentences should be properly revised. For instance, in line 360-366: “substitution of the Cys474 blocked m6A modification mediated by YTHDF3 in the MYC mRNA (Fig. S9I and S9J).”
4. Fig. 7K and 7L: In Figure legends, “(K, L) Genome browser tracks for input and RIP-seq (K) and meRIP-seq (L) data of YTHDF3 from GSE130173.” The authors have reanalyzed the public data. GSE130173 includes the data for MeRIP-seq, which allows for nucleotide positions for m6A modifications. Therefore, this MeRIP-seq is not a specialized seq for YTHDF3. The Figure legends and Fig. 7L should be properly revised. In Fig. 7L, IP1 and IP2 should be changed to m6A peaks or reads.
5. Fig. 7M. MYC-CRD: The authors measured the relative level of MYC mRNAs using specific primers for MYC-CRD. Although they used specific primers to anneal the CRD region, they are measuring MYC mRNA level, but not MYC-CRD. Therefore, the authors’ conclusion, “As expected, we confirmed m6A modification mediated by endogenous YTHDF3 in the MYC CRD in PANC-1 and AsPC-1 cells (Fig. 7M and 7N)”, is not supported by experimental evidence.
6. Fig. 7N: Explain the synthetic MYC-CRD RNA fragment in more detail. There is no information even in the Materials and Methods section.
7. Fig. 7O and 7P: According to authors’ claim, the level of CRD-wt should be higher than that of CRD-mut under vector-transfected conditions. However, the levels look similar. Please explain why.

REVIEWER COMMENTS

Reviewer #2 (Remarks to the Author):

The authors have satisfactorily addressed all my concerns.

I would suggest the authors include the HuR and IGF2BP1 data in rebuttal in the manuscript as supplementary data.

Authors' Response: We thank the Reviewer for the positive comments on our manuscript and great suggestions. Special thanks to the reviewer for these constructive comments, and we supplement the HuR and IGF2BP1 data as the Reviewer suggested.

Figure S9E-S9I

Reviewer #3 (Remarks to the Author):

The authors have addressed all of my previous questions, and greatly improved the manuscript.

I am now supporting the publication of this paper.

Authors' Response: We thank the Reviewer for the enthusiasm and recommendation for the publication of our manuscript without any additional changes.

Reviewer #4 (Remarks to the Author):

The authors have addressed my concerns. I recommend acceptance of the manuscript for publication in Nature Communications.

Authors' Response: We thank the Reviewer for the enthusiasm and recommendation for the publication of our manuscript without any additional changes.

Reviewer #5 (Remarks to the Author):

The newest version of the MS provides sufficient data to answer my comments. I believe the MS is ready to be published in Nature Communications. Thank you to Nature Communications for giving me the opportunity to review this work. Congratulations to the authors.

Authors' Response: We thank the Reviewer for recognizing the novelty and significance of our finding. We also thank the Reviewer for the insightful suggestions that have helped us improve our manuscript significantly.

Reviewer #6 (Remarks to the Author):

Zhang and colleagues have successfully addressed most of the reviewer's original concerns. However, one comment still remains questionable to this reviewer. In addition, this reviewer has a few minor comments.

Authors' Response: We sincerely appreciate the time and effort that you dedicated to providing feedback on our manuscript and are grateful for the insightful comments and valuable improvements to our paper.

1. Response to the original comment #1.3: During revision, the authors conducted immunoprecipitation experiments using a variety of ZDHHCs (Fig. 4H) and concluded that YTHDF3 was not a target of any other ZDHHC proteins in PANC-1 cells (Fig. 4H). The present data look great. However, the western blotting showing comparable levels of immunoprecipitations of ZDHHC proteins is missing.

Authors' Response: We would like to express our sincere thanks to the Reviewer for the constructive and positive comments. As suggested, we have supplemented the western blotting of immunoprecipitations of all ZDHHC proteins. And it can be seen that all Flag-ZDHHCs are efficiently immunoprecipitated as below.

Figure S6B

2. Response to the original comment #7: As mentioned in the original comment, the overall connection between YTHDF3 palmitoylation and m6A binding is not fully explored and discussed. The revised manuscript still lacks direct evidence addressing the question. In addition, as pointed out in the original comment #7.2, alternative interpretation is still possible. Although the structural reanalysis is outside of the scope of this manuscript, the authors should show additional experimental evidence supporting their model proposed in Fig. 10. In summary, one important question remains: YTHDF3 palmitoylation increases m6A MYC mRNA. Is this caused by either protein stabilization of YTHDF3 after palmitoylation, or an increased binding ability of YTHDF3 to m6A MYC mRNA after palmitoylation, or both?

Authors' Response: We really appreciate for this thoughtful comment. It is really true as Reviewer suggested that the binding ability of YTHDF3 to m6A MYC mRNA after palmitoylation should be taken into account. As the original comment #7 mentioned, the mutation of C474 into serine may be altering the conformation of the tryptophan-cage and partially disrupting the ability to bind m6A in a manner. With this in mind, we carried out RIP-qPCR analysis in the case of blocking the lysosomal degradation pathway of YTHDF3 by Barf A1. And we found that even though the lysosomal degradation pathway of YTHDF3 was blocked by Baf A1, the binding of YTHDF3 to MYC mRNA remained significantly different after palmitoylation blockade by 2-BP treatment or ZDHHC20 knockout. These results imply that the palmitoylation modification of Cys474 both stabilizes YTHDF3 and promotes the binding ability of YTHDF3 to m6A MYC mRNA. As discussed previously, though our present experimental results have indirectly demonstrated an increased binding ability of YTHDF3 to m6A MYC mRNA after palmitoylation, more direct evidence in terms of conformation is

needed to explore the underlying mechanisms in future studies.

“Considering that ZDHHC20-mediated palmitoylation occurs in the YTH domain, which has been reported to function in binding to RNA, we further explored whether palmitoylation regulates the YTHDF3-MYC mRNA interaction. In the case of blocking the lysosomal degradation pathway of YTHDF3 by Barf A1, 2-BP treatment or ZDHHC20 knockout still significantly down-regulated the binding of YTHDF3 to MYC mRNA (Fig. S10E and S10F).”

Figure S10E and S10F

3. Throughout the manuscript, the authors described “YTHDF3-mediated m6A modification”. This description is misleading, because YTHDF3 is a reader protein for m6A and thus this protein is not involved in m6A modification. The sentences should be properly revised. For instance, in line 360-366: “substitution of the Cys474 blocked m6A modification mediated by YTHDF3 in the MYC mRNA (Fig. S9I and S9J).”

Authors’ Response: We are very sorry for our incorrect writing. As Reviewer suggested that we rephrased the relevant sentences as below.

“**YTHDF3 stabilizes MYC mRNA in an m6A-dependent manner**”, in line 320.

“As expected, we confirmed m6A modification recognized by endogenous YTHDF3 in the MYC CRD in PANC-1 and AsPC-1 cells (Fig. 7M and 7N).”, in line 359.

“In addition, the RIP-qPCR and MeRIP-qPCR analysis showed that substitution of the Cys474 blocked m6A modification recognized by YTHDF3 in the MYC mRNA (Fig. S10G and S10H).”, in line 378

“It has been reported that several m6A modification-related proteins (IGF2BPs, etc.) promote the stability of MYC mRNA and play an essential oncogenic role in various kinds of cancers and likely also in the YTHDF3- recognized, m6A-dependent regulation of MYC mRNA stability in pancreatic cancer.”, in line 496.

4. Fig. 7K and 7L: In Figure legends, “(K, L) Genome browser tracks for input and RIP-seq (K) and meRIP-seq (L) data of YTHDF3 from GSE130173.” The authors have reanalyzed the public data. GSE130173 includes the data for MeRIP-seq, which allows for nucleotide positions for m6A modifications. Therefore, this MeRIP-seq is not a specialized seq for YTHDF3. The Figure legends and Fig. 7L should be properly revised. In Fig. 7L, IP1 and IP2 should be changed to m6A peaks or reads.

Authors’ Response: We thank the Reviewer for this great question. We apologize for the misdescription and have reworded the result description as follows.

“(K, L) Genome browser tracks for input and RIP-seq of YTHDF3 (K) and meRIP-seq (L) data from GSE130173; Input is indicated in blue and RIP/MeRIP in red, MYC-CRD domain is indicated in yellow.”

Figure 7L

5. Fig. 7M. MYC-CRD: The authors measured the relative level of MYC mRNAs using specific primers for MYC-CRD. Although they used specific primers to anneal the CRD region, they are measuring MYC mRNA level, but not MYC-CRD. Therefore, the authors' conclusion, "As expected, we confirmed m6A modification mediated by endogenous YTHDF3 in the MYC CRD in PANC-1 and AsPC-1 cells (Fig. 7M and 7N)", is not supported by experimental evidence.

Authors' Response: Thank you for this comment. We are very sorry for our negligence of detailed method description. When performing the MERIP assay for specific region, we use an RNA cleavage enzyme mix (CEM) to simultaneously fragment RNA size to be between 200-300 bps and cleave/remove any RNA sequences in both ends of the target m6A-containing sequences without affecting RNA regions occupied by the antibody. Short RNA fragments are generated only bound with anti-m6A antibody. True target m6A-enriched regions can therefore be reliably identified, and high-resolution m6A profiling achieved. On this basis, we designed primers for the MYC-CRD region to detect m6A modification of the CRD region fragment in the MeRIP product, but not other regions of total MYC mRNA.

6. Fig. 7N: Explain the synthetic MYC-CRD RNA fragment in more detail. There is no information even in the Materials and Methods section.

Authors' Response: Thank you for this comment. We have added the information of the synthetic MYC-CRD RNA fragment in the Methods as below.

"Biotin-labelled RNA oligonucleotides containing adenosine or m6A were synthesized by Sangon Biotin (Shanghai, China). The sequence of biotinylated probe was listed in Table S2. RNA probes were denatured at 95 °C for 5 minutes and put on ice immediately. The M-280 streptavidin magnetic beads (Invitrogen, USA) at 25 °C for 2-4 h, and then the cell lysates with Protease/Phosphatase Inhibitor Cocktail and RNase inhibitor added were incubated with MYC probe or oligo probe at 4 °C overnight. The RNA complexes bound to the beads were eluted and extracted and then were quantitated by qRT-PCR, and the RNA-protein binding mixture was boiled in SDS buffer and the eluted proteins were detected by western blotting."

7. Fig. 7O and 7P: According to authors' claim, the level of CRD-wt should be higher than that of CRD-mut under vector-transfected conditions. However, the levels look similar. Please explain why.

Authors' Response: We are very sorry for our negligence of description and have added the description of control group in the figure legend as below. In Figure 7O, the Fluc activity of the Vector group was used as a control for the normalization of the YTHDF3 OE group to show the fold change in Fluc activity by exogenous YTHDF3. That's why the two Vector groups are so similar to each other. Results showed that exogenous YTHDF3 significantly increased the Fluc activity of the wild-type reporter but not the mutant reporters. In Figure 7P, both CRD-wt+IgG and CRD-mut+IgG are consistently close to 0 because IgG antibodies were used as negative controls in the RIP experiments.

“(O) Relative firefly luciferase (Fluc) activity of wild-type (CRD-wt) or mutated (CRD-mut) CRD reporters in PANC-1 cells with ectopically expressed YTHDF3, Vector as relative control. Repeated in triplicates, two-tailed t-tests. (P) RIP-qPCR assay of Luc-CRD by using the IgG or YTHDF3 antibodies in PANC-1 and AsPC-1 cells infected with CRD-wt/CRD-mut plasmids, IgG as negative control. Repeated in triplicates, two-tailed t-tests.”

Figure 7O and 7P

Special thanks to you for your good comments.

REVIEWERS' COMMENTS

Reviewer #6 (Remarks to the Author):

Zhang and colleagues have successfully addressed all of my concerns and comments. I believe that the current version of the manuscript is now suitable for publication.